# The role of föhn winds in eastern Antarctic Peninsula rapid ice shelf collapse

Matthew K. Laffin[1], Charles S. Zender[1,2], Melchior van Wessem[3], Sebastián Marinsek[4]

[1]Department of Earth System Science, University of California, Irvine, USA
[2]Department of Computer Science, University of California, Irvine, USA
[3]Institute for Marine and Atmospheric Research Utrecht (IMAU), Utrecht University, Utrecht, Netherlands
[4]Instituto Antártico Argentino, Buenos Aires, Argentina

*Correspondence to*: Matthew K. Laffin (mlaffin@uci.edu)

**Abstract.** Ice shelf collapse reduces buttressing and enables grounded glaciers to contribute more rapidly to sea-level rise in a warming climate. The abrupt collapses of the Larsen A (1995) and B (2002) ice shelves on the Antarctic Peninsula (AP) occurred, at least for Larsen B, when long period ocean swells damaged the calving front and the ice shelf was inundated with melt lakes that led to large-scale hydrofracture cascades. During collapse, field and satellite observations indicate föhn winds were present on both ice shelves. Here we use a regional climate model and Machine Learning analyses to evaluate the contributory roles of föhn winds and associated melt events prior to and during the collapses for ice shelves on the AP. Föhn winds caused about 25% ± 3% of the total annual melt in just 9 days on Larsen A prior to and during collapse and were present during the Larsen B collapse which helped form extensive melt lakes. At the same time, the off-coast wind direction created by föhn winds helped melt and physically push sea ice away from the ice shelf calving fronts that allowed long period ocean swells to reach and damage the front, which has been theorised to have ultimately triggered collapse. Collapsed ice shelves experienced enhanced surface melt driven by föhn winds over a large spatial extent and near the calving front, whereas SCAR inlet and the Larsen C ice shelves are affected less by föhn wind-induced melt and do not experience large-scale melt ponds. These results suggest SCAR inlet and the Larsen C ice shelves may be less likely to experience rapid collapse due to föhn-driven melt so long as surface temperatures and föhn occurrence remain within historical bounds.

## 1 Introduction

Ice shelves, the floating extensions of grounded glaciers, subdue the discharge of grounded ice into the global ocean (Rignot et al., 2004; Scambos et al., 2004; Gudmundsson et al., 2013; Borstad et al., 2016). Re-examination of past ice shelf collapse events can help to shed light on the mechanisms of collapse and improve the understanding of ice shelf dynamics for future projections of sea-level rise (Rignot et al., 2004; Gudmundsson et al., 2013; Borstad et al., 2016). The final collapses of the Larsen A (LAIS) in 1995 and the Larsen B (LBIS) ice shelves in 2002 have been attributed to decreased structural integrity brought on by a combination of factors. Most notably, regional atmospheric warming (Scambos et al., 2000; Mulvaney et al.,

2012), extended melt seasons (Scambos et al., 2003), multi-year firn pore space depletion (Kuipers Munneke et al., 2014; Trusel et al., 2015), melt pond flooding (Glasser and Scambos (2008); Trusel et al., 2013; Leeson et al., 2020), crevasse expansion through hydrofracture (Scambos et al., 2003; Banwell et al., 2013; Pollard et al., 2015; Alley et al., 2018; Banwell et al., 2019; Robel and Banwell, 2019), glacier structural discontinuities (Glasser et al., 2008), basal melt (Pritchard et al., 2012; Rignot et al., 2013; Depoorter et al., 2013; Schodlok et al., 2016; Adusumilli et al., 2018), warm melt-water intrusion (Braun et al., 2009), melting of the ice melange within rifts conducive to rift propagation (Larour et al., 2021), and regional sea ice loss allowing ocean swell flexure stress on the calving front (Banwell et al., 2017; Massom et al., 2018).

    While the list of mechanisms that can destabilise ice shelves is extensive, a conceptual model for rapid ice shelf collapse proposed by Massom et al., (2018) identifies 4 essential prerequisites for sudden collapse: (1) extensive surface flooding and hydrofracture; (2) reduced sea ice or fast ice at the ice shelf front; (3) outer margin or terminus fracturing and rifting; and (4) initial calving trigger at the ice shelf margin. They theorise waves led to calving front damage and small calving events that breached the "compressive arch" of stability of both ice shelves proposed by Doake et al., (1998). At the same time the ice shelves were covered in extensive surface melt lakes that were unlikely to drain horizontally because of the relatively flat surface (Banwell et al., 2014). Satellite observations and ice shelf stability model studies determined the LBIS was covered with >2750 melt lakes that were on average 1 meter deep before collapse (Glasser and Scambos (2008); Banwell et al., 2013). Ice shelves inundated with surface melt lakes are susceptible to disintegration through a process known as hydrofracture, where meltwater applies outward and downward pressure to the walls and tip of crevasses that can propagate through the ice shelf (Scambos et al., 2003; Banwell et al., 2013; Bell et al., 2018; Lhermitte et al., 2020). Furthermore, melt lakes that rapidly drain by hydrofracture can create fracture patterns that split ice shelves into sections with aspect ratios that support unstable rollover, and hydrofracture cascades that begin when melt lakes drain and/or calving occurs at the ice shelf terminus (Scambos et al., 2003; Banwell et al., 2013; Burton et al., 2013; Robel and Banwell, 2019). The combination of ocean swell stress on the calving front and extensive melt ponds led to large-scale hydrofracture cascades that proposed by Massome et al., (2018) ultimately caused the rapid collapse of the LBIS and possibly the LAIS.

    In addition to a lack of sea ice and extensive melt ponds, meteorological and satellite observations identify clear skies and warm west/northwest föhn wind at the time of collapse (Figure 1c-f) (Rott et al., 1998; Rack and Rott (2004); Cape et al., 2015; Massom et al., 2018). Föhn winds form when relatively cool moist air is forced over a mountain barrier, often leading to precipitation on the windward side of the barrier that dries the air mass (Grosvenor et al., 2014; Elvidge et al., 2015). As the now drier air descends the leeward slope it warms adiabatically and promotes melt directly through sensible heat exchange, and indirectly by the associated clear skies that allow additional shortwave radiation to reach the surface in non-winter months (Turton et al., 2017, 2018; Kuipers Munneke et al., 2018; Elvidge et al., 2020; Laffin et al., 2021). Föhn winds and their capacity to cause surface melt have been studied extensively on the AP. Observations and model studies on the LCIS confirm the föhn mechanism that enhances sensible heat and shortwave radiation and alters local albedo which can

increase surface melt rates upwards of 50% compared to non-föhn conditions (Cape et al., 2015; Elvidge et al., 2015; King et al., 2015, 2017; Kuipers Munneke et al., 2012, 2018; Bevan et al., 2017; Lenaerts et al., 2017; Datta et al., 2019; Kirchgaessner, et al., 2021; Laffin et al., 2021, Wang et al., 2021). Late season föhn melt reduces firn pore space, and thus pre-conditions ice shelves to form melt ponds and are responsible for the increased firn density pattern east of the AP mountains on the LCIS (Holland et al., 2011; Kuipers Munneke et al., 2014; Datta et al., 2019). Föhn melt climatology studies have aimed to identify how much melt is caused by föhn and the locations most affected and found föhn winds account for up to 17 % of melt and are concentrated in the LCIS inlets (Turton et al., 2017; Datta et al., 2019; Laffin et al., 2021). Pressure gradient differences across the AP range lead to föhn winds that funnel through mountain gaps as highly concentrated föhn jets, particularly in inlets east of the AP range (Luckman et al., 2014; Elvidge et al., 2015; Kuipers Munneke et al., 2012; Grosvenor et al., 2014). In addition to enhancing surface melt rates, fohn winds exert force on sea/fast ice and drag it away from the calving front, thereby exposing the front to ocean waves (Bozkurt et al., 2018). Climatic studies of the Larsen B embayment indicate that föhn winds were coincident with collapse (Rack and Rott (2004); Leeson et al., 2017). However, it is unknown if concentrated föhn jets spilled onto the former LAIS and LBIS and, if so, whether those föhn winds contributed to their collapse. The questions, therefore, arise: 1) To what extent did föhn-induced melt contribute to the surface melt budget on each eastern AP ice shelf?; 2) Did föhn winds and associated melt play a role in triggering the collapses of the LAIS and LBIS?; 3) What are the implications of föhn-induced melt for the remaining eastern AP ice shelves?

To address these questions we consider three metrics: Section 3.1 explores the total annual surface melt quantity induced by föhn winds and how melt is spatially distributed across each ice shelf; Section 3.2 identifies the coincidence of föhn-induced melt preceding and during the collapse events, and the estimated melt-lake depth in response to melt events.; Section 3.3 identifies the contribution of föhn melt to the climatological surface liquid water budget comparing collapsed and extant ice shelves on the eastern AP. By constructing a timeline of melt and melt mechanisms and comparing melt metrics with collapsed and extant ice shelves, we can identify the contributory factors to collapse.

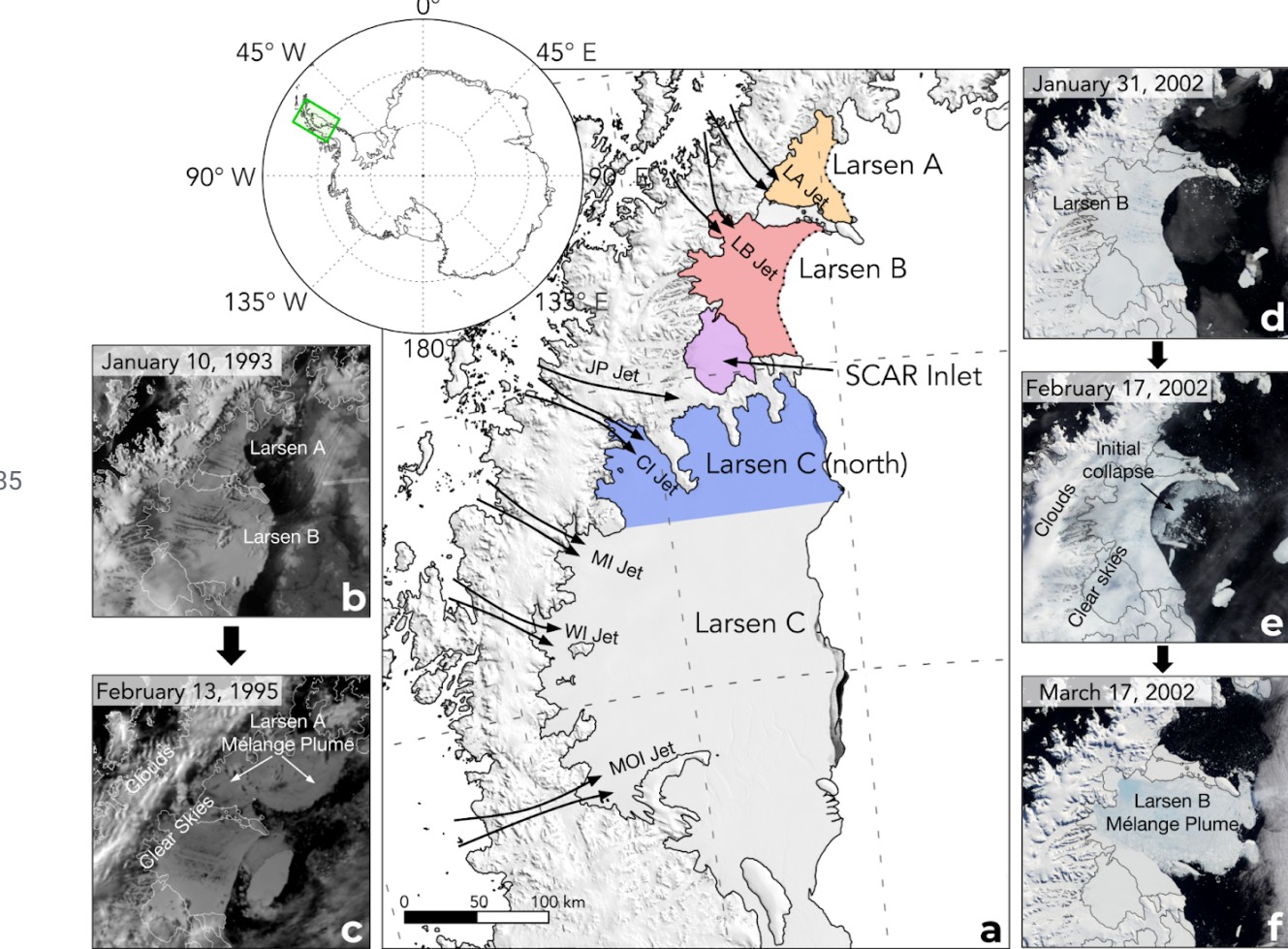

85

**Figure 1**. Map of the northern Antarctic Peninsula (a) showing locations of collapsed ice shelves (LAIS-January 25, 1995, LBIS-February 9, 2002), extant ice shelves (SCAR inlet and LCIS), and föhn jets (Larsen A jet (LA jet), Larsen B jet (LB jet), Jason Peninsula jet (JP jet), Cabinet inlet jet (CI jet), Mill inlet jet (MI jet), Whirlwind inlet jet (WI jet), Mobil Oil inlet jet (MOI jet)) with a MODIS Mosaic overlay. The colored regions indicate how this study separates ice shelves for climatic analysis. The dotted lines show the former extent of the Larsen A and Larsen B ice shelves at the time of collapse. Panels (b)-(f) are satellite images of the collapses of the LAIS and LBIS. (b) AVHRR (Advanced Very High-Resolution Radiometer) image of the northern AP two years before the collapse of the LAIS showing melt lakes on the surface of both ice shelves. (c) AVHRR image after the collapse of the LAIS. (d) NASA provided MODIS (Moderate Resolution Imaging Spectroradiometer) image showing the LBIS days before collapse began. (e) MODIS image showing a föhn wind event (clouds over the western AP, clear skies over the ice shelves) along with the initial collapse of the LBIS. (f) MODIS image of the complete collapse of the LBIS.

## 2 Data and methods

### 2.1 Regional Climate Model 2 Simulation (RACMO2)

We base our analysis on 3-hourly output from simulations by the Regional Atmospheric Climate Model 2 (RACMO2), version 2.3p2, with a horizontal resolution of 5.5km (0.05°) focused on the AP from 1979-2018. RACMO2 uses the physics package CY33r1 of the ECMWF Integrated Forecast System (IFS) (https://www.ecmwf.int/en/elibrary/9227-part-iv-physical-processes\textit{{ECMWF-IFS,} 2008}) in combination with atmospheric dynamics of the High-Resolution Limited Area Model (HIRLAM). When RACMO2 surface simulations are compared with AWS observations on the LCIS, surface air temperature has a slight warm bias likely because of model resolution and shortwave/longwave radiation are over/under estimated due to underestimation of clouds and moisture but overall reproduce surface observations (King et al., 2015; Leeson et al., 2017; Bozkurt et al., 2020; Laffin et al., 2021)

### 2.2 Föhn wind detection

We use the Föhn Detection Algorithm (FöhnDA) that identifies föhn winds that cause melt using 12 Automatic Weather Stations (AWS) on the AP previously developed and detailed in Laffin et al., (2021). FöhnDA identifies föhn-induced melt events using binary classification Machine Learning when 10 metre air temperature (T) is greater than 0°C, which ensures it captures föhn events that cause surface melt. Thresholds for relative humidity (RH) and wind speed (WS) are more dynamic because high wind speeds and low relative humidity do not guarantee temperatures above freezing, they only aid to identify föhn. FöhnDA uses quantile regression to identify these variable thresholds that take into account the climatology and seasonality at each AWS site. FöhnDA uses two empirically determined thresholds: the 60th percentile wind speed and 30th percentile relative humidity which are 2.85 m/s and 79% averaged at all AWS locations. We co-locate AWS with the nearest model grid cell and use FöhnDA results to train a ML model that detects föhn winds in RACMO2 output. Our ML model improves the accuracy of föhn detection by over 23% when compared to the simple binary classification method applied to RACMO2 output as described above. A sensitivity study detailed in Laffin et al., (2021) compares previous föhn detection methods (Cape et al., 2015; Datta et al., 2019) and shows that FöhnDA allows us to use in situ observations from AWS and expand föhn detection with RACMO2 output to regions and times when AWS observations are not available (Figure S1) (Table S1).

Föhn jet locations were identified using wind direction and strength during föhn events (Figure 2a) and by the surface melt pattern during föhn (Figure 3b). The RACMO2 topography pixel size is 5.5 km which is sufficient to produce the föhn jets identified on the LCIS (Elvidge et al., 2015), and allows for new föhn jet identification on the LAIS and LBIS despite lack of direct observation. However, föhn winds funnelled through local canyons and mountain gaps smaller than 5.5

km are not directly simulated. Therefore, we consider RACMO2 simulated estimates of surface melt caused by föhn winds
to be conservative and likely greater in regions where föhn winds are funnelled and concentrated.

## 2.3 Ice shelf intercomparison analysis

We split the ice shelves into areas shown in Figure 1a (Larsen A, Larsen B, SCAR inlet, Larsen C (north), and Larsen C) and
take the average of all model grid cells annually to create a climatology of surface melt, melt rate, melt hours, surface
temperature. We use a two-tailed t-test statistic to identify if the mean surface temperature and mean surface melt of both ice
shelves are statistically different from one another at the 95% confidence interval. We compare all ice shelves to the LBIS
because it was the most recent collapse event and is adjacent to collapsed and existing ice shelves. Qualitatively similar
results are obtained when comparing all ice shelves to the LAIS.
To compare ice shelf liquid water budgets we use a liquid-to-solid ratio (LSR) as a crude proxy for available firn air
content and can be estimated as,

$$LSR = \frac{Total\ liquid\ water\ (snowmelt + liquid\ precipitation)}{Total\ solid\ precipitation\ (snow)} \tag{1}$$


where areas with LSR < 1 represent an ice shelf that receives more solid precipitation than liquid water and is therefore less
likely to saturate with liquid water and form melt lakes than areas with LSR > 1.

## 2.4 Sea ice concentration analysis

We used 3-hourly meteorological data of sea ice concentration from the European Center for Medium-Range Weather
Forecasts (ECMWF) ERA5 reanalysis (Copernicus Climate Change Service, 2017). These data are available at a horizontal
resolution of about 30 km or 0.28°. ERA5 is created by assimilated satellite and in situ observations into ECMWF's
Integrated Forecast System (IFS). We compare sea ice concentration to the occurrence of föhn wind events to identify how
föhn winds impact sea ice concentration. We measure the mean sea ice concentration of the ocean 90km directly east of each
ice shelf (Larsen A, Larsen B, and Larsen C) in the Weddell Sea. We explore the relationship of summer föhn wind
occurrence and summer (DJF) sea ice concentration using a statistical pearson correlation method. When föhn winds are
present we compare the mean of all sea ice concentration pixels in the designated ice shelf region for all years from 1979 to

150 2018.

## 3 Results

### 3.1 Föhn jets and melt

Using RACMO2 historical simulations, informed by a Machine Learning algorithm (FöhnDA) that is trained with AWS observations (Laffin et al., 2021), we identify seven recurring föhn jets or "gap winds" that lead to high surface melt rates on the eastern AP ice shelves (Figure 2a). Four of these jets (CI, MI, WI, MOI) have been studied using airborne observations and model simulations (Grosvenor et al., 2014; Elvidge et al., 2015). The remaining three jets (LA, LB, and JP) are, to our knowledge, identified here for the first time. Overall, winds from the west and northwest direction lead to increased surface melt rates that can be up to 53% higher than melt when the wind is from other directions (Figure 2c) (van den Broeke (2005)). Additionally, the degree to which föhn winds impact surface melt on each ice shelf varies depending on föhn jet existence, location, and wind strength (Wiesenekker et al., 2018). These variations in fohn jet location may provide insight into why SCAR inlet and the LCIS remain intact while the LAIS and LBIS have collapsed other than the significant difference in annual surface temperature (Cook and Vaughan (2009); Bozkurt et al., 2020; Carrasco et al., 2021).

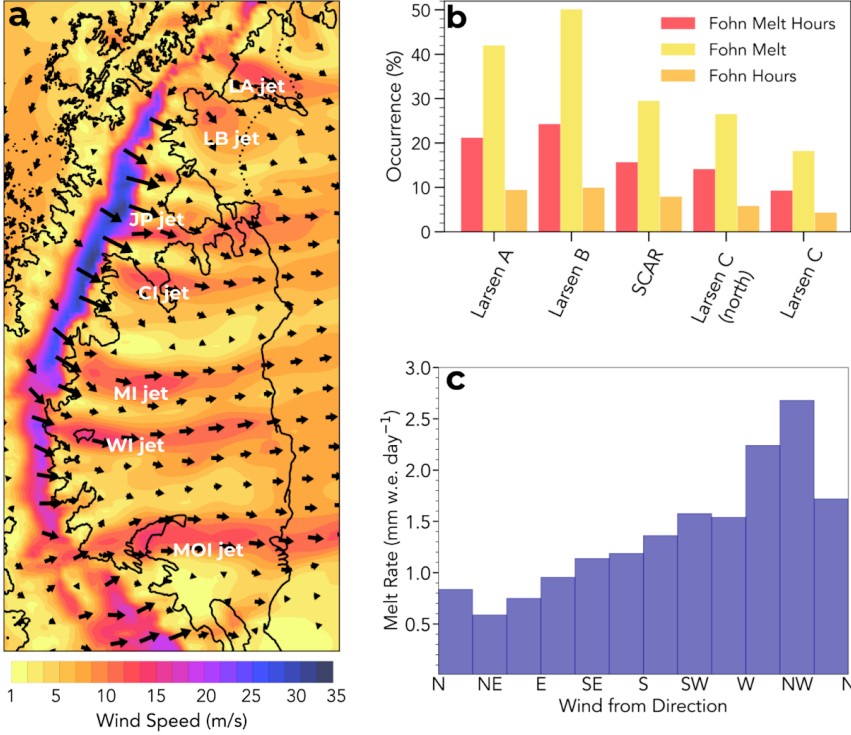

**Figure 2**. (a) The northern AP showing the RACMO2-simulated wind speed and direction vectors on January 24, 1995, just before the collapse of the LAIS. Föhn jet locations are indicated with names. (b) RACMO2 annual average föhn melt hour percent of total melt hours, föhn melt percent of total melt for each ice shelf from, and percent of total hours föhn winds occur from 1980-2002. (c) RACMO2 melt rate as a function of wind direction averaged for all ice shelf regions on the AP from 1980-2002.

Surface melt production is more pronounced under the influence of föhn jets, particularly for the LA and LB jets which produce 35.7% and 31.8% more melt respectively compared to regions not in the path of a föhn jet on each ice shelf (Figure 3). Föhn-induced surface melt accounts for 42% of the total annual melt between 1979 and 2002 on the LAIS and 51% of total melt on the LBIS but only represents 21% and 25% of total melt hours on the LAIS and LBIS (Figure 2b, 3c). In locations directly influenced by föhn jets, the mean annual föhn-induced melt was as high as 61% on the LAIS and 57% on the LBIS of total annual melt. By contrast, föhn-induced melt accounts for only 25% of 1979-2002 total melt on SCAR inlet and 17% on the LCIS. SCAR inlet is not directly impacted by a föhn jet, but still experiences clear skies and weak föhn influence from the overall descending air during föhn events. The LCIS is affected by numerous föhn jets (CI, MI, WI, MOI), accounting for up to 40% of the total annual melt in Cabinet and Whirlwind inlets, decreasing with distance east of the AP mountains. The stark contrast in surface melt amount and fraction caused by föhn winds on collapsed vs. intact ice shelves implicates föhn melt as a contributor to the LAIS and LBIS collapses. A clearer picture of the role of föhns emerges after we examine föhn-induced melt extent and timing.

The spatial distribution and extent of surface melt influence ice shelf stability. Surface melt and melt lakes near the ice shelf terminus can lead to calving front collapse and structural instability for the remaining portion of the ice shelf (Depoorter et al.,2013; Pollard et al., 2015). Consistent with this mechanism, the LA and LB föhn jets impact a large spatial area of the LAIS and LBIS, and reach the ice shelf calving fronts (Figure 3b). SCAR Inlet lacks a strong föhn jet/influence and does not regularly experience large-scale melt lakes even during high melt years (Figure 1b-f). This helps explain why SCAR Inlet is still intact, despite decreased sea ice buttress force and major structural changes observed after the collapse of the LBIS (Borstad et al., 2016; Qiao et al., 2020). LCIS on the other hand is impacted by four major jets and regularly experiences föhn-induced melt lakes, particularly in Cabinet inlet. However, the vast size of the LCIS limits the amount of föhn-induced melt at the terminus. The föhn melt mechanism breaks down by mixing with cold air which reduces the intensity of the föhn jets from their peak at the base of the AP mountains to the calving front (Figure 3b) (Elvidge et al., 2016; Turton et al., 2018). Having established that föhn winds significantly enhanced surface melt overall (Cape et al., 2015; Elvidge et al., 2015; Datta et al., 2019) and at the crucial calving front of LAIS and LBIS, we now examine the timing of föhn-induced melt events relative to collapse.

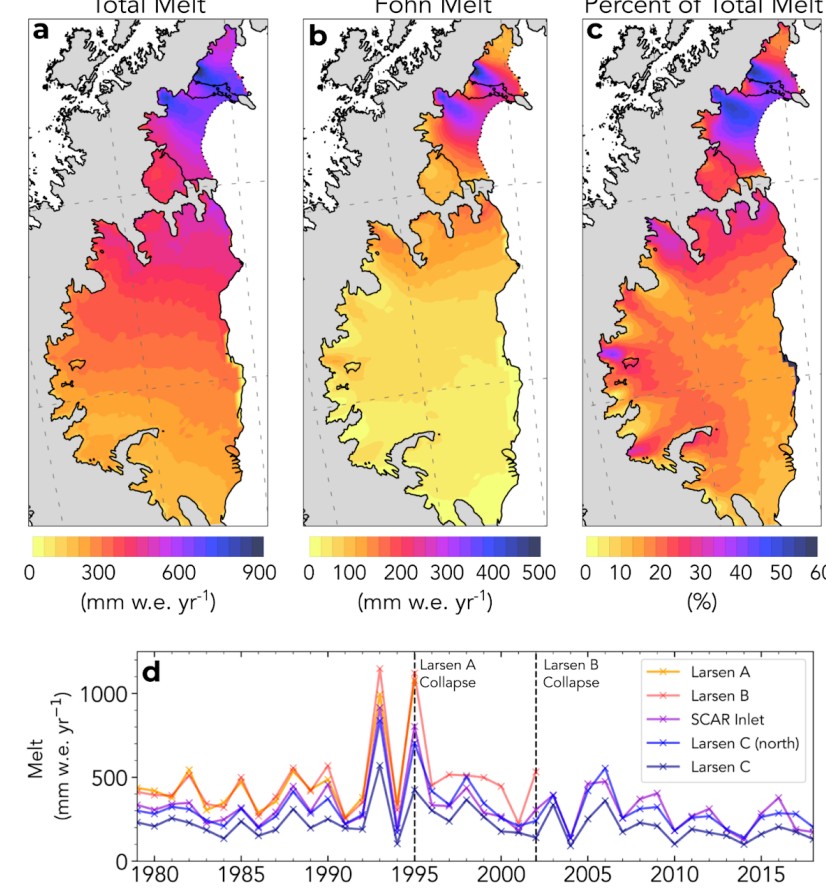

**Figure 3**. (a) RACMO2 average annual melt from 1980-2002. (b) RACMO2 average annual föhn wind-induced melt from 1980-2002. (c) RACMO2 percent of total melt concurrent with föhn wind from 1980-2002. (d) RACMO2 time series of the mean annual surface melt on each ice shelf from 1979-2018. Dashed vertical lines indicate the year in which each ice shelf collapsed. Note: The Larsen B graph often overlaps the Larsen A curve.

## 3.2 Coincidence of föhn winds with collapse

### 3.2.1 LAIS

Three föhn wind events occurred on LAIS between January 18 and 27, 1995, overlapping with the initial phase of the LAIS collapse that began on January 25 (Figure 4b) (Rott et al., 1998). These föhn events helped contribute to the collapse of the ice shelf in two ways: (1) Enhanced surface melt rates caused by the LA jet led to extensive melt lakes across the ice shelf that possibly promoted large-scale hydrofracture cascades because of the rapid (days to weeks) nature of collapse (Banwell et al., 2013); (2) The west/northwest wind direction actively pushed or melted sea ice and fast ice away from the calving front, allowing ocean waves to reach the ice shelf terminus (Rott et al., 1996; Massom et al., 2018). The föhn wind events

prior to and during collapse lasted an average of 3 days each and produced increased surface melt greater than any other
9-day period from 1979-2018, with mean cumulative melt of 268.5 mm w.e. or 25.2% of the total annual melt in the 1994/95
melt season. Total melt during the 1994/95 melt season was 127% higher than an average year (474 mm w.e./yr) and the
9-day föhn wind event produced 57% of the total melt of an average melt year. Therefore this 9-day föhn-induced melt event
and melt year are clearly anomalous in the observational record. We also identify a negative correlation between the
occurrence of föhn winds and sea ice concentration on all eastern AP ice shelves (Figure 5a), that is more correlated with
föhn wind occurrence than air temperature (Figure 5b). When föhn winds occur on the AP, sea ice concentration decreases
which is consistent with other wind types in Antarctica (katabatic winds) that form perennial wintertime polynya (Figure
5c-e)(Bromwich, 1984; Bozkurt et al., 2018; Wang et al., 2021). At the start of the 9-day fohn event, sea ice concentration
east of the LAIS was at or near 100% but by the time collapse began, sea ice concentration dropped significantly (Figure
5d-e).

219        We next examine the contribution of föhn-generated melt to other observables implicated in the collapse, namely

surface liquid water, melt lake depth, and melt lake extent (Scambos et al., 2003). We estimate the spatial extent and depth of
melt lakes prior to collapse on the LAIS using satellite images of melt lake surface area combined with model-simulated
available liquid water volume. The cumulative spatial melt pattern between January 18 and 27, 1995 identifies significant
melt on the LAIS ranging from 157-356 mm w.e. (Figure S2a), varying spatially with the influence of the LA jet. Satellite
imagery of the LAIS during the collapse in progress show melt lakes were present (Figure S3) however because the collapse
had already begun, it is likely many of the lakes had drained or had been altered so estimating melt lake extent is not
possible. However, Advanced Very High-Resolution Radiometer (AVHRR) imagery on December 8, 1992, provides
high-resolution cloudless images of the ice shelf taken at the end of a similar föhn-induced melt event during a year when
melt was comparable to the 1994/95 melt season, therefore we consider this melt lake extent analogous to the 1994/95 melt
season (Figure 4a). We find the melt lake surface area was likely between 5.1%-10.8% ($103$ km$^2$ - $219$ km$^2$) of the total LAIS
surface area (Figure S2b). Melt lake surface area is likely underestimated because the image was taken early in the 1992/93
melt season and does not easily identify small lakes or river systems. Liquid water pooling on the ice surface is modulated by
the local topography. If we assume all the available surface liquid water during the 9-day melt period, minus evaporation,
runoff, and refreeze, forms lakes that cover the same estimated surface area as the 1992/93 melt season, we can estimate melt
lake depth during the initial collapse. We find mean melt lake depth to be between 1.38-6.86 meters depending on lake
location and föhn influence, which exceeds the average lake depth of the LBIS lakes prior to collapse (1m) (Banwell et al.,
2014) and the modelled lake depth (5m) that could lead to large-scale hydrofracture cascades, especially under the influence
of the LA jet (Banwell et al., 2013).

**3.2.2 LBIS**
A föhn wind event coincided with the initial LBIS collapse on February 9, 2002, with two events just prior to collapse and
three additional events before complete collapse by March 17, 2002 (Figure 4c). Föhn events in the LBIS 2001/02 melt
season were relatively short, averaging less than 24 hours per event, and produced melt rates 27% higher than non-föhn melt
that year and 39% of the average föhn melt rate in all other years (Figure 4e). Similar to the LAIS collapse the off-coast wind
direction and enhanced surface melt rates during the föhn wind event helped push sea ice away from the calving front and
contributed to surface melt lakes that led to hydrofracture and collapse (Figure 5a) (Massom et al., 2018). Additionally,
previous high melt rate föhn events such as those in the 1992/93 and 1994/95 melt seasons likely preconditioned the LBIS
through firn densification to support melt lake formation, discussed in section 3.3.

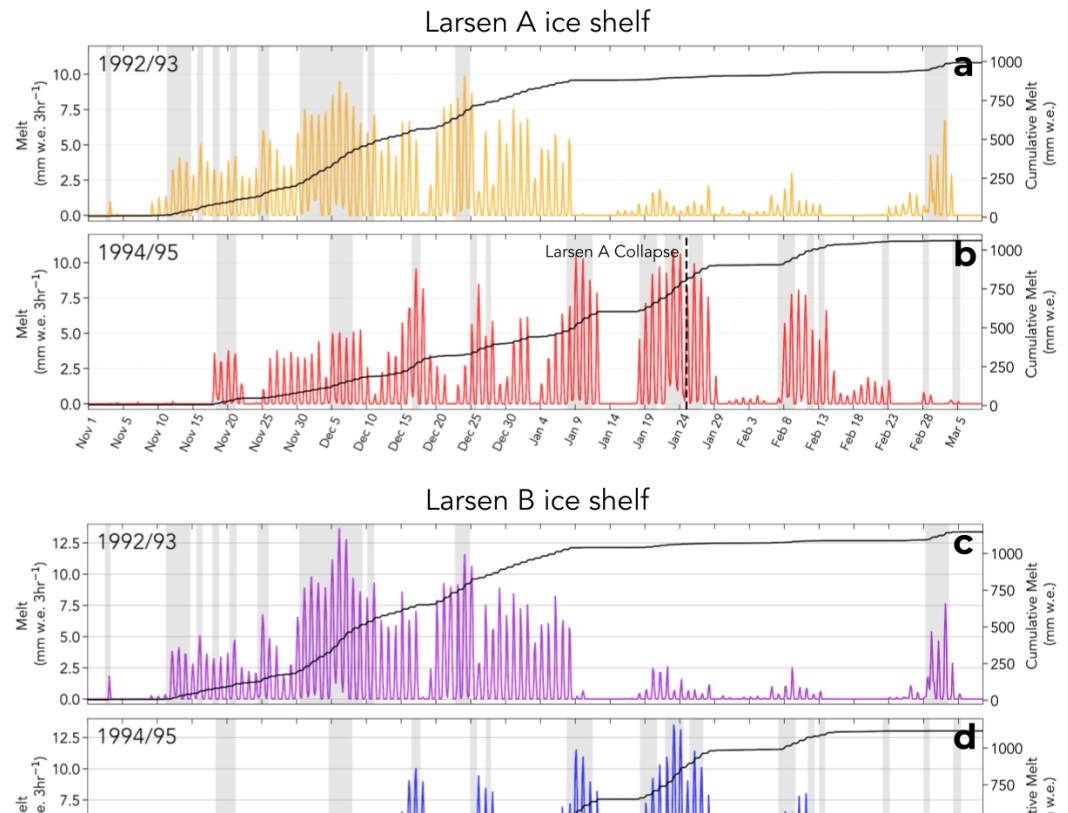

**Figure 4**. RACMO2 time series of surface melt production and cumulative melt during the Antarctic melt season averaged over the
indicated ice shelf. Grey shading indicates the presence of föhn winds. (a) 1992/1993 LAIS. (b) 1994/1995 LAIS. (c) 1992/1993 LBIS. (d)
1994/1995 LBIS. (e) 2001/2002 LBIS. *Note*: Surface melt that occurs after the collapse events indicated by the dashed vertical lines in (b)
and (e) are estimates of melt quantity if the ice shelves did not disintegrate.


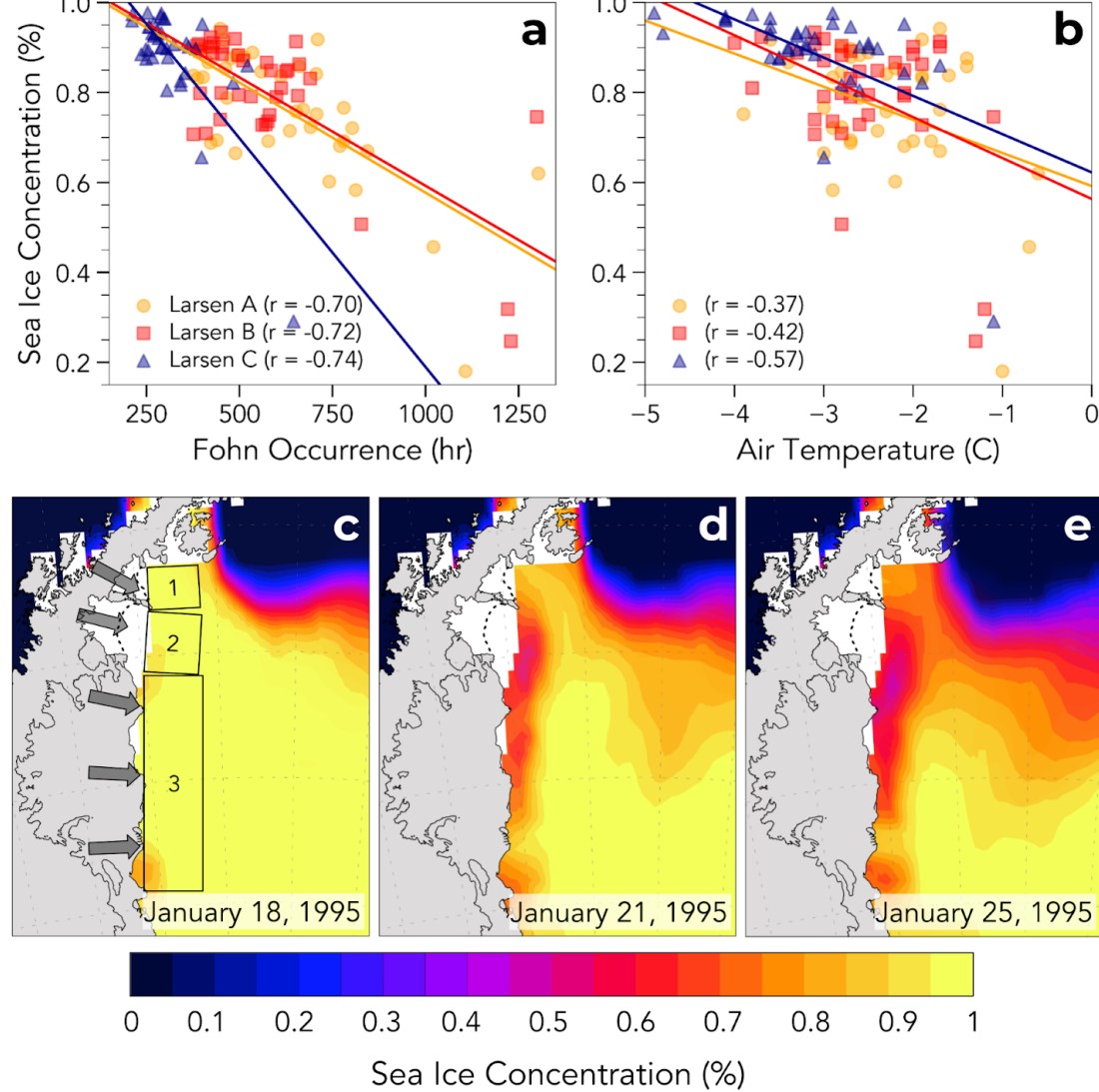

**Figure 5**. (a) Scatter plot of ERA5 summer (DJF) sea ice concentration and RACMO2 identified fohn occurrence hours on each ice shelf
from 1979-2018. (b) Scatter plot of ERA5 summer (DJF) sea ice concentration and RACMO2 mean summer air temperature on each ice
shelf from 1979-2018. ERA5 sea ice concentration at the start of a 9-day fohn melt event (c), the middle of the event (d), and on the day of
initial phase of the LAIS collapse (e). Grey arrows indicate the mean fohn wind direction and the numbered boxes indicate the sea ice
study region associated with the adjacent ice shelf for the correlation analysis (LAIS (1), LBIS (2), LCIS (3)).

## 3.3 Föhn melt and the surface liquid water budget

To better understand the role that föhn winds play in eastern AP ice shelf surface melt and stability we intercompare melt
climatologies and the surface liquid water budget of all eastern AP ice shelves (Larsen A, Larsen B, SCAR inlet, Larsen C).
A comparison of collapsed with intact ice shelves yields a clearer picture of the effects föhn winds have on ice shelf stability.
We identify whether annual surface melt production, melt rate, melt hours, and surface temperature variables from
1980-2002 are significantly different from the LBIS (Figure 6 and corresponding two-tailed t-test statistics in Table S2). We
compare to LBIS because it was centred between other ice shelves and was the most recent to collapse. Total surface melt
production on every ice shelf except LAIS differs significantly from LBIS melt (Mean annual melt over the ice shelf area;
LAIS-476 mm w.e., LBIS-479 mm w.e., SCAR-353 mm w.e., Larsen(north)-336 mm w.e., LCIS-238 mm w.e.) (Figure 6a),
which is expected when we consider the latitudinal location and mean annual air temperature (Figure 6d) (Table S2).
However, when föhn-induced melt is subtracted from total melt, the mean annual surface melt production on SCAR inlet and
Larsen C (north) are not statistically different from the LBIS (LAIS-337 mm w.e., LBIS-321 mm w.e., SCAR-286 mm w.e.,
Larsen(north)-278 mm w.e., LCIS-203 mm w.e.) (Figure 6b). In other words, with the exception of föhn-induced melt
(Figure 6c), melt production on SCAR Inlet and LCIS are statistically indistinguishable at the 95% confidence interval from
LBIS melt production. Föhn wind-induced surface melt impacted collapsed ice shelves significantly more than SCAR inlet
and the LCIS which further defines föhn melt as an important contributor to LAIS and LBIS melt budget.
Our analysis of firn density or available firn pore space identifies significant differences in ice shelves that have
collapsed (LAIS, LBIS) and those that remain intact (SCAR inlet, LCIS). The liquid-to-solid ratio (LSR) is a crude proxy for
available firn air content with extant ice shelves (SCAR inlet, LCIS) have an LSR just above 1 for the period 1980-2002 if
all surface melt is included (Figure 7a). The LSR for LAIS and LBIS is also just above 1 for this period, though only if
föhn-induced surface melt is excluded (Figure 7b). When surface melt caused by föhn wind is included, LSR exceeds 1.5
throughout extensive regions, including the ice shelf margins, of the LAIS and LBIS. Thus the collapsed ice shelves
experienced climatological LSRs significantly larger than the SCAR inlet and the LCIS, mainly due to föhn-induced melt. It
is important to note that there is evidence that the LCIS experiences regions of firn densification through melt processes,
however these regions are mostly focused close to the AP mountains, likely formed from the location of fohn jets (Hubbard
et al., 2016). This result suggests that föhn-induced melt helped precondition the LAIS and LBIS to produce extensive melt
lakes by long-term firn densification.

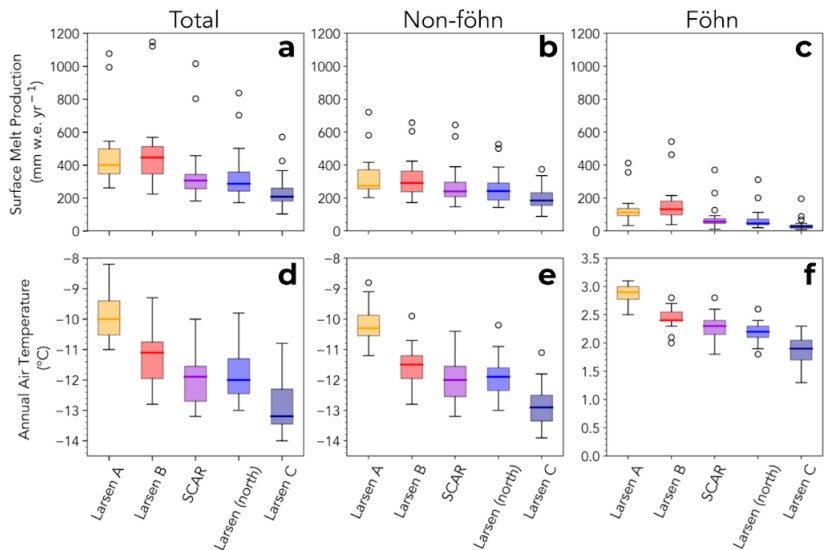

**Figure 6**. Box and whisker plots intercompare ice shelves with RACMO2-simulations from 1980-2002. Annual surface melt production (a) all melt, (b) non-föhn melt, (c) föhn-induced melt. (d) Mean annual air temperature, (e) air temperature without föhn winds, (f) air temperature during föhn winds. *Note*: the LAIS estimates are hypothetical after 1995, but are still resolved in the model simulations.

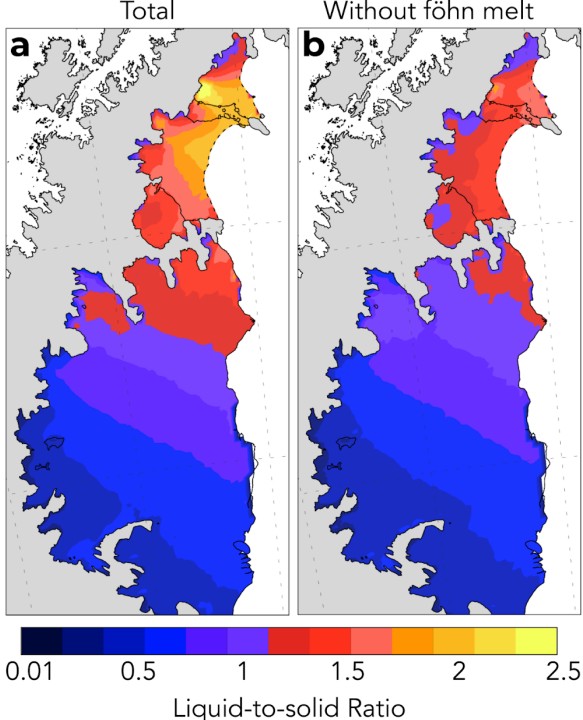

**Figure 7**. RACMO2 firn liquid-to-solid ratio or mean annual liquid water divided by mean annual frozen precipitation from 1979-2002 for
(a) total melt and (b) all liquid water except föhn-induced melt. *Note*: the LAIS estimates are hypothetical after 1995, but are still resolved
in the model simulations.
**4 Discussion**
The north/south temperature gradient present on the eastern AP ice shelves contributes to the differences in the ice shelf melt
regime (Figure 6). Warmer ice shelves can be more vulnerable to long-term thinning and retreat that accelerate disintegration
(Scambos et al., 2003; Morris and Vaughan, 2003). However, the temperature gradient alone does not explain the substantial
increase in surface melt on the LAIS and LBIS relative to more southerly ice shelves. Only with the addition of föhn-induced
surface melt (Figure 6c) do the LAIS and LBIS stand out significantly from the other eastern AP ice shelves (Figure 6a,b).
Temperature gradient however, could explain why fohn wind events cause less melt on more southern ice shelves and may
cause super melt events on collapsed ice shelves because temperature is already elevated on more northern ice shelves prior
to the effect fohn has on temperature. With that in mind, we have examined liquid water processes on the spatio-temporal
scales pertinent to AP ice shelf stability. For instance, the structural flow discontinuities or suture zones, where tributary
glaciers merge together to form an ice shelf, are mechanically weak points that impact stability (Sandhager et al., 2005;
Glasser and Scambos (2008); Glasser et al., 2009). These suture zones are further weakened through lateral shear depending
on the difference in tributary glacier flow. All ice shelves in the region are composed of numerous outflow glaciers sutured
together, and while some studies suggest this is a major contributor to ice shelf instability, only two of the ice shelves have
collapsed (Borstad et al., 2016; Glasser and Scambos, 2008). Further research suggests that marine accretion of ice on the
bottom of the ice shelves, specifically LCIS, may stabilise these suture zones, which may be why SCAR inlet has remained
intact despite major rift formation (McGrath et al., 2014; Borstad et al., 2016).

314       The timing of surface melt and melt enhanced by föhn winds within the melt season may also provide insight into

the fate of LAIS and LBIS, including why neither ice shelf collapsed in the anomalously strong 1992/93 melt season (Figure
3d). Pore space within the upper snow and firn layers buffers surface melt before lakes begin to form (Polashenski et al.,
2017). Late season melt is more likely to form surface melt lakes because meltwater from the preceding fall, winter, and
spring has partially or completely filled available pore space. On both the LAIS and LBIS, 92% of surface melt during the
1992/93 melt season occurred before January 9th when there was more pore space to buffer the anomalous surface melt than
at the onsets of their collapses in late January 1995 and early February 2002, respectively (Figure 4a, c). Melt lakes were
present on both ice shelves throughout the 1992/93 melt season, though melt production slowed dramatically after
mid-January, 1993 (Scambos et al., 2000). The high melt rates in late November and early December 1992 on the LAIS were
perhaps too early in the melt season, and after too many years of nominal melt, to form substantial melt lakes and trigger
hydrofracture that season. Nevertheless, the 1992/93 melt could have preconditioned the shelf for collapse in January 1995.
The LBIS collapse began in February 2002 after the surface melt had returned to nominal, 1980s levels for six years. How
much pore space had recovered during those six years is unknown, and an important question for future research. Satellite
images of surface melt lakes indicate 11% of the ice shelf was covered in melt lakes prior to collapse (Glasser and Scambos
(2008)). However, the preceding melt year (2000/2001) had low melt and high precipitation, which added additional snow
and water mass to the unstable ice shelf (Leeson et al., 2017).

330       Another possible reason collapse of the LAIS and LBIS did not occur in the 92/93 melt season or other years prior

to collapse was a possible misalignment of the four prerequisites for rapid collapse theorised by Massom et al., (2018). An
AVHRR image of the LAIS taken on December 8, 1992, just after a series of major föhn wind events that lead to 252 mm
w.e. of surface melt in the 8 days prior to the image (Figure 4a), show significant melt lakes across the LAIS, which make
hydrofracture cascades possible. However, in the same image, sea ice/melange are shown to be at the calving front,
protecting the front from long period ocean swells that could trigger collapse. It may have been to early in the melt season to
have substantial gaps in sea ice, the ocean temperature may have been to cold, ocean circulation could have help stabilise the
sea ice at the front, the föhn winds speed could have been to weak to push the ice away or may have been in the wrong
direction, all of which could have not allowed a proper trigger for collapse even though substantial melt ponds were present.
Even if there were years or instances that sea ice extent was low and substantial melt lakes were present, there could have
been a lack of long period ocean swells that are thought to trigger collapse.

341       Regardless of other possible contributors to ice shelf instability not considered here (e.g., basal melting),

föhn-induced surface melt and associated melt lakes, and the off-coast wind direction likely played an important role in
pushing the LAIS and LBIS past a structural tipping point. The estimated surface melt lake depth caused by the 9-day föhn
melt event on the LAIS surpassed a melt lake depth identified by modelled and satellite-derived lake depths before the
collapse of the LBIS (Banwell et al., 2013; Banwell et al., 2014). The LAIS was likely the same thickness (200m) or thinner
at the time of collapse so the estimate of critical surface lake depth for the LBIS that is applied to the LAIS may reflect an
upper limit of melt lake depth of stability for the LAIS. Melt lake depth is likely underestimated because our estimation only
accounts for melt during the 9-day melt event. Melt before this time period already exceeded an average melt year by 23%
(118 mm w.e.) so melt lakes probably already existed.
**5 Conclusions**
The converging lines of evidence in these results show that observed and inferred föhn-driven melt is present in sufficient
amounts, and at the right locations and times, to cause extensive surface melt lakes, while the off-coast föhn wind direction
pushed sea ice away from the calving front. The fact that the LAIS and LBIS collapsed catastrophically within weeks and
not through long-term thinning and retreat like other ice shelves (Prince Gustav, Wordie) suggests sudden disintegration is

anomalous and requires forcings to match vulnerabilities (Scambos et al., 2003). We conclude that föhn winds and the associated surface melt played a significant role in the collapses of the LAIS and LBIS, while extant eastern AP ice shelves are not likely to collapse from föhn-induced melt and hydrofracture in today's current climate. We have come to these conclusions with the following forms of evidence:

- First, both the LAIS and LBIS are impacted by powerful melt-inducing föhn jets that affect a large spatial portion of each ice shelf and reach the ice shelf terminus. Surface melt and melt lakes near the ice shelf terminus can lead to calving front collapse and structural instability for the remaining portion of the ice shelves (Depoorter et al., 2013; Pollard et al., 2015). SCAR inlet and the LCIS are either not directly affected by a föhn jet, are too vast to have any significant effect near the terminus, or are too far south to experience major melt events.

- Second, strong föhn winds were present prior to and during collapse for the LAIS and LBIS. A series of three föhn events on the LAIS lasted nine days total and produced over 25% of the total annual melt for the 1994/95 melt season, while föhn was present prior to and during the collapse of the LBIS which enhanced surface melt rates. Enhanced melt, filled new and existing melt lakes above the melt lake depth observed on the LBIS (1m) and modelled lake depth (5m) that could trigger large-scale hydrofracture cascades. The föhn winds on both ice shelves actively pushed/melted sea ice away from the calving front allowing long period ocean swells to trigger large-scale hydrofracture cascades on the LBIS and possibly LAIS, exacerbated by extensive surface melt that originated from the ice shelf terminus.

- Third, in the absence of föhn wind-induced melt, the surface liquid budgets of collapsed and intact ice shelves are climatically similar, which points to föhn winds as a driver of increased surface melt and extensive melt lakes on collapsed ice shelves. The additional föhn induced-melt on the LAIS and LBIS compared to intact ice shelves helped precondition the LAIS and LBIS to produce extensive melt lakes by long-term firn densification.

This research clarifies the roles of föhn-induced melt for collapsed and extant ice shelves on the eastern AP. Future analyses of these ice shelf collapse events using advanced firn density models coupled with ice-ocean-atmospheric coupled simulations may be useful to better understand the role of surface melt in ice shelf instability. Further, the AP föhn wind regime has remained stable over the past half-century (Laffin et al., 2021) which points to enhanced surface temperatures and increased liquid phase precipitation as more important contributors to the future surface liquid budget on remaining ice shelves and is an important area of future research (Bozkurt et al., 2020; Bozkurt et al., 2021). However, changes in climate drivers such as the Southern Annular Mode (SAM), which influences the north-south movement of the westerlies in the region, may alter the temperature and föhn occurrence that will likely enhance surface melt in locations farther south, and therefore make morth southern ice shelves more vulnerable (Abram et al., 2014; Zheng et al., 2013; Lim et al., 2016; ).

Nevertheless, this research highlights a new understanding behind föhn melt mechanisms for ice shelf collapse and suggests
that SCAR inlet and the LCIS may remain stable so long as surface liquid water from melt and precipitation remains within
historical bounds.


*Author contributions*. M.K.L and C.S.Z designed the study. M.V.W. and S.M. curated the model simulation output and
surface observations. M.K.L performed statistical data analysis. M.K.L. wrote the article with valuable input from all
authors.

*Competing interests*. The authors declare no conflict of interest.

*Acknowledgments*. MKL was supported by the National Science Foundation (NRT-1633631) and NASA AIST
(80NSSC17K0540). CSZ gratefully acknowledges support from the DOE BER ESM and SciDAC programs
(DE-SC0019278, LLNL-B639667, LANL-520117). JMVW acknowledges support by PROTECT and was partly funded by
the NWO (Netherlands Organisation for Scientific Research) VENI grant VI.Veni.192.083. We thank Dr. Helmut Rott for
generously providing detailed in-person observations of the LAIS months before collapse. We also thank the Institute for
Marine and Atmospheric research Utrecht (IMAU) for providing RACMO2 output. RACMO2 model data are available by
request at https://www.projects.science.uu.nl/iceclimate/models/antarctica.php, however, a subset (2001-2018) of the data are
hosted online at https://zenodo.org/record/3677642#.X-pXAFNKjUI. This work utilized the infrastructure for
high-performance and high-throughput computing, research data storage and analysis, and scientific software tool integration
built, operated, and updated by the Research Cyberinfrastructure Center (RCIC) at the University of California, Irvine (UCI).
The RCIC provides cluster-based systems, application software, and scalable storage to directly support the UCI research
community. https://rcic.uci.edu

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
