# Peer review of "The role of föhn winds in eastern Antarctic Peninsula rapid ice shelf"

_The Cryosphere, 2021_

## Referee Comment (RC1)

**Review for "*Antarctic Peninsula ice shelf collapse triggered by föhn wind-induced melt*" by Laffin et al.**

General comments

This paper has the potential to be a very interesting study about the possible influence of föhn winds on the large-scale collapse past events of the Larsen A and B ice shelves, and on potential future break-up events of Antarctic Peninsula ice shelves. However, in its current state, it is poorly written and badly structured in many places, e.g. why are föhn winds only defined and described in the Results and not in the Introduction? I give more examples in my line-by-line comments below.

Additionally, the current paper includes extremely limited references to relevant work that has already been done (particularly regarding föhn winds, but also regarding surface melt processes in general). A good example of this is the sentence in the abstract (line 13/14) which reads: "However, no studies examine the timing, magnitude, and location of surface melt processes immediately preceding these disintegrations." This statement about the Larsen A and B ice shelves is entirely incorrect as there have been many studies that have examined surface melt processes on these ice shelves, e.g. Scambos et al (2000, 2003, 2004) Glasser and Scambos (2008), Leeson et al (2017, 2020), Banwell et al (2013; 2014), Kuipers Munneke et al (2014), Lenaerts et al (2017) and Robel and Banwell (2019), to name just a few. I suggest that this sentence (and similar sentences in the Introduction) are reworded to specifically focus on the research to-date regarding effects of foehn winds on surface melt on ice shelves. Currently this paper references only a few such föhn wind studies; the following key studies about föhn wind induced ice-shelf melt are missing: Datta et al (2019), Wiesenekker et al (2018), Bozkurt et al (2018), Kirchgaessner et al (2021), and I suspect a good few others. Kirchgaessner et al (2021) is particularly relevant to the current study as it also focusses on AP ice shelves. As I am not 100% up to date with the ice-shelf melt-related föhn wind literature myself, it has been hard for me to give this paper thorough review given that the authors have not placed their study in the context of existing knowledge from other literature.

Finally, unlike the LAIS, I think I agree with the statement that the 'LBIS collapse was not directly related to the impact of föhn-induced melt', e.g. as the authors state on line 190 and in the Conclusion. However as the initial LBIS collapse on Feb 9 2002 coincided with a föhn wind event, I wonder if the authors have considered the idea that that föhn wind event may have helped produce sufficient surface meltwater such that the drainage of multiple surface lakes via hydrofracture cascades may have been triggered (i.e. 'chain reaction' lake drainage), thereby resulting in LBIS's near complete collapse a couple of weeks later (see Banwell et al 2013, Robel and Banwell, 2019). So in that sense, I am wondering what the authors think about the idea of föhn winds having been an indirect cause of LBIS's break-up?

Specific comments

Line 11: 'Add 'grounded' before 'glaciers'.

12/13: In addition to surface melting, a mention of lake drainage via hydrofracture, and/or cascades (or a chain reaction) of lake drainage events could be mentioned here.

13/14: See 'general comment' above.

16: Mention the paper's specific focus on Antarctic Peninsula shelves.

18: 'less' vulnerable compared to what?

22: 'Forensic' is the wrong word as there is no link with crime.

26 – 28: Similar to the comment I made about line 13/14 in the abstract, this sentence is entirely incorrect and does not reference prior key studies regarding both surface melt processes on ice shelves and föhn winds specifically. I suggest you add at least the references I mention above, but I will have missed some.

29 – 30: Be clear that you are using a ML method you developed in a previous study (at least that is what I am guessing), i.e. Laffin et al (2021), and reference that. Currently this sentence is vague.

30- 32: You state that your method is the 'most accurate', but you do not state what other methods/studies you are comparing it too, and nor do you state how you came to such a conclusion? Did you do some sort of intercomparison study? If so, that should be briefly explained.

33 – 41: This is interesting, as it totally contradicts the statements made in the Abstract and Introduction about there being no studies that have looked at such ice shelf melt/collapse processes! Additionally, by 'warm water intrusion', I assume you are referring to enhanced basal melting? And another good example of a study that demonstrated how sea swell caused ice shelf frontal break up is Banwell et al (2017).

43: For the 1 metre lake depth reference for LBIS, the two references given are incorrect. They should be Glasser and Scambos (2008) and Banwell et al (2014).

47: Regarding 'ice shelves into sections with aspect ratios that support unstable rollover', Burton et al (2013) would be a very appropriate reference to add.

48: Robel et al (2019) is incorrect. It should be Robel and Banwell (2019).

49 - 51: The first part of the following sentence requires references, and the second part is incorrect (for the reasons I give above in General Comments): '*Previous research acknowledges enhanced surface melt during years of collapse and the presence of föhn wind events in the region, however, no attempt to produce a timeline of total melt quantity or melt caused by föhn before and during ice shelf breakup has been undertaken*'

52/53: Poor English. Reword.

55 – 58: These questions are good; clear and precise.

59/60: 'spatial distribution' of what? Poor English.

85: It needs to be much clearer that the current study uses a föhn detection algorithm developed in a prior study (Laffin et al. 2021), and NOT in this study (at least that is my understanding from the current paper).

86 – 97: It would be interesting for the authors to compare how their algorithm compares to that used by Datta et al 2019 ('Foehn Index'; also used in Banwell et al. 2021) and perhaps other existing algorithms too. E.g., on what basis/using what evidence can the authors state that there *'method is the most accurate compared to previous work'* (without even giving reference to that previous work).

105: I think these should more accurately be described as ice shelf "areas" given that Larsen C is split into two areas. Also, I suggest listing those ice shelves/areas in this sentence.

113: you have already defined AWS elsewhere.

116 – 120: This useful definition/description about föhn winds needs to be moved into the Introduction; it does not belong here.

121: *'AP winds from the west and northwest (*föhn *influence)'* is not clear. Are you suggesting that all winds from the W and NW on the AP are föhn? (If so, that isn't clear, and I assume not all winds from that direction are föhn?

121/122: I assume this is a result from the current study, but that needs to be made clear if so.

129: *'The degree to which föhn winds impact surface melt on each ice shelf varies…'* state what timescale(s) are being considered here.

131: Figure 5 is mentioned before figures 3 and 4 have been mentioned.

140/141: I simply do not know what the authors are trying to state by the following sentence: *'However no single factor, including föhn-induced melt rate, lessens the influence of all the other factors that contributed to these collapses.'*

153/54: For the first part of this sentence, please acknowledge (and reference) other studies that have also established this fact.

168: Banwell et al (2013) did not study Larsen A.

190: Please see the final paragraph in my 'general comments' above.

211-225: It seems like some of this material (inc. equation1) should be in the Methods, not Results?

229/230: Again, discuss this statement in the context of the findings of other studies.

251: Glasser et al 2018 should be 'Glasser and Scambos (2008)', and Glasser et al (2021) is not in the reference list.

274: Satellite-derived depths of lakes are in Banwell et al (2014).

278 – 281: The authors state the following two sentences, which I disagree with: '*The large melt volume in a relatively short amount of time spatially expanded and increased melt lake formation and depth, filled crevasses, increased water pressure on the crevasse tip and walls and triggered large-scale hydrofracture cascades that led to catastrophic disintegration of the LAIS (Scambos et al., 2000; Banwell et al., 2013). The same cannot be said about the LBIS*'. The processes described in the first part of the sentence are what various studies have proposed caused the ultimate collapse of the LBIS, but I am not aware of any study have proposed the same mechanism for LAIS (Scambos et al 2000 or Banwell et al 2013 certainly did not).

290: George VI is not a good example to use here as it has very constrained, compressed ice flow.

293: '*more stable*' than what? This is vague.

294: 'than previously thought' – by who? Give references.

Figures

Figure 2: I assume the data shown in panels b) and c) are from RACMO2, but that should be clarified.

Figure 3: Again, where is the data shown in this figure derived from?

Figure 4: Again, please state the source of the data.

Figure 5: Again, state the source of the data in the caption, and specify what kind of data it is. 'data' is vague.

Figure 6: Again, state data source. And for a), should this be 'total melt'?

References (those in **bold** are not referenced in the current paper)

**Bozkurt, D., Rondanelli, R., Marin, J. C., & Garreaud, R. Foehn event triggered by an atmospheric river underlies record-setting temperature along continental Antarctica. Journal of Geophysical Research: Atmospheres, 123, 3871–3892. https://doi.org/10.1002/ 2017JD027796, 2018.**

**Banwell, A. F., Cabellero, M., Arnold, N., Glasser, N., Cath- les, L. M., and MacAyeal, D.: Supraglacial lakes on the Larsen B Ice Shelf, Antarctica, and Paakitsoq Region, Greenland: a comparative study, Ann. Glaciol., 55, 1–8, https://doi.org/10.3189/2014AoG66A049, 2014.**

**Banwell, A.F., Willis, I.C., Goodsell, B., Macdonald, G.J., Mayer, D., Powell, A. and MacAyeal, D.R. Calving and Rifting on McMurdo Ice Shelf, Antarctica. Annals of Glaciology. doi: 10.1017/aog.2017.12, 2017.**

**Banwell, A. F., Datta, R. T., Dell, R. L., Moussavi, M., Brucker, L., Picard, G., Shuman, C. A., and Stevens, L. A. The 32-year record-high surface melt in 2019/2020 on the northern George VI Ice Shelf, Antarctic Peninsula, The Cryosphere, 15, 909–925, https://doi.org/10.5194/tc-15-909-2021, 2021.**

Burton, J., L. Mac Cathles, W. Grant Wilder, The role of cooperative iceberg capsize in ice-shelf disintegration. Ann. Glaciol. 54, 84–90, 2013.

Cape, M. R., Vernet, M., Skvarca, P., Marinsek, S., Scambos, T., & Domack, E. Foehn winds link climate-driven warming to ice shelf evolution in Antarctica. Journal of Geophysical Research-Atmospheres, 120(21), 11,037–11,057. https://doi.org/10.1002/2015JD023465, 2015.

Datta, R. T., Tedesco, M., Fettweis, X., Agosta, C., Lher- mitte, S., Lenaerts, J. T. M., and Wever, N.: The effect of Foehn-induced surface melt on firn evolution over the north- east Antarctic peninsula, Geophys. Res. Lett., 46, 3822–3831, https://doi.org/10.1029/2018GL080845, 2019.

Glasser, N. F., & Scambos, T. A. A structural glaciological analysis of the 2002 Larsen B ice-shelf collapse. Journal of Glaciology, 54(184), 3–16, 2008.

Kirchgaessner, A., King, J. C., & Anderson, P. S. The impact of Föhn conditions across the Antarctic Peninsula on local meteorology based on AWS measurements. Journal of Geophysical Research: Atmospheres, 126, e2020JD033748. https://doi. org/10.1029/2020JD033748, 2021.

Kuipers Munneke, P., Ligtenberg, S. R. M., Van Den Broeke, M. R., and Vaughan, D. G.: Firn air depletion as a precur- sor of Antarctic ice-shelf collapse, J. Glaciol., 60, 205–214, https://doi.org/10.3189/2014JoG13J183, 2014.

Leeson, A. A., Van Wessem, J. M., Ligtenberg, S. R. M., Shepherd, A., Van Den Broeke, M. R., Killick, R., et al. Regional climate of the Larsen B embayment 1980–2014. Journal of Glaciology, 63(240), 683–690. https://doi.org/10.1017/jog.2017.39, 2017.

Leeson, A. A., Forster, E., Rice, A., Gourmelen, N., & van Wessem, J. M. (2020). Evolution of supraglacial lakes on the Larsen B ice shelf in the decades before it collapsed. Geophysical Research Letters, 47, e2019GL085591. https://doi.org/10.1029/2019GL085591

Lenaerts, J. T. M., Lhermitte, S., Drews, R., Ligtenberg, S. R. M., Berger, S., Helm, V., Smeets, C. J. P. P., Broeke, M. R. van den, van de Berg, W. J., van Meijgaard, E., Eijkelboom, M., Eisen, O., and Pattyn, F.: Meltwater produced by wind–albedo interaction stored in an East Antarctic ice shelf, Nat. Clim. Chang. 2017 71, 7, 58–62, https://doi.org/10.1038/nclimate3180, 2016.

Robel, A. A. and Banwell, A. F.: A speed limit on ice shelf collapse through hydrofracture, Geophys. Res. Lett., 46, 12092–12100, https://doi.org/10.1029/2019gl084397, 2019.

Scambos, T., Hulbe, C., and Fahnestock, M.: Climate-induced ice shelf disintegration in the antarctic peninsula, Antarctic Penin- sula climate variability: Historical and paleoenvironmental per- spectives, Vol. 79, 79–92, American Geophysical Union. Antarct Res. Ser., Washington, DC, 2003.

Scambos, T. A., Hulbe, C., Fahnestock, M., and Bohlander, J.: The link between climate warming and break-up of ice shelves in the Antarctic Peninsula, J. Glaciol., 46, 516–530, 2000.

Scambos, T. A., Bohlander, J. A., Shuman, C. U., and Skvarca, P.: Glacier acceleration and thinning after ice shelf collapse in the Larsen B embayment, Antarctica, Geophys. Res. Lett., 31, L18402, https://doi.org/10.1029/2004GL020670, 2004.

Wiesenekker, J., Kuipers Munneke, P., van den Broeke, M., & Smeets, C. A multidecadal analysis of Föhn winds over Larsen C ice shelf from a combination of observations and modeling. Atmosphere, 9(5), 172. https://doi.org/10.3390/atmos9050172, 2018

---

## Referee Comment (RC2)

General comments

The authors here use a foehn wind detection algorithm to quantify surface melt magnitude and timing to claim that a foehn wind event pushed the Larsen A ice shelf past a critical stability threshold ultimately leading to its collapse in 1995. Meanwhile, since the Larsen B ice shelf experienced weaker foehn-related melt prior to its collapse in 2002, foehn winds likely preconditioned the ice shelf for collapse. While the foehn detection algorithm provides new, detailed insights into foehn jet positions and foehn wind related melt magnitude, the conclusions regarding ice shelf stability and collapse are underdeveloped and unsupported by the results. I give line-by-line results later, but globally I believe this manuscript suffers from two key elements.

The first is the lack of references to already published work that describe ice-shelf stability processes. Other times, relevant papers are cited, but their conclusions are misrepresented or not mentioned in the text. I give more detailed examples below, but one glaring example is the exemption of discussion from Massom et al., 2018 which discusses of ice shelf collapse triggered by sea ice loss and ocean swells. This paper is cited in the manuscript, but the results about how sea-ice loss and exposure to ocean swells triggered the collapse of the Larsen A and B are never discussed in this manuscript. The authors should consider these processes before claiming foehn winds triggered the collapse of the Larsen A.

The authors also cite Scambos et al., 2000, but appear to miss some important observations from that study. The authors in that study cite a storm as the trigger for the final disintegration of the Larsen A, but this fact does not appear in this paper's discussion of the Larsen A collapse. Is the foehn wind event here related to that storm mentioned in Scambos et al., 2000? Also, Scambos et al., 2000 mentions the Larsen A suffered major retreats in 1987 and 1989 which did not appear to be major foehn event years according to this study but did precondition the ice shelf for collapse which contradicts one of the authors' conclusions.

The second issue is claiming one particular process could trigger an ice shelf collapse is a very high bar to pass given the multitude of other processes known to cause ice shelf instability. This manuscript would be much easier to accept as a reader if the authors move their focus away from the supposed novelty of their research and towards the value this research brings to an already rich field of research relating foehn-wind and ice shelf stability. In fact, there are moments when the authors claim to demonstrate a result for the first time when this result was already discussed in previous literature (see comment on line 51). The manuscript would be much easier to digest if the authors moved away from the claim that foehn winds triggered ice shelf collapse and instead focused on highlighting foehn winds as one of many processes that lead to ice shelf instability and the timing of the foehn winds may have played a supporting role in the collapse of the Larsen A.

Line 13: Saying that there are no studies examining surface melt prior to disintegrations is incorrect. You should revisit the Van Den Brooke, 2005 GRL paper that you cited that explicitly studies surface melt on the Larsen B prior to its collapse.

Line 17-19: This claim is based on a premise that foehn wind and surface temperatures remain within historical bounds. The Antarctic Peninsula already experiences large temperature variability and is projected to become warmer which would actually make the extant ice shelves more vulnerable to foehn winds in the future (Siegert et al., 2019; Chyhareva et al., 2019).

Line 25-27: The claim of novelty seems unwarranted here. Plenty of studies already cited in this manuscript plus some others discuss fohn-related melt mechanisms on the Larsen B ice shelf (see Datta et al. 2019). Plus, Van den Brooke et al., 2005 claims surface melt accelerated the rate of ice shelf retreat, but did not claim it was a leading contributor to the final collapse

Line 33-41: I don't understand why the manuscripts claims surface melt as the lead cause of the ice shelf final collapse in the previous paragraph and then point out all the other well-documented processes that also affect ice shelf final collapse.

Line 30: This is a strange claim to make in the introduction. If this claim is valid, then it should first be proven in the results and then mentioned in the conclusion

Line 51: This is repeating a claim from the first paragraph that incorrectly states no previous research has been done on foehn-related melt around ice shelf collapses. This study may certainly give further detail on the intensity and spatial distribution of the foehn wind, but certainly is not the first.

Line 54: The temperature trends on the Antarctic Peninsula are a bit more complicated than this. Bozkurt et al., 2020, Carrasco et al., 2021, and Turner et al., 2016 paint a different picture where temperature trends are periodic and dependent on the location along the AP

Line 57: Questions 1 and 3 are very important and reasonable questions to address in this manuscript. Question 2 is much harder to answer with certainty without considering all the other processes (atmospheric and non-atmospheric) that could affect ice-shelf stability.

Line 87: What height is the air temperature measured at?

Line 95: It is stated again that is foehn detection method is the most accurate compared to previous work without explaining what this previous work is or why it is the most accurate. I also believe this is not the first foehn detection algorithm to incorporate station observations and model output (see Turton et al., 2018). The authors should include some information comparing the foehn detection of their algorithm against other foehn detection algorithms even if that data is presented in Laffin et al., 2021.

Line 108: Perhaps explain which variables you used to make the two-tailed t-test statistic. "Mean of both ice shelves" is vague

Line 115-119: This seems like background information on the physics of foehn winds that would be better suited in the introduction section.

Line 131: This might be a personal preference, but you should change your figure numbers/order if you are referring to figure 5 before figure 3.

Line 132: You should present some results on foehn frequency before presenting the foehn-related melt percentage. This would help put these melt-percentages in a better context.

Line 137: If the SCAR inlet is not impacted by a foehn jet, where is the foehn wind influence coming from?

Line 139 – 142: You are contradicting yourself or at least unclear in these two sentences. First you claim that the disparity in foehn-related melt percentages among the ice shelves implicates the foehn as a contributor to the LAIS and LBIS collapse. This is a very strong assertion. It explains differences in melt rates on the ice shelves but saying this contributes to their collapse is a stretch. Then the next sentence is confusing and muddles your message about whether foehn is important or not to collapse. Probably easier to say that your results indicate foehn is one of many processes that weakened the LAIS and LBIS.

Line 149-152: If extensive foehn wind jets help explain why the LAIS and LBIB collapsed, then why have they not caused the collapse of the LCIS? Is there research showing that having melting at the terminus is essential for an ice shelf collapse?

Line 153-154: Previous literature already shows that foehn winds have a major impact on ice shelf surface melt and the framing of this sentence makes your results sound novel when in fact it would be more accurate to say that your results back up and enhance preexisting knowledge while citing these sources.

Line 181: It's a bit confusing to see the authors use satellite imagery from the 1992/1993 melt season as an analogue to the 1994/1995 melt season, but then later argue that despite the two seasons had similar amounts of foehn-related melt, the reason the Larsen A collapsed in 1995 and not in 1994/1995 was the timing of the surface melt. This argument needs more analysis of the background state of the Larsen A in 1992/1993 versus 1994/1995 to explain more clearly what was so special in 1994/1995.

Line 204-205: The total surface melt results are interesting, but would considering the size of the ice shelves change the perception of importance in regard to ice shelf destabilization? For instance, the Larsen C is much larger than the SCAR inlet ice shelf so total melt amounts would be difficult to compare. Melt per area would be a better metric.

Line 212-214: The statement about the future resilience of the other ice shelves is problematic as it ignores potential future changes in foehn wind patterns. Especially since I believe your foehn wind detection algorithm only detects foehn winds when the temperature is above 0°C. There could be foehn events that currently do not push the temperature above this threshold which are not considered by your algorithm. But theoretically, if air temperatures rises along the Larsen C, then your algorithm would start detecting more foehn wind events.

Line 227-228: The liquid-to-solid ratio (LSR) analysis here includes foehn-related melt and non-foehn related melt. As mentioned earlier, it would be helpful to know the foehn wind frequency according to your detection algorithm in order to judge the significance of this result.

Line 244-245: There are likely many other differences between the Larsen A and B and the other ice shelves beyond foehn wind patterns. At the very least, sea-ice coverage and ocean forcings are different (see Massom et al., 2018). As I am not a glaciology expert, I cannot say for certain what the differences are structurally between these ice shelves, but it probably is wise to cite some papers regarding ice dynamics to verify this statement.

Line 254: One thing missing about this discussion on the timing of the ice shelf collapses is if ice shelves have existed for thousands of years and foehn winds are a quasi-permeant feature on the Larsen ice shelves, why did foehn winds only trigger the Larsen A collapse relatively recently?

Line 270: I feel like you cannot conclude foehn-related surface melt triggered the Larsen A collapse without taking into consideration factors like basal melting.

Line 282-283: How are you certain that a combination of factors also did not trigger the final disintegration of the Larsen A? In fact, in Massom et al., 2018, it was observed that sea-ice loss allowed ocean swells to apply a strain along the ice-shelf front which is cited as a possible trigger of the Larsen A collapse. This needs to be considered and discussed in this manuscript.

Line 289-290: This sentence disregards the gradual retreat of the ice shelves like the major retreats the Larsen A experienced in 1987 and 1989 mentioned in Scambos et al., 2000.

Line 292-293: You cannot come to this conclusion if your foehn detection algorithm only detects foehn when the temperature is above 0°C which will likely occur more often over the Larsen C according to future climate projections (Siegert et al., 2019) (Chyhareva et al., 2019).

**References:**

Bozkurt, D., Bromwich, D. H., Carrasco, J., Hines, K. M., Maureira, J. C., and Rondanelli, R.: Recent Near-surface Temperature Trends in the Antarctic Peninsula from Observed, Reanalysis and Regional Climate Model Data, Adv. Atmos. Sci., 37, 477–493, https://doi.org/10.1007/s00376-020-9183-x, 2020.

Carrasco, J. F., Bozkurt, D., and Cordero, R. R.: A review of the observed air temperature in the Antarctic Peninsula. Did the warming trend come back after the early 21st hiatus?, 100653, https://doi.org/10.1016/j.polar.2021.100653, 2021.

Chyhareva, A., Krakovska, S., and Pishniak, D.: Climate projections over the Antarctic Peninsula region to the end of the 21st century. Part 1: cold temperature indices, UAJ, 62–74, https://doi.org/10.33275/1727-7485.1(18).2019.131, 2019.

Datta, R. T., Tedesco, M., Fettweis, X., Agosta, C., Lhermitte, S., Lenaerts, J. T. M., and Wever, N.: The Effect of Foehn-Induced Surface Melt on Firn Evolution Over the Northeast Antarctic Peninsula, Geophysical Research Letters, 46, 3822–3831, https://doi.org/10.1029/2018GL080845, 2019.

Massom, R. A., Scambos, T. A., Bennetts, L. G., Reid, P., Squire, V. A., and Stammerjohn, S. E.: Antarctic ice shelf disintegration triggered by sea ice loss and ocean swell, Nature, 558, 383–389, https://doi.org/10.1038/s41586-018-0212-1, 2018.

Turner, J., Lu, H., White, I., King, J. C., Phillips, T., Hosking, J. S., Bracegirdle, T. J., Marshall, G. J., Mulvaney, R., and Deb, P.: Absence of 21st century warming on Antarctic Peninsula consistent with natural variability, Nature, 535, 411–415, https://doi.org/10.1038/nature18645, 2016.

Turton, J. V., Kirchgaessner, A., Ross, A. N., and King, J. C.: The spatial distribution and temporal variability of föhn winds over the Larsen C ice shelf, Antarctica, Q.J.R. Meteorol. Soc., 144, 1169–1178, https://doi.org/10.1002/qj.3284, 2018.

Scambos, T. A., Hulbe, C., Fahnestock, M., and Bohlander, J.: The link between climate warming and break-up of ice shelves in the Antarctic Peninsula, J. Glaciol., 46, 516–530, https://doi.org/10.3189/172756500781833043, 2000.

Siegert, M., Atkinson, A., Banwell, A., Brandon, M., Convey, P., Davies, B., Downie, R., Edwards, T., Hubbard, B., Marshall, G., Rogelj, J., Rumble, J., Stroeve, J., and Vaughan, D.: The Antarctic Peninsula Under a 1.5°C Global Warming Scenario, Front. Environ. Sci., 7, 102, https://doi.org/10.3389/fenvs.2019.00102, 2019.

van den Broeke, M.: Strong surface melting preceded collapse of Antarctic Peninsula ice shelf, Geophys. Res. Lett., 32, L12815, https://doi.org/10.1029/2005GL023247, 2005.

---

## Community Comment (CC1)

**Short Comment** on "Antarctic Peninsula ice shelf collapse triggered by föhn wind-induced melt" by M. K. Laffin et al., submitted to *The Cryosphere Discussions*, 25 Oct. 2021.

*Commenter:* Helmut Rott

The authors explored mechanisms triggering the rapid collapse events of the Larsen A and B ice shelves, using a regional climate model and Machine Learning analysis in order to investigate the influence of föhn winds and associated melt on the surface liquid water budget. They conclude that increased surface melt due to föhn supplied water to melt lakes, inducing the crossing of a critical stability of water depth that triggered the rapid Larsen A collapse. The authors claim a lack of high resolution satellite imagery during the collapse and deduce estimates on melt lake surface area from an AVHRR image (1 km spatial resolution) of 8 December 1992. In fact, high resolution synthetic aperture radar (SAR) images (ca. 25 m resolution) of the ESA ERS-1 satellite were acquired during the disintegration event, on 25, 28 and 30 January and 2. February 1995. Some of the ERS-1 SAR images acquired over Larsen Ice shelf are shown in Rott et al., 1998 (paper cited by Laffin et al.). Furthermore, Rott et al., 1996 (not cited), show ERS SAR images acquired during the event and present a report on the state of Larsen A Ice Shelf two months before the collapse, built on field observations.

The ERS SAR images, as well as the report on the field observations in October and November 1994 disprove the hypothesis of Laffin et al. that the Larsen A collapse in January 1995 was triggered in the short-term by hydrofracture processes. According to the ERS image of 25 Jan. 1995, close to the start of the main disintegration event, the extent of surface lakes on Larsen A Ice Shelf amounted to about 1% of the total area (Fig. C1 below). In Oct./Nov. 1994 the ice shelf was already heavily fractured. Cold temperatures and an extended pre-frontal cover of fast ice kept the ice shelf from breaking apart. Details are reported in Rott et al., 1996, e.g. referring to an ice wedge protruding from the level ice shelf several km inland of the front (Fig. C2), on cracks along the border between the Larsen A and Seal Nunataks ice shelves, and rifts along the coastline line to the peninsula.

Regarding the temporal sequence of the collapse event, the ice shelf section downstream of Dinsmoor-Bombardier-Edgeworth (DBE) glaciers retreated to the grounding line faster than downstream of Drygalski Glacier along which Laffin et al. show the location of the LA föhn jet. The pre-collapse crack density was highest on the DBE ice shelf section. Another striking incident was the rapid off-coast drift of detached icebergs and growlers, gaining 40 km in distance between 28 and 30 January 1995, an indication for oceanic mechanic forcing as main factor for the rapidity of disintegration.

This comment is not a review of the paper. Nevertheless I want to address some further issues:

*Melt pattern on Larsen A Ice Shelf*: The model simulations (Figs. 3 and 6) indicate reduced melt in the northern section of the ice shelf (downstream of DBE glaciers). ERS SAR images, acquired in years preceding the collapse, as well as the 25 Jan. 1994 image (Fig. C1), do not show any significant difference in melt intensity between different ice shelf sections.

*Position of föhn jets*: The uniqueness of the föhn jets on Larsen A (Drygalski Glacier) and on Larsen B (Hektoria – Green glaciers) needs to be reconsidered. ERS SAR images during the Larsen A event show high reflectivity of the ocean surface and rapid of-coast displacement of ice downstream of DBE glaciers, indications for strong off-shore winds. During the Larsen B disintegration event rapid off-coast drift of icebergs was observed also downstream of Crane and Jorum glaciers (Rack et al., 2004).

*Line 289:* "... the LAIS and LBIS collapsed catastrophically within weeks and not through long-term thinning and retreat like other ice shelves..." The gradual retreat of the Larsen A front between Seal

Nunataks over 20 years up to the collapse is documented by means of satellite images starting in 1975 (Skvarca, 1993; Rott et al., 1996).

*References:*

Rack, W. and H. Rott, H.: Pattern of retreat and disintegration of the Larsen B ice shelf, Antarctic Peninsula, Ann. Glaciol., 39, 505 – 510, 2004.
Rott H., Skvarca, P., and Nagler, T.: Rapid collapse of northern Larsen Ice Shelf, Antarctica, Science, Vol. 271, Issue 5250, 788-792, 1996.
Skvarca, P.: Fast recession of the northern Larsen Ice Shelf monitored by space images, Ann. Glaciol., 17, 317-321, 1993.

[Figure]

*Fig. C1*. Section of ERS-1 SAR image, covering Larsen A Ice Shelf, 25 January 1995. CW – Cape Worsley, D – Drygalski Glacier, DBE – Dinsmoor-Bombardier-Edgeworth glaciers, S - Sobral Peninsula.

[Figure]

*Fig. C2*. Ice wedge on Larsen A Ice Shelf, located several km inland of the front (left). Cracks near the ice front (right). Photos H. Rott, 24 Oct. 1994.

---

## Author Comment (AC1)

Thank you for your comments and suggestions. We believe this manuscript will improve significantly with your suggestions and we sincerely appreciate your valuable contributions. We have addressed your comments below marked with [Author Response].

**Review for "*Antarctic Peninsula ice shelf collapse triggered by fohn wind -induced melt*" by Laffin et al.**

General comments

This paper has the potential to be a very interesting study about the possible influence of föhn winds on the large-scale collapse past events of the Larsen A and B ice shelves, and on potential future break-up events of Antarctic Peninsula ice shelves. However, in its current state, it is poorly written and badly structured in many places, e.g. why are föhn winds only defined and described in the Results and not in the Introduction? I give more examples in my line-by-line comments below.

[Author Response] - We agree that the structure of the manuscript can be improved, especially with an overview of föhn winds in the introduction section. This manuscript was originally submitted to a short form journal which is why it was structured differently then a typical Cryosphere article. We overlooked this fact when we re-wrote the manuscript for The Cryosphere and will make changes to the manuscript that are more in line with The Cryosphere structure.

Additionally, the current paper includes extremely limited references to relevant work that has already been done (particularly regarding föhn winds, but also regarding surface melt processes in general). A good example of this is the sentence in the abstract (line 13/14) which reads: "However, no studies examine the timing, magnitude, and location of surface melt processes immediately preceding these disintegrations." This statement about the Larsen A and B ice shelves is entirely incorrect as there have been many studies that have examined surface melt processes on these ice shelves, e.g. Scambos et al (2000, 2003, 2004) Glasser and Scambos (2008), Leeson et al (2017, 2020), Banwell et al (2013; 2014), Kuipers Munneke et al (2014), Lenaerts et al (2017) and Robel and Banwell (2019), to name just a few. I suggest that this sentence (and similar sentences in the Introduction) are reworded to specifically focus on the research to-date regarding effects of foehn winds on surface melt on ice shelves. Currently this paper references only a few such föhn wind studies; the following key studies about föhn wind induced ice-shelf melt are missing: Datta et al (2019), Wiesenekker et al (2018), Bozkurt et al (2018), Kirchgaessner et al (2021), and I suspect a good few others. Kirchgaessner et al (2021) is particularly relevant to the current study as it also focuses on AP ice shelves. As I am not 100% up to date with the ice-shelf melt-related föhn wind literature myself, it has been hard for me to give this paper thorough review given that the authors have not placed their study in the context of existing knowledge from other literature.

[Author Response] - We agree this manuscript is limited in it's references and in particular articles about föhn winds and föhn-induced surface melt. As with the comment above this manuscript was originally submitted to a short form journal, which limited the number of references. We felt, at the time we submitted this manuscript to The Cryosphere, the amount of references and background regarding föhn winds was sufficient. However, after your valuable comments we plan to remedy the lack of background by changing the manuscript, specifically the introduction section to provide a

clear overview of the current research to date on föhn winds and föhn-induced melt in the region. Also, in regard to your comment about Line 13/14:, *"However, no studies examine the timing, magnitude, and location of surface melt processes immediately preceding these disintegrations."*, and others like it, the passages were meant to show that little research was done on time scales shorter than annual or seasonal, however, we see that the way these comments are written make it seem like there is no research on föhn winds and surface melt. We will change all passages in the manuscript to better frame this study among the rich array of studies on föhn winds in the region.

Finally, unlike the LAIS, I think I agree with the statement that the 'LBIS collapse was not directly related to the impact of föhn-induced melt', e.g. as the authors state on line 190 and in the Conclusion. However as the initial LBIS collapse on Feb 9 2002 coincided with a föhn wind event, I wonder if the authors have considered the idea that that föhn wind event may have helped produce sufficient surface meltwater such that the drainage of multiple surface lakes via hydrofracture cascades may have been triggered (i.e. 'chain reaction' lake drainage), thereby resulting in LBIS's near complete collapse a couple of weeks later (see Banwell et al 2013, Robel and Banwell, 2019). So in that sense, I am wondering what the authors think about the idea of föhn winds having been an indirect cause of LBIS's break-up?

[Author Response] - This is a very interesting question that inspired us to change the manuscript. After reading Massom et al., 2018, which produced a useful conceptual framework for rapid ice shelf collapse and identifies large period ocean swells as the trigger mechanism for the collapse of the Larsen A and B ice shelves, we decided to alter our interpretation of our findings. Fohn winds were present at the time of collapse for both ice shelves which produced enhanced surface melt rates that caused extensive melt ponds over each ice shelf. Additionally, the direction of fohn winds (from the west/northwest direction) pushed/melted sea ice and fast ice away from the calving front of both ice shelves which allowed large ocean waves to trigger collapse, which was also discussed in Banwell et al (2017). We will change the manuscript to show that without the extensive melt ponds and lakes enhanced by fohn winds, and the wind direction that pushed protective sea ice away from the calving front, large-scale hydrofracture cascaded and subsequent collapse would not have taken place. We will also change the title of the manuscript to not suggest fohn winds triggered collapse, but instead played a supporting role in the rapid collapse of LA and LB ice shelves.

Specific comments

Line 11: 'Add 'grounded' before 'glaciers'.
[Author Response] - This will be changed to clarify grounded glaciers.

12/13: In addition to surface melting, a mention of lake drainage via hydrofracture, and/or cascades (or a chain reaction) of lake drainage events could be mentioned here.
[Author Response] - We will add in lake drainage via hydrofracture to the abstract. It was already discussed in the manuscript but will be helpful for clarification to add it into the abstract.

13/14: See 'general comment' above.
[Author Response] - We will add in lake drainage via hydrofracture to the abstract. It was already

discussed in the manuscript but will be helpful for clarification to add it into the abstract.

16: Mention the paper's specific focus on Antarctic Peninsula shelves.
[Author Response] - **We will clarify the region of study.**

18: 'less' vulnerable compared to what?
[Author Response] - **We agree this is not a useful comparison so we will clarify our remarks and compare collapsed ice shelves and extant ice shelves.**

22: 'Forensic' is the wrong word as there is no link with crime.
[Author Response] - **We meant to say that examination of past events is useful so we will take out this word and replace it for clarification.**

26 – 28: Similar to the comment I made about line 13/14 in the abstract, this sentence is entirely incorrect and does not reference prior key studies regarding both surface melt processes on ice shelves and föhn winds specifically. I suggest you add at least the references I mention above, but I will have missed some.
[Author Response] - **We will completely re-write the introduction to include valuable background and references as well as frame our findings in the context of other studies.**

29 – 30: Be clear that you are using a ML method you developed in a previous study (at least that is what I am guessing), i.e. Laffin et al (2021), and reference that. Currently this sentence is vague.
[Author Response] - **Yes, this method was developed in Laffin et al., 2021. We will make this more clear in the updated manuscript.**

30- 32: You state that your method is the 'most accurate', but you do not state what other methods/studies you are comparing it too, and nor do you state how you came to such a conclusion? Did you do some sort of intercomparison study? If so, that should be briefly explained.
[Author Response] - **We did complete an intercomparison sensitivity study detailed in Laffin et al., 2021, comparing other identification methods. We will make sure to discuss this study in this manuscript as well as provide the summary statistics from that study in the supplement.**

33 – 41: This is interesting, as it totally contradicts the statements made in the Abstract and Introduction about there being no studies that have looked at such ice shelf melt/collapse processes! Additionally, by 'warm water intrusion', I assume you are referring to enhanced basal melting? And another good example of a study that demonstrated how sea swell caused ice shelf frontal break up is Banwell et al (2017).
[Author Response] - **We do see the contradictions in this statement and those made in the abstract and throughout the manuscript. We will do a more thorough background summary in the introduction and fix these contradictions. We will also clarify our mention of "warm water intrusion" to basil melt as well as reference Banwell et al., (2017) in regard to ocean swell stress on the calving front.**

43: For the 1 meter lake depth reference for LBIS, the two references given are incorrect. They

should be Glasser and Scambos (2008) and Banwell et al (2014).
[Author Response] - **We will make sure to fix this embarrassing oversight.**

47: Regarding 'ice shelves into sections with aspect ratios that support unstable rollover', Burton et al (2013) would be a very appropriate reference to add.
[Author Response] - **We will add this reference and appreciate the suggestion.**

48: Robel et al (2019) is incorrect. It should be Robel and Banwell (2019).
[Author Response] - **We will fix this reference error.**

49 - 51: The first part of the following sentence requires references, and the second part is incorrect (for the reasons I give above in General Comments): '*Previous research acknowledges enhanced surface melt during years of collapse and the presence of föhn wind events in the region, however, no attempt to produce a timeline of total melt quantity or melt caused by föhn before and during ice shelf breakup has been undertaken*'
[Author Response] - **We will add references to the beginning of this sentence as well as clarify and change the second part of the sentence. The change in the introduction to include more background on foehn winds and ice shelf dynamics will likely make this sentence change completely.**

52/53: Poor English. Reword.
[Author Response] - **We will clarify this sentence.**

55 – 58: These questions are good; clear and precise.
[Author Response] - **Thanks!**

59/60: 'spatial distribution' of what? Poor English.
[Author Response] - **We meant to say the distribution of foehn-induced surface melt. This sentence will be changed to reflect this change and clarify our meaning.**

85: It needs to be much clearer that the current study uses a föhn detection algorithm developed in a prior study (Laffin et al. 2021), and NOT in this study (at least that is my understanding from the current paper).
[Author Response] - **We will change this sentence to make it clearer that this identification method was developed previously in Laffin et al., 2021.**

86 – 97: It would be interesting for the authors to compare how their algorithm compares to that used by Datta et al 2019 ('Foehn Index'; also used in Banwell et al. 2021) and perhaps other existing algorithms too. E.g., on what basis/using what evidence can the authors state that there '*method is the most accurate compared to previous work*' (without even giving reference to that previous work).
[Author Response] - **We did complete an intercomparison sensitivity study detailed in Laffin et al., 2021, comparing Datta et al 2019 and other identification methods. We will make sure to discuss this study in this manuscript as well as provide the summary statistics from that study in the supplement.**

105: I think these should more accurately be described as ice shelf "areas" given that Larsen C is split into two areas. Also, I suggest listing those ice shelves/areas in this sentence.
[Author Response] - We agree that this is not clear and will change the sentence to say ice shelf areas, as well as name those areas in reference to Figure 1.

113: you have already defined AWS elsewhere.
[Author Response] - Noted, we will adjust the manuscript.

116 – 120: This useful definition/description about föhn winds needs to be moved into the Introduction; it does not belong here.
[Author Response] - This is a great point. We will provide an in depth definition and thorough reference background in the introduction.

121: '*AP winds from the west and northwest (*föhn *influence)'* is not clear. Are you suggesting that all winds from the W and NW on the AP are föhn? (If so, that isn't clear, and I assume not all winds from that direction are föhn?
[Author Response] - In this region, because of the location of the Antarctic Peninsula range, most winds will have some fohn influence. We will expand more on this in the manuscript and include other research articles as well.

121/122: I assume this is a result from the current study, but that needs to be made clear if so.
[Author Response] - Yes, this is a result from this study. We included this information in Figure 2, however, we will make it more clear with specific percentages from our findings to compliment the figure.

129: '*The degree to which föhn winds impact surface melt on each ice shelf varies…*' state what timescale(s) are being considered here.
[Author Response] - For this sentence we meant to convey the difference in fohn melt from ice shelf to ice shelf and under the influence of fohn jets on single ice shelves. We will clarify this sentence to reflect our sentiment.

131: Figure 5 is mentioned before figures 3 and 4 have been mentioned.
[Author Response] - We will be re-working the manuscript and will alter the mention of Figure 5 to after the other figures.

140/141: I simply do not know what the authors are trying to state by the following sentence: '*However no single factor, including föhn-induced melt rate, lessens the influence of all the other factors that contributed to these collapses.*'
[Author Response] - We agree this sentence is confusing and will be removed from the manuscript.

153/54: For the first part of this sentence, please acknowledge (and reference) other studies that have also established this fact.
[Author Response] - Yes, there are other studies who have established this fact which we will reference.

168: Banwell et al (2013) did not study Larsen A.
[Author Response] - **We will correct this oversight.**

190: Please see the final paragraph in my 'general comments' above.
[Author Response] - **See our response above.**

211-225: It seems like some of this material (inc. equation1) should be in the Methods, not Results?
[Author Response] - **Yes, since the manuscript was originally submitted to a short form journal it was best placed in this section. We agree it is now better suited in the methods section and will be adjusted.**

229/230: Again, discuss this statement in the context of the findings of other studies.
[Author Response] - **As we mentioned above, we plan to alter the story of the manuscript to better include this work among previous research.**

251: Glasser et al 2018 should be 'Glasser and Scambos (2008)', and Glasser et al (2021) is not in the reference list.
[Author Response] - **We will make sure to fix this embarrassing oversight.**

274: Satellite-derived depths of lakes are in Banwell et al (2014).
[Author Response] - **We will make sure to fix this embarrassing oversight.**

278 – 281: The authors state the following two sentences, which I disagree with: '*The large melt volume in a relatively short amount of time spatially expanded and increased melt lake formation and depth, filled crevasses, increased water pressure on the crevasse tip and walls and triggered large-scale hydrofracture cascades that led to catastrophic disintegration of the LAIS (Scambos et al., 2000; Banwell et al., 2013). The same cannot be said about the LBIS*. The processes described in the first part of the sentence are what various studies have proposed caused the ultimate collapse of the LBIS, but I am not aware of any study have proposed the same mechanism for LAIS (Scambos et al 2000 or Banwell et al 2013 certainly did not).
[Author Response] - **Thank you for this comment. This is one of the reasons we have decided to shift the focus of the manuscript story to put fohn winds and associated melt in a support role for collapse and not the trigger. This change is discussed in more detail above.**

290: George VI is not a good example to use here as it has very constrained, compressed ice flow.
[Author Response] - **We agree and will alter this sentence.**

293: '*more stable*' than what? This is vague.
[Author Response] - **We agree this is vague. We meant to compare ice shelves that have collapsed and extant ice shelves. We will clarify this in the manuscript.**

294: 'than previously thought' – by who? Give references.
[Author Response] - **We agree this is vague. We meant to compare ice shelves that have collapsed and extant ice shelves. We will clarify this in the manuscript.**

Figures

Figure 2: I assume the data shown in panels b) and c) are from RACMO2, but that should be clarified.
[Author Response] - **Yes, the data shown is from RACMO2, we will make this more clear in the figure captions.**

Figure 3: Again, where is the data shown in this figure derived from?
[Author Response] - **Yes, the data shown is from RACMO2, we will make this more clear in the figure captions.**

Figure 4: Again, please state the source of the data.
[Author Response] - **Yes, the data shown is from RACMO2, we will make this more clear in the figure captions.**

Figure 5: Again, state the source of the data in the caption, and specify what kind of data it is. 'data' is vague.
[Author Response] - **Yes, the data shown is from RACMO2, we will make this more clear in the figure captions.**

Figure 6: Again, state data source. And for a), should this be 'total melt'?
[Author Response] - **Yes, the data shown is from RACMO2, we will make this more clear in the figure captions.**

References (those in **bold** are not referenced in the current paper)

**Bozkurt, D., Rondanelli, R., Marin, J. C., & Garreaud, R. Foehn event triggered by an  atmospheric river underlies record-setting temperature along continental Antarctica. Journal of  Geophysical Research: Atmospheres, 123, 3871–3892. https://doi.org/10.1002/ 2017JD027796,  2018.**

**Banwell, A. F., Cabellero, M., Arnold, N., Glasser, N., Cath- les, L. M., and MacAyeal, D.: Supraglacial lakes on the Larsen B Ice Shelf, Antarctica, and Paakitsoq Region, Greenland: a comparative study, Ann. Glaciol., 55, 1–8, https://doi.org/10.3189/2014AoG66A049, 2014.**

**Banwell, A.F., Willis, I.C., Goodsell, B., Macdonald, G.J., Mayer, D., Powell, A. and MacAyeal,  D.R. Calving and Rifting on McMurdo Ice Shelf, Antarctica. Annals of Glaciology. doi: 10.1017/aog.2017.12, 2017.**

**Banwell, A. F., Datta, R. T., Dell, R. L., Moussavi, M., Brucker, L., Picard, G., Shuman, C. A., and  Stevens, L. A. The 32-year record-high surface melt in 2019/2020 on the northern George VI Ice  Shelf, Antarctic Peninsula, The Cryosphere, 15, 909–925, https://doi.org/10.5194/tc-15-909- 2021, 2021.**
**Burton, J., L. Mac Cathles, W. Grant Wilder, The role of cooperative iceberg capsize in ice-shelf disintegration. Ann. Glaciol. 54, 84–90, 2013.**

**Cape, M. R., Vernet, M., Skvarca, P., Marinsek, S., Scambos, T., & Domack, E. Foehn winds link**

climate-driven warming to ice shelf evolution in Antarctica. Journal of Geophysical Research Atmospheres, 120(21), 11,037–11,057. https://doi.org/10.1002/2015JD023465, 2015.

Datta, R. T., Tedesco, M., Fettweis, X., Agosta, C., Lher- mitte, S., Lenaerts, J. T. M., and Wever, N.: The effect of Foehn-induced surface melt on firn evolution over the north- east Antarctic peninsula, Geophys. Res. Lett., 46, 3822–3831, https://doi.org/10.1029/2018GL080845, 2019.

Glasser, N. F., & Scambos, T. A. A structural glaciological analysis of the 2002 Larsen B ice-shelf collapse. Journal of Glaciology, 54(184), 3–16, 2008.

Kirchgaessner, A., King, J. C., & Anderson, P. S. The impact of Föhn conditions across the Antarctic Peninsula on local meteorology based on AWS measurements. Journal of Geophysical Research: Atmospheres, 126, e2020JD033748. https://doi. org/10.1029/2020JD033748, 2021.

Kuipers Munneke, P., Ligtenberg, S. R. M., Van Den Broeke, M. R., and Vaughan, D. G.: Firn air depletion as a precur- sor of Antarctic ice-shelf collapse, J. Glaciol., 60, 205–214, https://doi.org/10.3189/2014JoG13J183, 2014.

Leeson, A. A., Van Wessem, J. M., Ligtenberg, S. R. M., Shepherd, A., Van Den Broeke, M. R., Killick, R., et al. Regional climate of the Larsen B embayment 1980–2014. Journal of Glaciology, 63(240), 683–690. https://doi.org/10.1017/jog.2017.39, 2017.

Leeson, A. A., Forster, E., Rice, A., Gourmelen, N., & van Wessem, J. M. (2020). Evolution of supraglacial lakes on the Larsen B ice shelf in the decades before it collapsed. Geophysical Research Letters, 47, e2019GL085591. https://doi.org/10.1029/2019GL085591

Lenaerts, J. T. M., Lhermitte, S., Drews, R., Ligtenberg, S. R. M., Berger, S., Helm, V., Smeets, C. J. P. P., Broeke, M. R. van den, van de Berg, W. J., van Meijgaard, E., Eijkelboom, M., Eisen, O., and Pattyn, F.: Meltwater produced by wind–albedo interaction stored in an East Antarctic ice shelf, Nat. Clim. Chang. 2017 71, 7, 58–62, https://doi.org/10.1038/nclimate3180, 2016.

Robel, A. A. and Banwell, A. F.: A speed limit on ice shelf collapse through hydrofracture, Geophys. Res. Lett., 46, 12092–12100, https://doi.org/10.1029/2019gl084397, 2019.

Scambos, T., Hulbe, C., and Fahnestock, M.: Climate-induced ice shelf disintegration in the antarctic peninsula, Antarctic Penin- sula climate variability: Historical and paleoenvironmental per- spectives, Vol. 79, 79–92, American Geophysical Union. Antarct Res. Ser., Washington, DC, 2003.

Scambos, T. A., Hulbe, C., Fahnestock, M., and Bohlander, J.: The link between climate warming and break-up of ice shelves in the Antarctic Peninsula, J. Glaciol., 46, 516–530, 2000.
Scambos, T. A., Bohlander, J. A., Shuman, C. U., and Skvarca, P.: Glacier acceleration and thinning after ice shelf collapse in the Larsen B embayment, Antarctica, Geophys. Res. Lett., 31, L18402, https://doi.org/10.1029/2004GL020670, 2004.

Wiesenekker, J., Kuipers Munneke, P., van den Broeke, M., & Smeets, C. A multidecadal analysis of Föhn winds over Larsen C ice shelf from a combination of observations and modeling. Atmosphere, 9(5), 172. https://doi.org/10.3390/atmos9050172, 2018

---

## Author Comment (AC2)

Thank you for your comments and suggestions. We believe this manuscript will improve significantly with your suggestions and we sincerely appreciate your valuable contributions. We have addressed your comments below marked with [Author Response].

General comments

The authors here use a foehn wind detection algorithm to quantify surface melt magnitude and timing to claim that a foehn wind event pushed the Larsen A ice shelf past a critical stability threshold ultimately leading to its collapse in 1995. Meanwhile, since the Larsen B ice shelf experienced weaker foehn-related melt prior to its collapse in 2002, foehn winds likely preconditioned the ice shelf for collapse. While the foehn detection algorithm provides new, detailed insights into foehn jet positions and foehn wind related melt magnitude, the conclusions regarding ice shelf stability and collapse are underdeveloped and unsupported by the results. I give line-by-line results later, but globally I believe this manuscript suffers from two key elements.

The first is the lack of references to already published work that describe ice-shelf stability processes. Other times, relevant papers are cited, but their conclusions are misrepresented or not mentioned in the text. I give more detailed examples below, but one glaring example is the exemption of discussion from Massom et al., 2018 which discusses of ice shelf collapse triggered by sea ice loss and ocean swells. This paper is cited in the manuscript, but the results about how sea-ice loss and exposure to ocean swells triggered the collapse of the Larsen A and B are never discussed in this manuscript. The authors should consider these processes before claiming foehn winds triggered the collapse of the Larsen A.

[Author Response] - Thank you for your comments and we agree. We agree this manuscript is limited in it's references and in particular articles about föhn winds, föhn-induced surface melt, and ice shelf stability. This manuscript was originally submitted to a short form journal, which limited the number of references. We felt, at the time we submitted this manuscript to The Cryosphere, the amount of references and background regarding föhn winds and ice shelf collapse processes was sufficient. However, after your valuable comments we plan to remedy the lack of background by changing the manuscript, specifically the introduction section to provide a clear overview of the current research to date on föhn winds and föhn-induced melt in the region.

Additionally, with your valuable comments and suggestions and after re-reading many previous studies we have decided to alter the story that we are telling. After reading Massom et al., 2018, which produced a useful conceptual framework for rapid ice shelf collapse and identifies large period ocean swells as the trigger mechanism for the collapse of the Larsen A and B ice shelves, we decided to alter our interpretation of our findings. Fohn winds were present at the time of collapse for both ice shelves which produced enhanced surface melt rates that caused extensive melt ponds over each ice shelf. Additionally, the direction of fohn winds (from the west/northwest direction) pushed/melted sea ice and fast ice away from the calving front of both ice shelves which allowed large ocean waves to trigger collapse, which was also discussed in Banwell et al (2017). We will alter the story of the manuscript to show that without the extensive melt ponds and lakes enhanced by

fohn winds, and the wind direction that pushed protective sea ice away from the calving front, large-scale hydrofracture cascaded and subsequent collapse would not have taken place. We will also change the title of the manuscript to not suggest fohn winds triggered collapse, but instead played a supporting role in the rapid collapse of LA and LB ice shelves.

The authors also cite Scambos et al., 2000, but appear to miss some important observations from that study. The authors in that study cite a storm as the trigger for the final disintegration of the Larsen A, but this fact does not appear in this paper's discussion of the Larsen A collapse. Is the foehn wind event here related to that storm mentioned in Scambos et al., 2000? Also, Scambos et al., 2000 mentions the Larsen A suffered major retreats in 1987 and 1989 which did not appear to be major foehn event years according to this study but did precondition the ice shelf for collapse which contradicts one of the authors' conclusions.

[Author Response] - After extensive research to learn more about the storm mentioned in Scambos et al., 2000, and reviewing surface observations and model simulations, we determined that "the storm" mentioned was the powerful fohn wind events discussed in this study as there were no other major storm systems in the region.

In response to your comment about Scambos et al., 2000 and the Larsen A retreats of 1987 and 1989 and the lack of fohn winds in those years. Just because there was a low fohn melt year does not mean fohn winds could not have played a role in collapse events. Also, tt is hard to distinguish what caused these events because of the lack of observation and timing. We know that the events occurred in 1987 and 1989, but we do not have a clear research, first hand observations, or satellite observations of when the events actually occurred, so it is difficult to attribute a cause for these events. I have extensively researched satellite observations to try and triangulate the timing of these events, but I was unable to powerpoint the exact time of collapse. I also tried to identify other collapse event timing, such as the collapse of Larsen inlet, north of Larsen A in 1987, as well as the minor collapse events of Larsen B in 1998, 1999, and 2000, but again lack satellite observations.

The second issue is claiming one particular process could trigger an ice shelf collapse is a very high bar to pass given the multitude of other processes known to cause ice shelf instability. This manuscript would be much easier to accept as a reader if the authors move their focus away from the supposed novelty of their research and towards the value this research brings to an already rich field of research relating foehn-wind and ice shelf stability. In fact, there are moments when the authors claim to demonstrate a result for the first time when this result was already discussed in previous literature (see comment on line 51). The manuscript would be much easier to digest if the authors moved away from the claim that foehn winds triggered ice shelf collapse and instead focused on highlighting foehn winds as one of many processes that lead to ice shelf instability and the timing of the foehn winds may have played a supporting role in the collapse of the Larsen A.

[Author Response] - We agree. As we mentioned in our first response, we will be altering the title and the direction of the manuscript to show that fohn winds played a role in collapse, but did not trigger collapse. See above for a more detailed explanation.

Line 13: Saying that there are no studies examining surface melt prior to disintegrations is incorrect. You should revisit the Van Den Brooke, 2005 GRL paper that you cited that explicitly studies surface melt on the Larsen B prior to its collapse.

[Author Response] - **These passages and others like it were meant to identify that little research fohn melt was done on time scales shorter than annual or seasonal, however, we see that the way these comments are written make it seem like there is no research on föhn winds and surface melt. We will change all passages in the manuscript to better frame this study among the rich array of studies on föhn winds and surface melt in the region.**

Line 17-19: This claim is based on a premise that foehn wind and surface temperatures remain within historical bounds. The Antarctic Peninsula already experiences large temperature variability and is projected to become warmer which would actually make the extant ice shelves more vulnerable to foehn winds in the future (Siegert et al., 2019; Chyhareva et al., 2019).

[Author Response] - **We agree but wanted to make it clear that when we say extant ice shelves are less vulnerable to collapse than collapsed ice shelves because of large scale surface melt, that it is with the caveat that climate change is not considered here. We plan to change the abstract to better reflect the new direction of the paper and will alter this sentence to be clearer.**

Line 25-27: The claim of novelty seems unwarranted here. Plenty of studies already cited in this manuscript plus some others discuss fohn-related melt mechanisms on the Larsen B ice shelf (see Datta et al. 2019). Plus, Van den Brooke et al., 2005 claims surface melt accelerated the rate of ice shelf retreat, but did not claim it was a leading contributor to the final collapse

[Author Response] - **We did not mean to suggest that there have been no studies that explore surface melt processes in the region, we only meant to identify small gaps in the research such as a lack of short time scale melt rates from fohn winds on Larsen A and B. We see that the current claim does not reflect this sentiment and will be altered to address your comments.**

Line 33-41: I don't understand why the manuscripts claims surface melt as the lead cause of the ice shelf final collapse in the previous paragraph and then point out all the other well documented processes that also affect ice shelf final collapse.

[Author Response] - **There have been a few studies that point to large scale hydrofracture cascades caused by extensive melt ponds as a major factor that led to rapid collapse (Massom et al., 2018, Banwell et al., 2017), however, we understand your point and believe the sentence can be written better. We plan to completely alter the introduction to better fit the new direction of the manuscript so this will likely be changed.**

Line 30: This is a strange claim to make in the introduction. If this claim is valid, then it should first be proven in the results and then mentioned in the conclusion. (Changed this sentence to reflect work in previous study)

[Author Response] - **Yes, we agree this statement feels strange and will be altered with the addition**

**of adding reference to previous work.**

Line 51: This is repeating a claim from the first paragraph that incorrectly states no previous research has been done on foehn-related melt around ice shelf collapses. This study may certainly give further detail on the intensity and spatial distribution of the foehn wind, but certainly is not the first.
[Author Response] - **We agree as stated above, these comments and others like it will be changed.**

Line 54: The temperature trends on the Antarctic Peninsula are a bit more complicated than this. Bozkurt et al., 2020, Carrasco et al., 2021, and Turner et al., 2016 paint a different picture where temperature trends are periodic and dependent on the location along the AP.
[Author Response] - **We agree and will alter this sentence.**

Line 57: Questions 1 and 3 are very important and reasonable questions to address in this manuscript. Question 2 is much harder to answer with certainty without considering all the other processes (atmospheric and non-atmospheric) that could affect ice-shelf stability.
[Author Response] - **Thank you, we plan to alter Question 2 to fit more in line with the new direction of the manuscript. We plan for it to say something like: " 2) What role did fohn winds and associated melt play for the trigger for the collapse of the LAIS and LBIS?"**

Line 87: What height is the air temperature measured at?
[Author Response] - **10 Meters, which we will add to the manuscript.**

Line 95: It is stated again that is foehn detection method is the most accurate compared to previous work without explaining what this previous work is or why it is the most accurate. I also believe this is not the first foehn detection algorithm to incorporate station observations and model output (see Turton et al., 2018). The authors should include some information comparing the foehn detection of their algorithm against other foehn detection algorithms even if that data is presented in Laffin et al., 2021.
[Author Response] - **We did complete an intercomparison sensitivity study detailed in Laffin et al., 2021, comparing other identification methods (Datta et al., 2019, Turton et al., 2018). We will make sure to discuss this study in this manuscript as well as provide the summary statistics from that study in the supplement.**

Line 108: Perhaps explain which variables you used to make the two-tailed t-test statistic. "Mean of both ice shelves" is vague
[Author Response] - **We will clarify which variables were used in our t-test.**

Line 115-119: This seems like background information on the physics of foehn winds that would be better suited in the introduction section.
[Author Response] - **Yes, We agree and will include a section in the introduction that discusses how fohn winds form as well as previous research. Again this was tied to the original short for journal.**

Line 131: This might be a personal preference, but you should change your figure numbers/order if you are referring to figure 5 before figure 3.
[Author Response] - Yes, we recognize this only provides confusion and will alter the figure numbers.

Line 132: You should present some results on foehn frequency before presenting the foehn related melt percentage. This would help put these melt-percentages in a better context.
[Author Response] - Yes, we agree and will add in melt frequency stats.

Line 137: If the SCAR inlet is not impacted by a foehn jet, where is the foehn wind influence coming from?
[Author Response] - SCAR inlet is not directly impacted by a föhn jet, but still experiences clear skies and weak föhn wind influence from the overall descending air that leads to warm winds and more importantly enhances shortwave radiation.

Line 139 – 142: You are contradicting yourself or at least unclear in these two sentences. First you claim that the disparity in foehn-related melt percentages among the ice shelves implicates the foehn as a contributor to the LAIS and LBIS collapse. This is a very strong assertion. It explains differences in melt rates on the ice shelves but saying this contributes to their collapse is a stretch. Then the next sentence is confusing and muddles your message about whether foehn is important or not to collapse. Probably easier to say that your results indicate foehn is one of many processes that weakened the LAIS and LBIS.
[Author Response] - We agree our assertion is unfounded and should state that fohn winds are one of the reasons the ice shelves destabilized. We will clarify these sentences with less conflicting words.

Line 149-152: If extensive foehn wind jets help explain why the LAIS and LBIB collapsed, then why have they not caused the collapse of the LCIS? Is there research showing that having melting at the terminus is essential for an ice shelf collapse? Discussed above in the beginning of paragraph
[Author Response] - We state in the manuscript *"LCIS on the other hand is impacted by four major jets and regularly experiences föhn-induced melt lakes, particularly in Cabinet inlet. However, the vast size of the LCIS does not allow the föhn-induced melt to reach the terminus. The föhn melt mechanism breaks down by mixing with cold air which reduces the intensity of the föhn jets from their peak at the base of the AP mountains to the calving front (Figure 3b)".* Massom et al., 2018 states that extensive melt ponds are an essential prerequisite for rapid collapse. With the change in direction of the manuscript we will make sure to fit our findings in the context of other research about collapse in a more clear manner.

Line 153-154: Previous literature already shows that foehn winds have a major impact on ice shelf surface melt and the framing of this sentence makes your results sound novel when in fact it would be more accurate to say that your results back up and enhance preexisting knowledge while citing these sources. (Find fohn melt research on LA and LB)
[Author Response] - We see how this statement makes our research sound novel so we will

alter this sentence to better explain how our research fits in the context of other research.

Line 181: It's a bit confusing to see the authors use satellite imagery from the 1992/1993 melt season as an analogue to the 1994/1995 melt season, but then later argue that despite the two seasons had similar amounts of foehn-related melt, the reason the Larsen A collapsed in 1995 and not in 1994/1995 was the timing of the surface melt. This argument needs more analysis of the background state of the Larsen A in 1992/1993 versus 1994/1995 to explain more clearly what was so special in 1994/1995.
[Author Response] - We do see how this argument may seem contradictory. With the new direction of this manuscript, we will also discuss a lack of sea ice in 1995 that triggered collapse, while in 92/93 sea ice was present during most of the summer and so protected the calving front from collapse.

Line 204-205: The total surface melt results are interesting, but would considering the size of the ice shelves change the perception of importance in regard to ice shelf destabilization? For instance, the Larsen C is much larger than the SCAR inlet ice shelf so total melt amounts would be difficult to compare. Melt per area would be a better metric.
[Author Response] - We already calculated for the mean melt over the entire ice shelf, and will clarify this in the manuscript

Line 212-214: The statement about the future resilience of the other ice shelves is problematic as it ignores potential future changes in foehn wind patterns. Especially since I believe your foehn wind detection algorithm only detects foehn winds when the temperature is above 0°C. There could be foehn events that currently do not push the temperature above this threshold which are not considered by your algorithm. But theoretically, if air temperatures rises along the Larsen C, then your algorithm would start detecting more foehn wind events. Deleted and discussed later in the conclusions and discussion.
[Author Response] - You are correct that our algorithm only identifies fohn winds above freezing and with climate change more southern locations will receive more fohn induced melt. Its hard to identify what that impact will be however but would be a great future direction of study. We will make sure this point is discussed in the discussion section and highlighted in the conclusions.

Line 227-228: The liquid-to-solid ratio (LSR) analysis here includes foehn-related melt and non foehn related melt. As mentioned earlier, it would be helpful to know the foehn wind frequency according to your detection algorithm in order to judge the significance of this result.
[Author Response] - Agreed, we will add melt frequency statistics.

Line 244-245: There are likely many other differences between the Larsen A and B and the other ice shelves beyond foehn wind patterns. At the very least, sea-ice coverage and ocean forcings are different (see Massom et al., 2018). As I am not a glaciology expert, I cannot say for certain what the differences are structurally between these ice shelves, but it probably is wise to cite some papers regarding ice dynamics to verify this statement.

[Author Response] - We agree and will add more references with our research. As stated above we will be adjusting the direction of the paper.

Line 254: One thing missing about this discussion on the timing of the ice shelf collapses is if ice shelves have existed for thousands of years and foehn winds are a quasi-permeant feature on the Larsen ice shelves, why did foehn winds only trigger the Larsen A collapse relatively recently?
[Author Response] - Great point! We will add this to the discussion section.

Line 270: I feel like you cannot conclude foehn-related surface melt triggered the Larsen A collapse without taking into consideration factors like basal melting.
[Author Response] - We agree which is why we are altering the direction of the paper indicating a supporting role of fohn winds for ice shelf collapse, and not trigger.

Line 282-283: How are you certain that a combination of factors also did not trigger the final disintegration of the Larsen A? In fact, in Massom et al., 2018, it was observed that sea-ice loss allowed ocean swells to apply a strain along the ice-shelf front which is cited as a possible trigger of the Larsen A collapse. This needs to be considered and discussed in this manuscript.
[Author Response] - We agree which is why we are altering the direction of the paper indicating a supporting role of fohn winds for ice shelf collapse, and not trigger.

Line 289-290: This sentence disregards the gradual retreat of the ice shelves like the major retreats the Larsen A experienced in 1987 and 1989 mentioned in Scambos et al., 2000.
[Author Response] - We will look at satellite observations for these events to see if we can pinpoint the lime of collapse to assess if winds helped with these events as well.

Line 292-293: You cannot come to this conclusion if your foehn detection algorithm only detects foehn when the temperature is above 0°C which will likely occur more often over the Larsen C according to future climate projections (Siegert et al., 2019) (Chyhareva et al., 2019).
[Author Response] - We will look more into these conclusions to better assess possible future research directions, but will likely take out this assertion.

References:

Bozkurt, D., Bromwich, D. H., Carrasco, J., Hines, K. M., Maureira, J. C., and Rondanelli, R.: Recent Near-surface Temperature Trends in the Antarctic Peninsula from Observed, Reanalysis and Regional Climate Model Data, Adv. Atmos. Sci., 37, 477–493, https://doi.org/10.1007/s00376-020-9183-x, 2020.

Carrasco, J. F., Bozkurt, D., and Cordero, R. R.: A review of the observed air temperature in the Antarctic Peninsula. Did the warming trend come back after the early 21st hiatus?, 100653, https://doi.org/10.1016/j.polar.2021.100653, 2021.

Chyhareva, A., Krakovska, S., and Pishniak, D.: Climate projections over the Antarctic  Peninsula region to the end of the 21st century. Part 1: cold temperature indices, UAJ, 62–74, https://doi.org/10.33275/1727-7485.1(18).2019.131, 2019.

Datta, R. T., Tedesco, M., Fettweis, X., Agosta, C., Lhermitte, S., Lenaerts, J. T. M., and Wever,  N.: The Effect of Foehn-Induced Surface Melt on Firn Evolution Over the Northeast Antarctic  Peninsula, Geophysical Research Letters, 46, 3822–3831, https://doi.org/10.1029/2018GL080845, 2019.

Massom, R. A., Scambos, T. A., Bennetts, L. G., Reid, P., Squire, V. A., and Stammerjohn, S.  E.: Antarctic ice shelf disintegration triggered by sea ice loss and ocean swell, Nature, 558, 383– 389, https://doi.org/10.1038/s41586-018-0212-1, 2018.

Turner, J., Lu, H., White, I., King, J. C., Phillips, T., Hosking, J. S., Bracegirdle, T. J., Marshall,  G. J., Mulvaney, R., and Deb, P.: Absence of 21st century warming on Antarctic Peninsula  consistent with natural variability, Nature, 535, 411–415, https://doi.org/10.1038/nature18645,  2016.

Turton, J. V., Kirchgaessner, A., Ross, A. N., and King, J. C.: The spatial distribution and  temporal variability of föhn winds over the Larsen C ice shelf, Antarctica, Q.J.R. Meteorol. Soc.,  144, 1169–1178, https://doi.org/10.1002/qj.3284, 2018.

Scambos, T. A., Hulbe, C., Fahnestock, M., and Bohlander, J.: The link between climate warming and break-up of ice shelves in the Antarctic Peninsula, J. Glaciol., 46, 516–530, https://doi.org/10.3189/172756500781833043, 2000.

Siegert, M., Atkinson, A., Banwell, A., Brandon, M., Convey, P., Davies, B., Downie, R.,  Edwards, T., Hubbard, B., Marshall, G., Rogelj, J., Rumble, J., Stroeve, J., and Vaughan, D.: The  Antarctic Peninsula Under a 1.5°C Global Warming Scenario, Front. Environ. Sci., 7, 102, https://doi.org/10.3389/fenvs.2019.00102, 2019.

van den Broeke, M.: Strong surface melting preceded collapse of Antarctic Peninsula ice shelf, Geophys. Res. Lett., 32, L12815, https://doi.org/10.1029/2005GL023247, 2005.

---

## Author Comment (AC3)

Thank you Dr. Rott for your comments and suggestions. I really enjoyed our conversation. We have addressed your comments below marked with [Author Response].

**Short Comment** on "Antarctic Peninsula ice shelf collapse triggered by föhn wind-induced melt" by M. K. Laffin et al., submitted to *The Cryosphere Discussions*, 25 Oct. 2021.

*Commenter:* Helmut Rott

The authors explored mechanisms triggering the rapid collapse events of the Larsen A and B ice shelves, using a regional climate model and Machine Learning analysis in order to investigate the influence of föhn winds and associated melt on the surface liquid water budget. They conclude that increased surface melt due to föhn supplied water to melt lakes, inducing the crossing of a critical stability of water depth that triggered the rapid Larsen A collapse. The authors claim a lack of high resolution satellite imagery during the collapse and deduce estimates on melt lake surface area from an AVHRR image (1 km spatial resolution) of 8 December 1992. In fact, high resolution synthetic aperture radar (SAR) images (ca. 25 m resolution) of the ESA ERS-1 satellite were acquired during the disintegration event, on 25, 28 and 30 January and 2. February 1995. Some of the ERS-1 SAR images acquired over Larsen Ice shelf are shown in Rott et al., 1998 (paper cited by Laffin et al.). Furthermore, Rott et al., 1996 (not cited), show ERS SAR images acquired during the event and present a report on the state of Larsen A Ice Shelf two months before the collapse, built on field observations.

[Author Response] - Thank you for bringing some of these images to our attention. We knew of the images but did not know they were at a high resolution. We will be sure to better study these high resolution images.

The ERS SAR images, as well as the report on the field observations in October and November 1994 disprove the hypothesis of Laffin et al. that the Larsen A collapse in January 1995 was triggered in the short-term by hydrofracture processes. According to the ERS image of 25 Jan. 1995, close to the start of the main disintegration event, the extent of surface lakes on Larsen A Ice Shelf amounted to about 1% of the total area (Fig. C1 below). In Oct./Nov. 1994 the ice shelf was already heavily fractured. Cold temperatures and an extended pre-frontal cover of fast ice kept the ice shelf from breaking apart. Details are reported in Rott et al., 1996, e.g. referring to an ice wedge protruding from the level ice shelf several km inland of the front (Fig. C2), on cracks along the border between the Larsen A and Seal Nunataks ice shelves, and rifts along the coastline line to the peninsula.

[Author Response] - We agree that some of the images, specifically the 25 Jan 1995 do not show that melt ponds are present on the Larsen A Ice Shelf prior to collapse. As you suggest this shows that at the time of collapse, the ice shelf could not support melt ponds and did not experience large scale hydrofracture cascades as we and others have suggested. However, as we discussed in our zoom meeting, the ERS image from 28 Jan 1995, shows that melt ponds are present on what remains of the ice shelf before total collapse (See the Figure A1 below). There are a multitude of reasons why there are no melt ponds in the 25 Jan image but are in the 28 Jan image, but it does show that to some capacity the Larsen A ice shelf was able to support melt ponds even though its surface was no longer smooth and cracks were present.

[Figure]

Figure A1: ERS 1 SAR image from 28 Jan 1995 of the Larsen A ice shelf with melt ponds/lakes circled in orange.

Regarding the temporal sequence of the collapse event, the ice shelf section downstream of Dinsmoor Bombardier-Edgeworth (DBE) glaciers retreated to the grounding line faster than downstream of Drygalski Glacier along which Laffin et al. show the location of the LA föhn jet. The pre-collapse crack density was highest on the DBE ice shelf section. Another striking incident was the rapid off coast drift of detached icebergs and growlers, gaining 40 km in distance between 28 and 30 January 1995, an indication for oceanic mechanic forcing as main factor for the rapidity of disintegration.

[Author Response] - Yes, you are correct that the sequence of events of collapse began with a faster disintegration of the downstream section of DBE outlet glaciers, however as we discussed this would make sense because of the already fractured nature of that portion of the ice shelf. The DBE section of the ice shelf surface was inherently rough and cracked due to the suturing of the three glaciers and flow direction.

As far as the rapid off coast drift of the detached icebergs, you are correct that there was some form of ocean forcing. Also however, the powerful fohn winds that were present during the collapse as identified in this and other studies also pushed the icebergs off coast leading to this iceberg travel pattern. As we discussed it is unfortunate that ocean observations are not available during this time, but this theory may benefit from modeling studies to better understand what was happening from an ocean standpoint.

This comment is not a review of the paper. Nevertheless I want to address some further issues:

*Melt pattern on Larsen A Ice Shelf*: The model simulations (Figs. 3 and 6) indicate reduced melt in the northern section of the ice shelf (downstream of DBE glaciers). ERS SAR images, acquired in years preceding the collapse, as well as the 25 Jan. 1994 image (Fig. C1), do not show any significant difference in melt intensity between different ice shelf sections.

[Author Response] - This is certainly a good point. One way to corroborate if the fohn jet locations simulated by RACMO2 are in the correct place is to confirm the location of other jets, such as those found on the Larsen C ice shelf and studies extensively using airborne observation, surface

observations, and model simulations (Kuipers Munneke et al., 2014; Elvidge et al., 2016). When we compare the RACMO2 identified locations of fohn jets on the Larsen C with those observed, we confirm they are in the correct location.

*Position of föhn jets*: The uniqueness of the föhn jets on Larsen A (Drygalski Glacier) and on Larsen B (Hektoria – Green glaciers) needs to be reconsidered. ERS SAR images during the Larsen A event show high reflectivity of the ocean surface and rapid off-coast displacement of ice downstream of DBE glaciers, indications for strong off-shore winds. During the Larsen B disintegration event rapid off coast drift of icebergs was observed also downstream of Crane and Jorum glaciers (Rack et al., 2004). [Author Response] - You are correct about the images showing strong winds, but this does not disqualify the location of the fohn jets identified in this study. The location of the fohn jets are indicated by fohn-induced surface melt and not wind speed. It is entirely possible that the winds north and south of these fohen jets on both Larsen A and B are just as strong as within the jet, but the warming and melt effect may be different. Additionally, even though a location is not affected by a fohn jet, it is still possible to experience a weaker fohn wind that produces a similar wind/wave roughness identifiable for SAR imagery.

*Line 289:* "*... the LAIS and LBIS collapsed catastrophically within weeks and not through long-term thinning and retreat like other ice shelves...*" The gradual retreat of the Larsen A front between Seal Nunataks over 20 years up to the collapse is documented by means of satellite images starting in 1975 (Skvarca, 1993; Rott et al., 1996). [Author Response] - Thank you for your comment, I will make sure to mention the gradual thinning of the ice shelves prior to rapid collapse.

*References:*

Rack, W. and H. Rott, H.: Pattern of retreat and disintegration of the Larsen B ice shelf, Antarctic Peninsula, Ann. Glaciol., 39, 505 – 510, 2004.

Rott H., Skvarca, P., and Nagler, T.: Rapid collapse of northern Larsen Ice Shelf, Antarctica, Science, Vol. 271, Issue 5250, 788-792, 1996.

Skvarca, P.: Fast recession of the northern Larsen Ice Shelf monitored by space images, Ann. Glaciol., 17, 317-321, 1993.

[Figure]

*Fig. C1.* Section of ERS-1 SAR image, covering Larsen A Ice Shelf, 25 January 1995. CW – Cape Worsley, D – Drygalski Glacier, DBE – Dinsmoor-Bombardier-Edgeworth glaciers, S - Sobral Peninsula.

[Figure]

[Figure]

*Fig. C2.* Ice wedge on Larsen A Ice Shelf, located several km inland of the front (left). Cracks near the ice front (right). Photos H. Rott, 24 Oct. 1994.

---

## Author Response (AR1)

**List of all relevant changes**

1. Changed the title of the manuscript.
2. Created a new narrative of the manuscript to put fohn winds in a supporting role and not the trigger of collapse.
3. New Introduction to provide new narrative and more in depth reference review of fohn winds in the region. (also includes shifting a review of fohn winds from the results section to intro)
4. Moved the liquid-to-solid ratio (LSR) equation from results to the methods section.
5. Added a previously done fohn identification sensitivity study in the methods from Laffin et al., (2021).
6. Altered Figure 2b to include fohn melt frequency.
7. Added additional discussion about sea ice during collapse events.
8. Altered the conclusions to not suggest fohn winds trigger collapse, but played a role in it.
9. Added other clarifying remarks throughout.
10. Added 17 new references not previously mentioned in past manuscript versions.

**Point by Point Response to Reviewers**

**RC1-**

Thank you for your comments and suggestions. We believe this manuscript will improve significantly with your suggestions and we sincerely appreciate your valuable contributions. We have addressed your comments below marked with [Author Response].

Review for "*Antarctic Peninsula ice shelf collapse triggered by fohn wind -induced melt*" by Laffin et al.

General comments

This paper has the potential to be a very interesting study about the possible influence of föhn winds on the large-scale collapse past events of the Larsen A and B ice shelves, and on potential future break-up events of Antarctic Peninsula ice shelves. However, in its current state, it is poorly written and badly structured in many places, e.g. why are föhn winds only defined and described in the Results and not in the Introduction? I give more examples in my line-by-line comments below.

[Author Response] - We agree that the structure of the manuscript can be improved, especially with an overview of föhn winds in the introduction section. This manuscript was originally submitted to a short form journal which is why it was structured differently then a typical Cryosphere article. We overlooked this fact when we re-wrote the manuscript for The Cryosphere and have made changes to the manuscript that are more in line with The Cryosphere structure. Please see the below comments that show and explain the changes to this manuscript in more detail.

Additionally, the current paper includes extremely limited references to relevant work that has already been done (particularly regarding föhn winds, but also regarding surface melt processes in general). A good example of this is the sentence in the abstract (line 13/14) which reads: "However, no studies examine the timing, magnitude, and location of surface melt processes immediately preceding these disintegrations." This statement about the Larsen A and B ice shelves is entirely incorrect as there have been many studies that have examined surface melt processes on these ice shelves, e.g. Scambos et al (2000, 2003, 2004) Glasser and Scambos (2008), Leeson et al (2017, 2020), Banwell et al (2013; 2014), Kuipers Munneke et al (2014), Lenaerts et al (2017) and Robel and Banwell (2019), to name just a few. I suggest that this sentence (and similar sentences in the Introduction) are reworded to specifically focus on the research to-date regarding effects of foehn winds on surface melt on ice shelves. Currently this paper references only a few such föhn wind studies; the following key studies about föhn wind induced ice-shelf melt are missing: Datta et al (2019), Wiesenekker et al (2018), Bozkurt et al (2018), Kirchgaessner et al (2021), and I suspect a good few others. Kirchgaessner et al (2021) is particularly relevant to the current study as it also focuses on AP ice shelves. As I am not 100% up to date with the ice-shelf melt-related föhn wind literature myself, it has been hard for me to give this paper thorough review given that the authors have not placed their study in the context of existing knowledge from other literature.

[Author Response] - We agree this manuscript is limited in it's references and in particular articles about föhn winds and föhn-induced surface melt. See lines 60-84 of the new manuscript which includes a more detailed explanation of fohn winds. We have also added an overview of the most relevant research on fohn winds in the region.

Also, in regard to your comment about Line 13/14:, *"However, no studies examine the timing, magnitude, and location of surface melt processes immediately preceding these disintegrations."*, and others like it, the passages were meant to show that little research was done on time scales shorter than annual or seasonal, however, we see that the way these comments are written make it seem like there is no research on föhn winds and surface melt. We have changed all passages in the manuscript to better frame this study among the rich array of studies on föhn winds in the region.

Finally, unlike the LAIS, I think I agree with the statement that the 'LBIS collapse was not directly related to the impact of föhn-induced melt', e.g. as the authors state on line 190 and in the Conclusion. However as the initial LBIS collapse on Feb 9 2002 coincided with a föhn wind event, I wonder if the authors have considered the idea that that föhn wind event may have helped produce sufficient surface meltwater such that the drainage of multiple surface lakes via hydrofracture cascades may have been triggered (i.e. 'chain reaction' lake drainage), thereby resulting in LBIS's near complete collapse a couple of weeks later (see Banwell et al 2013, Robel and Banwell, 2019). So in that sense, I am wondering what the authors think about the idea of föhn winds having been an indirect cause of LBIS's break-up?

[Author Response] - This is a very interesting question that inspired us to change the manuscript. After reading Massom et al., 2018, which theorized a useful conceptual framework for rapid ice shelf collapse and identifies large period ocean swells as the trigger mechanism for the collapse of the Larsen A and B ice shelves, we decided to alter our interpretation of our findings. Fohn winds were present at the time of collapse for both ice shelves which produced enhanced surface melt rates that caused extensive melt ponds over each ice shelf. Additionally, the direction of fohn winds (from the

west/northwest direction) pushed/melted sea ice and fast ice away from the calving front of both ice shelves which allowed large ocean waves to trigger collapse, which was also discussed in Banwell et al (2017). We have changed the manuscript to show that without the extensive melt ponds and lakes enhanced by fohn winds, and the wind direction that pushed protective sea ice away from the calving front, large-scale hydrofracture cascaded and subsequent collapse would not have taken place. See the new abstract/introduction/conclusion that re-frames our results with this narrative. We have also changed the title of the manuscript to not suggest fohn winds triggered collapse, but instead played a supporting role in the rapid collapse of LA and LB ice shelves.

Specific comments

Line 11: 'Add 'grounded' before 'glaciers'.
[Author Response] - This was changed to clarify grounded glaciers. See line: 9

12/13: In addition to surface melting, a mention of lake drainage via hydrofracture, and/or cascades (or a chain reaction) of lake drainage events could be mentioned here.
[Author Response] - We have altered the abstract to include hydrofracture cascades. Line: 12

13/14: See 'general comment' above.
[Author Response] - We have altered the abstract to include hydrofracture cascades. Line: 12. It was already discussed in the manuscript but we have also added additional clarification in the introduction. .

16: Mention the paper's specific focus on Antarctic Peninsula shelves.
[Author Response] - We clarified the study region. Line: 14

18: 'less' vulnerable compared to what?
[Author Response] - We agree and have altered the abstract significantly and have taken this language out for clarification. See line: 18-21

22: 'Forensic' is the wrong word as there is no link with crime.
[Author Response] - We meant to say that examination of past events is useful. We altered "forensic" with re-evaluation. Line: 27

26 – 28: Similar to the comment I made about line 13/14 in the abstract, this sentence is entirely incorrect and does not reference prior key studies regarding both surface melt processes on ice shelves and föhn winds specifically. I suggest you add at least the references I mention above, but I will have missed some.
[Author Response] - We have completely re-written the introduction to include valuable background and references as well as frame our findings in the context of other studies. See the new introduction.f

29 – 30: Be clear that you are using a ML method you developed in a previous study (at least that

is what I am guessing), i.e. Laffin et al (2021), and reference that. Currently this sentence is vague.
[Author Response] - **Yes, this method was developed in Laffin et al., 2021. We have clarified this fact in Line: 109**

30- 32: You state that your method is the 'most accurate', but you do not state what other methods/studies you are comparing it too, and nor do you state how you came to such a conclusion? Did you do some sort of intercomparison study? If so, that should be briefly explained.
[Author Response] - **We did complete an intercomparison sensitivity study detailed in Laffin et al., 2021, comparing other identification methods. We have added the sensitivity study statistic table from Laffine et al., 2021 into the supplement. See supplement and Line: 119**

33 – 41: This is interesting, as it totally contradicts the statements made in the Abstract and Introduction about there being no studies that have looked at such ice shelf melt/collapse processes! Additionally, by 'warm water intrusion', I assume you are referring to enhanced basal melting? And another good example of a study that demonstrated how sea swell caused ice shelf frontal break up is Banwell et al (2017).
[Author Response] - **We do see the contradictions in this statement and those made in the abstract and throughout the manuscript. We added additional background for ice shelf stability and fohn winds in the region. We also clarify our mention of "warm water intrusion" to basil melt as well as reference Banwell et al., (2017) in regard to ocean swell stress on the calving front.**

43: For the 1 meter lake depth reference for LBIS, the two references given are incorrect. They should be Glasser and Scambos (2008) and Banwell et al (2014).
[Author Response] - **We have fixed this embarrassing oversight.**

47: Regarding 'ice shelves into sections with aspect ratios that support unstable rollover', Burton et al (2013) would be a very appropriate reference to add.
[Author Response] - **We have added this reference. Line: 52**

48: Robel et al (2019) is incorrect. It should be Robel and Banwell (2019).
[Author Response] - **Fixed. Line: 52**

49 - 51: The first part of the following sentence requires references, and the second part is incorrect (for the reasons I give above in General Comments): '*Previous research acknowledges enhanced surface melt during years of collapse and the presence of föhn wind events in the region, however, no attempt to produce a timeline of total melt quantity or melt caused by föhn before and during ice shelf breakup has been undertaken*'
[Author Response] - **With the alteration of the narrative we have completely taken this sentence out of the manuscript.**

52/53: Poor English. Reword.
[Author Response] - **We have changed the manuscript and taken this sentence out, and replaced it with Line: 57.**

55 – 58: These questions are good; clear and precise.

[Author Response] - Thanks!

59/60: 'spatial distribution' of what? Poor English.
[Author Response] - We meant to say the distribution of foehn-induced surface melt. We have clarified this sentence to read, "To address these questions we consider three metrics: Section 3.1 explores the total annual surface melt quantity induced by föhn winds and how melt is spatially distributed across each ice shelf...".

85: It needs to be much clearer that the current study uses a föhn detection algorithm developed in a prior study (Laffin et al. 2021), and NOT in this study (at least that is my understanding from the current paper).
[Author Response] - Yes, this method was developed in Laffin et al., 2021. We have clarified this fact in Line: 109

86 – 97: It would be interesting for the authors to compare how their algorithm compares to that used by Datta et al 2019 ('Foehn Index'; also used in Banwell et al. 2021) and perhaps other existing algorithms too. E.g., on what basis/using what evidence can the authors state that there '*method is the most accurate compared to previous work*' (without even giving reference to that previous work).
[Author Response] - We did complete an intercomparison sensitivity study detailed in Laffin et al., 2021, comparing Datta et al 2019 and other identification methods. We have added the sensitivity study statistic table from Laffine et al., 2021 into the supplement. See supplement and Line: 119

105: I think these should more accurately be described as ice shelf "areas" given that Larsen C is split into two areas. Also, I suggest listing those ice shelves/areas in this sentence.
[Author Response] - We agree that this is not clear and have changed the sentence to say ice shelf areas, as well as name those areas in reference to Figure 1. Line: 130

113: you have already defined AWS elsewhere.
[Author Response] - We adjusted the manuscript.

116 – 120: This useful definition/description about föhn winds needs to be moved into the Introduction; it does not belong here.
[Author Response] - This is a great point. We have taken this out of the results and added a deeper look into fohn winds in the region into the introduction.

121: '*AP winds from the west and northwest (*föhn *influence)*' is not clear. Are you suggesting that all winds from the W and NW on the AP are föhn? (If so, that isn't clear, and I assume not all winds from that direction are föhn?
[Author Response] - In this region, because of the location of the Antarctic Peninsula range, most winds from the W/NW will have some fohn influence. We have added more references and discussion of fohn winds in the introduction. Lines: 57-78

121/122: I assume this is a result from the current study, but that needs to be made clear if so.
[Author Response] - Yes, this is a result from this study. We included this information in Figure 2, however, we have made our results more clear with specific percentages from our findings to

compliment the figure. Line: 163

129: '*The degree to which föhn winds impact surface melt on each ice shelf varies...*' state what timescale(s) are being considered here.
[Author Response] - We have altered this sentence to clarify the timescales. Line: 161

131: Figure 5 is mentioned before figures 3 and 4 have been mentioned.
[Author Response] - We have re-worked the manuscript and ensured all figures are identified chronologically.

140/141: I simply do not know what the authors are trying to state by the following sentence: '*However no single factor, including föhn-induced melt rate, lessens the influence of all the other factors that contributed to these collapses.*'
[Author Response] - We agree this sentence is confusing and have removed it from the manuscript.

153/54: For the first part of this sentence, please acknowledge (and reference) other studies that have also established this fact.
[Author Response] - Yes, there are other studies who have established this fact which we have referenced. Line: 183

168: Banwell et al (2013) did not study Larsen A.
[Author Response] - We have corrected this oversight.

190: Please see the final paragraph in my 'general comments' above.
[Author Response] - See our response above.

211-225: It seems like some of this material (inc. equation1) should be in the Methods, not Results?
[Author Response] - Yes, since the manuscript was originally submitted to a short form journal it was best placed in this section. We agree it is now better suited in the methods section. See lines 138-142

229/230: Again, discuss this statement in the context of the findings of other studies.
[Author Response] - We have altered the narrative to include out study with many other notable studies about ice shelf stability and fohn winds.

251: Glasser et al 2018 should be 'Glasser and Scambos (2008)', and Glasser et al (2021) is not in the reference list.
[Author Response] - We have fixed this embarrassing oversight everywhere in the manuscript.

274: Satellite-derived depths of lakes are in Banwell et al (2014).
[Author Response] - We have fixed this embarrassing oversight.

278 – 281: The authors state the following two sentences, which I disagree with: '*The large melt volume in a relatively short amount of time spatially expanded and increased melt lake formation*

*and depth, filled crevasses, increased water pressure on the crevasse tip and walls and triggered large-scale hydrofracture cascades that led to catastrophic disintegration of the LAIS (Scambos et al., 2000; Banwell et al., 2013). The same cannot be said about the LBIS'.* The processes described in the first part of the sentence are what various studies have proposed caused the ultimate collapse of the LBIS, but I am not aware of any study have proposed the same mechanism for LAIS (Scambos et al 2000 or Banwell et al 2013 certainly did not).

**[Author Response] - Thank you for this comment. This is one of the reasons we have decided to shift the focus of the manuscript story to put fohn winds and associated melt in a support role for collapse and not the trigger. This change is discussed in more detail above.**

290: George VI is not a good example to use here as it has very constrained, compressed ice flow.
**[Author Response] - We agree and have altered this sentence. Line: 328**

293: '*more stable*' than what? This is vague.

294: 'than previously thought' – by who? Give references.
**[Author Response] - We agree this is vague. We have altered this sentence, Line: 329 "We conclude that föhn winds and the associated surface melt played a significant role in the collapses of the LAIS and LBIS, while extant AP ice shelves are not likely to collapse from föhn-induced melt and hydrofracture in today's current climate."**

Figures

Figure 2: I assume the data shown in panels b) and c) are from RACMO2, but that should be clarified.
**[Author Response] - Yes, the data shown is from RACMO2, we have clarified this in the manuscript caption.**

Figure 3: Again, where is the data shown in this figure derived from?
**[Author Response] - Yes, the data shown is from RACMO2, we have clarified this in the manuscript caption.**

Figure 4: Again, please state the source of the data.
**[Author Response] - Yes, the data shown is from RACMO2, we have clarified this in the manuscript caption.**

Figure 5: Again, state the source of the data in the caption, and specify what kind of data it is. 'data' is vague.
**[Author Response] - Yes, the data shown is from RACMO2, we have clarified this in the manuscript caption.**

Figure 6: Again, state data source. And for a), should this be 'total melt'?
**[Author Response] - Yes, the data shown is from RACMO2, we have clarified this in the manuscript caption.**

References (those in **bold** are not referenced in the current paper)

Bozkurt, D., Rondanelli, R., Marin, J. C., & Garreaud, R. Foehn event triggered by an atmospheric river underlies record-setting temperature along continental Antarctica. Journal of Geophysical Research: Atmospheres, 123, 3871–3892. https://doi.org/10.1002/ 2017JD027796, 2018.

Banwell, A. F., Cabellero, M., Arnold, N., Glasser, N., Cath- les, L. M., and MacAyeal, D.: Supraglacial lakes on the Larsen B Ice Shelf, Antarctica, and Paakitsoq Region, Greenland: a comparative study, Ann. Glaciol., 55, 1–8, https://doi.org/10.3189/2014AoG66A049, 2014.

Banwell, A.F., Willis, I.C., Goodsell, B., Macdonald, G.J., Mayer, D., Powell, A. and MacAyeal, D.R. Calving and Rifting on McMurdo Ice Shelf, Antarctica. Annals of Glaciology. doi: 10.1017/aog.2017.12, 2017.

Banwell, A. F., Datta, R. T., Dell, R. L., Moussavi, M., Brucker, L., Picard, G., Shuman, C. A., and Stevens, L. A. The 32-year record-high surface melt in 2019/2020 on the northern George VI Ice Shelf, Antarctic Peninsula, The Cryosphere, 15, 909–925, https://doi.org/10.5194/tc-15-909- 2021, 2021.

Burton, J., L. Mac Cathles, W. Grant Wilder, The role of cooperative iceberg capsize in ice-shelf disintegration. Ann. Glaciol. 54, 84–90, 2013.

Cape, M. R., Vernet, M., Skvarca, P., Marinsek, S., Scambos, T., & Domack, E. Foehn winds link climate-driven warming to ice shelf evolution in Antarctica. Journal of Geophysical Research Atmospheres, 120(21), 11,037–11,057. https://doi.org/10.1002/2015JD023465, 2015.

Datta, R. T., Tedesco, M., Fettweis, X., Agosta, C., Lher- mitte, S., Lenaerts, J. T. M., and Wever, N.: The effect of Foehn-induced surface melt on firn evolution over the north- east Antarctic peninsula, Geophys. Res. Lett., 46, 3822–3831, https://doi.org/10.1029/2018GL080845, 2019.

Glasser, N. F., & Scambos, T. A. A structural glaciological analysis of the 2002 Larsen B ice-shelf collapse. Journal of Glaciology, 54(184), 3–16, 2008.

Kirchgaessner, A., King, J. C., & Anderson, P. S. The impact of Föhn conditions across the Antarctic Peninsula on local meteorology based on AWS measurements. Journal of Geophysical Research: Atmospheres, 126, e2020JD033748. https://doi. org/10.1029/2020JD033748, 2021.

Kuipers Munneke, P., Ligtenberg, S. R. M., Van Den Broeke, M. R., and Vaughan, D. G.: Firn air depletion as a precur- sor of Antarctic ice-shelf collapse, J. Glaciol., 60, 205–214, https://doi.org/10.3189/2014JoG13J183, 2014.

Leeson, A. A., Van Wessem, J. M., Ligtenberg, S. R. M., Shepherd, A., Van Den Broeke, M. R., Killick, R., et al. Regional climate of the Larsen B embayment 1980–2014. Journal of Glaciology, 63(240), 683–690. https://doi.org/10.1017/jog.2017.39, 2017.

Leeson, A. A., Forster, E., Rice, A., Gourmelen, N., & van Wessem, J. M. (2020). Evolution of supraglacial lakes on the Larsen B ice shelf in the decades before it collapsed. Geophysical Research Letters, 47, e2019GL085591. https://doi.org/10.1029/2019GL085591

Lenaerts, J. T. M., Lhermitte, S., Drews, R., Ligtenberg, S. R. M., Berger, S., Helm, V., Smeets, C. J. P. P., Broeke, M. R. van den, van de Berg, W. J., van Meijgaard, E., Eijkelboom, M., Eisen, O., and Pattyn, F.: Meltwater produced by wind–albedo interaction stored in an East Antarctic ice shelf, Nat. Clim. Chang. 2017 71, 7, 58–62, https://doi.org/10.1038/nclimate3180, 2016.

Robel, A. A. and Banwell, A. F.: A speed limit on ice shelf collapse through hydrofracture, Geophys. Res. Lett., 46, 12092–12100, https://doi.org/10.1029/2019gl084397, 2019.

Scambos, T., Hulbe, C., and Fahnestock, M.: Climate-induced ice shelf disintegration in the antarctic peninsula, Antarctic Penin- sula climate variability: Historical and paleoenvironmental per- spectives, Vol. 79, 79–92, American Geophysical Union. Antarct Res. Ser., Washington, DC, 2003.

Scambos, T. A., Hulbe, C., Fahnestock, M., and Bohlander, J.: The link between climate warming and break-up of ice shelves in the Antarctic Peninsula, J. Glaciol., 46, 516–530, 2000.

Scambos, T. A., Bohlander, J. A., Shuman, C. U., and Skvarca, P.: Glacier acceleration and thinning after ice shelf collapse in the Larsen B embayment, Antarctica, Geophys. Res. Lett., 31, L18402, https://doi.org/10.1029/2004GL020670, 2004.

Wiesenekker, J., Kuipers Munneke, P., van den Broeke, M., & Smeets, C. A multidecadal analysis of Föhn winds over Larsen C ice shelf from a combination of observations and modeling. Atmosphere, 9(5), 172. https://doi.org/10.3390/atmos9050172, 2018

RC2-
Thank you for your comments and suggestions. We believe this manuscript will improve significantly with your suggestions and we sincerely appreciate your valuable contributions. We have addressed your comments below marked with [Author Response].

General comments

The authors here use a foehn wind detection algorithm to quantify surface melt magnitude and timing to claim that a foehn wind event pushed the Larsen A ice shelf past a critical stability threshold ultimately leading to its collapse in 1995. Meanwhile, since the Larsen B ice shelf experienced weaker foehn-related melt prior to its collapse in 2002, foehn winds likely preconditioned the ice shelf for collapse. While the foehn detection algorithm provides new, detailed insights into foehn jet positions and foehn wind related melt magnitude, the conclusions regarding ice shelf stability and collapse are underdeveloped and unsupported by the results. I give line-by-line results later, but globally I believe this manuscript suffers from two key elements.

The first is the lack of references to already published work that describe ice-shelf stability processes. Other times, relevant papers are cited, but their conclusions are misrepresented or not mentioned in the text. I give more detailed examples below, but one glaring example is the exemption of discussion from Massom et al., 2018 which discusses of ice shelf collapse triggered by sea ice loss and ocean swells. This paper is cited in the manuscript, but the results about how sea-ice loss and exposure to ocean swells triggered the collapse of the Larsen A and B are never discussed in this manuscript. The authors should consider these processes before claiming foehn winds triggered the collapse of the Larsen A.

[Author Response] - Thank you for your comments and we agree. We agree this manuscript is limited in it's references and in particular articles about föhn winds, föhn-induced surface melt, and ice shelf stability. This manuscript was originally submitted to a short form journal, which limited the number of references. We felt, at the time we submitted this manuscript to The Cryosphere, the amount of references and background regarding föhn winds and ice shelf collapse processes was sufficient. However, after your valuable comments we have altered the introduction section to provide a clear overview of the current research to date on föhn winds and föhn-induced melt in the region along with the most relevant studies that aim to identify ice shelf collapse mechanisms. .

Additionally, with your valuable comments and suggestions and after re-reading many previous studies we have decided to alter the narrative of the manuscript. After re-reading Massom et al., 2018, which produced a useful conceptual framework for rapid ice shelf collapse and identifies large period ocean swells as the trigger mechanism for the collapse of the Larsen A and B ice shelves, we decided to alter our interpretation of our findings. Fohn winds were present at the time of collapse for both ice shelves which produced enhanced surface melt rates that caused extensive melt ponds over each ice shelf. Additionally, the direction of fohn winds (from the west/northwest direction) pushed/melted sea ice and fast ice away from the calving front of both ice shelves which allowed large ocean waves to trigger collapse, which was also discussed in Banwell et al (2017). We have altered the story of the manuscript to show that without the extensive melt ponds and lakes enhanced by fohn winds, and the wind direction that pushed protective sea ice away from the calving front, large-scale hydrofracture cascaded and subsequent collapse would not have taken

place. We have also changed the title of the manuscript to not suggest fohn winds triggered collapse, but instead played a supporting role in the rapid collapse of LA and LB ice shelves.

The authors also cite Scambos et al., 2000, but appear to miss some important observations from that study. The authors in that study cite a storm as the trigger for the final disintegration of the Larsen A, but this fact does not appear in this paper's discussion of the Larsen A collapse. Is the foehn wind event here related to that storm mentioned in Scambos et al., 2000? Also, Scambos et al., 2000 mentions the Larsen A suffered major retreats in 1987 and 1989 which did not appear to be major foehn event years according to this study but did precondition the ice shelf for collapse which contradicts one of the authors' conclusions.

[Author Response] - After extensive research to learn more about the storm mentioned in Scambos et al., 2000, and reviewing surface observations and model simulations, we determined that "the storm" mentioned was the powerful fohn wind events discussed in this study as there were no other major storm systems in the region.

In response to your comment about Scambos et al., 2000 and the Larsen A retreats of 1987 and 1989 and the lack of fohn winds in those years. Just because there was a low fohn melt year does not mean fohn winds could not have played a role in collapse events. Also, it is hard to distinguish what caused these events because of the lack of observation and timing. We know that the events occurred in 1987 and 1989, but we do not have a clear research, first hand observations, or satellite observations of when the events actually occurred, so it is difficult to attribute a cause for these events. I have extensively researched satellite observations to try and triangulate the timing of these events, but I was unable to pinpoint the exact time of collapse. I also tried to identify other collapse event timing, such as the collapse of Larsen inlet, north of Larsen A in 1987, as well as the minor collapse events of Larsen B in 1998, 1999, and 2000, but again lack satellite observations.

The second issue is claiming one particular process could trigger an ice shelf collapse is a very high bar to pass given the multitude of other processes known to cause ice shelf instability. This manuscript would be much easier to accept as a reader if the authors move their focus away from the supposed novelty of their research and towards the value this research brings to an already rich field of research relating foehn-wind and ice shelf stability. In fact, there are moments when the authors claim to demonstrate a result for the first time when this result was already discussed in previous literature (see comment on line 51). The manuscript would be much easier to digest if the authors moved away from the claim that foehn winds triggered ice shelf collapse and instead focused on highlighting foehn winds as one of many processes that lead to ice shelf instability and the timing of the foehn winds may have played a supporting role in the collapse of the Larsen A.

[Author Response] - We agree. As we mentioned in our first response, we have altered the title and the direction of the manuscript to show that fohn winds played a role in collapse, but did not trigger collapse. See above for a more detailed explanation.

Line 13: Saying that there are no studies examining surface melt prior to disintegrations is incorrect. You should revisit the Van Den Brooke, 2005 GRL paper that you cited that explicitly studies surface melt on the Larsen B prior to its collapse.

[Author Response] - These passages and others like it were meant to identify that little research fohn melt was done on time scales shorter than annual or seasonal, however, we see that the way

these comments are written make it seem like there is no research on föhn winds and surface melt. We have changed all passages in the manuscript to better frame this study among the rich array of studies on föhn winds and surface melt in the region. See the new introduction.

Line 17-19: This claim is based on a premise that foehn wind and surface temperatures remain within historical bounds. The Antarctic Peninsula already experiences large temperature variability and is projected to become warmer which would actually make the extant ice shelves more vulnerable to foehn winds in the future (Siegert et al., 2019; Chyhareva et al., 2019).
[Author Response] - We agree but wanted to make it clear that when we say extant ice shelves are less vulnerable to collapse than collapsed ice shelves because of large scale surface melt, that it is with the caveat that climate change is not considered here. We have changed the abstract (Line: 20-22) and the rest of the manuscript for clarification of our meaning.

Line 25-27: The claim of novelty seems unwarranted here. Plenty of studies already cited in this manuscript plus some others discuss fohn-related melt mechanisms on the Larsen B ice shelf (see Datta et al. 2019). Plus, Van den Brooke et al., 2005 claims surface melt accelerated the rate of ice shelf retreat, but did not claim it was a leading contributor to the final collapse
[Author Response] - We did not mean to suggest that there have been no studies that explore surface melt processes in the region, we only meant to identify small gaps in the research such as a lack of short time scale melt rates from fohn winds on Larsen A and B. We see that the current claim does not reflect this sentiment and have added a much deeper review of fohn win research in the AP region. See lines 39-82

Line 33-41: I don't understand why the manuscripts claims surface melt as the lead cause of the ice shelf final collapse in the previous paragraph and then point out all the other well documented processes that also affect ice shelf final collapse.
[Author Response] - There have been a few studies that point to large scale hydrofracture cascades caused by extensive melt ponds as a major factor that led to rapid collapse (Massom et al., 2018, Banwell et al., 2017), however, we understand your point and believe the sentence can be written better. We have completely changed the narrative of the manuscript and how our research fits within the plethora of previous research.

Line 30: This is a strange claim to make in the introduction. If this claim is valid, then it should first be proven in the results and then mentioned in the conclusion. (Changed this sentence to reflect work in previous study)
[Author Response] - Yes, we agree this statement feels strange. We changed the manuscript omitting this statement while also adding additional references.

Line 51: This is repeating a claim from the first paragraph that incorrectly states no previous research has been done on foehn-related melt around ice shelf collapses. This study may certainly give further detail on the intensity and spatial distribution of the foehn wind, but certainly is not the first.
[Author Response] - We agree as stated above, these comments and others like it have been changed.

Line 54: The temperature trends on the Antarctic Peninsula are a bit more complicated than this. Bozkurt et al., 2020, Carrasco et al., 2021, and Turner et al., 2016 paint a different picture where temperature trends are periodic and dependent on the location along the AP.
[Author Response] - **We have completely changed this statement in the new introduction and narrative reframing.**

Line 57: Questions 1 and 3 are very important and reasonable questions to address in this manuscript. Question 2 is much harder to answer with certainty without considering all the other processes (atmospheric and non-atmospheric) that could affect ice-shelf stability.
[Author Response] - **Thank you, we plan to alter Question 2 to fit more in line with the new direction of the manuscript. We have altered question 2 to say: " 2) Did föhn winds and associated melt play a role in triggering the collapses of the LAIS and LBIS?"**

Line 87: What height is the air temperature measured at?
[Author Response] - **10 Meters, which we have added to the manuscript.**

Line 95: It is stated again that is foehn detection method is the most accurate compared to previous work without explaining what this previous work is or why it is the most accurate. I also believe this is not the first foehn detection algorithm to incorporate station observations and model output (see Turton et al., 2018). The authors should include some information comparing the foehn detection of their algorithm against other foehn detection algorithms even if that data is presented in Laffin et al., 2021.
[Author Response] - **We did complete an intercomparison sensitivity study detailed in Laffin et al., 2021, comparing other identification methods (Datta et al., 2019, Cape et al., 2015). We Discuss this in the methods section (Line: 119), and we have also added the sensitivity study statistics table in the supplement.**

Line 108: Perhaps explain which variables you used to make the two-tailed t-test statistic. "Mean of both ice shelves" is vague
[Author Response] - **We have clarified which variables were used in our t-test. Line: 133**

Line 115-119: This seems like background information on the physics of foehn winds that would be better suited in the introduction section.
[Author Response] - **We agree and we have altered the introduction to include additional explanation and references of fohn winds in the Antarctic Peninsula region. Line: 57-82**

Line 131: This might be a personal preference, but you should change your figure numbers/order if you are referring to figure 5 before figure 3.
[Author Response] - **Yes, we recognize this only provides confusion. We have altered the figure numbers to occur chronologically.**

Line 132: You should present some results on foehn frequency before presenting the foehn related melt percentage. This would help put these melt-percentages in a better context.

[Author Response] - Yes, we agree and have altered Figure 2b to include foehn occurrence and fohn melt occurrence discussed in line: 163.

Line 137: If the SCAR inlet is not impacted by a foehn jet, where is the foehn wind influence coming from?

[Author Response] - SCAR inlet is not directly impacted by a föhn jet, but still experiences clear skies and weak föhn wind influence from the overall descending air that leads to warm winds and more importantly for this shelf, enhances shortwave radiation.

Line 139 – 142: You are contradicting yourself or at least unclear in these two sentences. First you claim that the disparity in foehn-related melt percentages among the ice shelves implicates the foehn as a contributor to the LAIS and LBIS collapse. This is a very strong assertion. It explains differences in melt rates on the ice shelves but saying this contributes to their collapse is a stretch. Then the next sentence is confusing and muddles your message about whether foehn is important or not to collapse. Probably easier to say that your results indicate foehn is one of many processes that weakened the LAIS and LBIS.

[Author Response] - We agree our assertion is unfounded and should state that fohn winds are one of the reasons the ice shelves destabilized. We have altered the narrative to place fohn winds in a more supporting role for collapse, rather than a trigger.

Line 149-152: If extensive foehn wind jets help explain why the LAIS and LBIB collapsed, then why have they not caused the collapse of the LCIS? Is there research showing that having melting at the terminus is essential for an ice shelf collapse? Discussed above in the beginning of paragraph

[Author Response] - We state in the manuscript *"LCIS on the other hand is impacted by four major jets and regularly experiences föhn-induced melt lakes, particularly in Cabinet inlet. However, the vast size of the LCIS does not allow the föhn-induced melt to reach the terminus. The föhn melt mechanism breaks down by mixing with cold air which reduces the intensity of the föhn jets from their peak at the base of the AP mountains to the calving front (Figure 3b)".* Massom et al., 2018 states that extensive melt ponds are an essential prerequisite for rapid collapse. With the change in direction of the manuscript we have fit our findings in the context of other research about collapse in a more clear manner.

Line 153-154: Previous literature already shows that foehn winds have a major impact on ice shelf surface melt and the framing of this sentence makes your results sound novel when in fact it would be more accurate to say that your results back up and enhance preexisting knowledge while citing these sources. (Find fohn melt research on LA and LB)

[Author Response] - We see how this statement makes our research sound novel. We have included a deeper reference pool for fohn winds in the region in the new introduction and references in the results section where we confirm previous work.

Line 181: It's a bit confusing to see the authors use satellite imagery from the 1992/1993 melt season as an analogue to the 1994/1995 melt season, but then later argue that despite the two seasons had similar amounts of foehn-related melt, the reason the Larsen A collapsed in 1995 and not in 1994/1995 was the timing of the surface melt. This argument needs more analysis of the background state of the Larsen A in 1992/1993 versus 1994/1995 to explain more clearly what was so special in 1994/1995.

[Author Response] - We do see how this argument may seem contradictory. With the new

direction of this manuscript, we discuss a lack of sea ice in 1995 that triggered collapse, while in 92/93 sea ice was present during most of the summer and so protected the calving front from collapse. See Lines: 304-314

Line 204-205: The total surface melt results are interesting, but would considering the size of the ice shelves change the perception of importance in regard to ice shelf destabilization? For instance, the Larsen C is much larger than the SCAR inlet ice shelf so total melt amounts would be difficult to compare. Melt per area would be a better metric.
[Author Response] - We already calculated for the mean melt over the entire ice shelf, but have clarified this in the manuscript. Line: 244

Line 212-214: The statement about the future resilience of the other ice shelves is problematic as it ignores potential future changes in foehn wind patterns. Especially since I believe your foehn wind detection algorithm only detects foehn winds when the temperature is above 0°C. There could be foehn events that currently do not push the temperature above this threshold which are not considered by your algorithm. But theoretically, if air temperatures rises along the Larsen C, then your algorithm would start detecting more foehn wind events. Deleted and discussed later in the conclusions and discussion.
[Author Response] - You are correct that our algorithm only identifies fohn winds above freezing and with climate change more southern locations will receive more fohn induced melt. Its hard to identify what that impact will be however it would be a great future direction of study. We have taken this statement out of the manuscript.

Line 227-228: The liquid-to-solid ratio (LSR) analysis here includes foehn-related melt and non foehn related melt. As mentioned earlier, it would be helpful to know the foehn wind frequency according to your detection algorithm in order to judge the significance of this result.
[Author Response] - Agreed, we added frequency stats see above. .
Line 244-245: There are likely many other differences between the Larsen A and B and the other ice shelves beyond foehn wind patterns. At the very least, sea-ice coverage and ocean forcings are different (see Massom et al., 2018). As I am not a glaciology expert, I cannot say for certain what the differences are structurally between these ice shelves, but it probably is wise to cite some papers regarding ice dynamics to verify this statement.
[Author Response] - We agree and have added ice shelf dynamics studies in the introduction. See above comments.

Line 270: I feel like you cannot conclude foehn-related surface melt triggered the Larsen A collapse without taking into consideration factors like basal melting.
[Author Response] - We agree which is why we have altered the narrative of the manuscript.

Line 282-283: How are you certain that a combination of factors also did not trigger the final disintegration of the Larsen A? In fact, in Massom et al., 2018, it was observed that sea-ice loss allowed ocean swells to apply a strain along the ice-shelf front which is cited as a possible trigger of the Larsen A collapse. This needs to be considered and discussed in this manuscript.
[Author Response] - We agree which is why we have altered the narrative of the manuscript indicating a supporting role of fohn winds for ice shelf collapse, and not trigger.

Line 289-290: This sentence disregards the gradual retreat of the ice shelves like the major retreat the Larsen A experienced in 1987 and 1989 mentioned in Scambos et al., 2000.
[Author Response] - We did research to see if we could triangulate other collapse events but could not find corroborating satellite images or in-person observations to clarify possible collapse mechanisms. .

Line 292-293: You cannot come to this conclusion if your foehn detection algorithm only detects foehn when the temperature is above 0°C which will likely occur more often over the Larsen C according to future climate projections (Siegert et al., 2019) (Chyhareva et al., 2019).
[Author Response] - We agree so we have altered our conclusions to included changes in the Southern Annular Mode (SAM) and climate change. Lines (351-361)

References:

Bozkurt, D., Bromwich, D. H., Carrasco, J., Hines, K. M., Maureira, J. C., and Rondanelli, R.:  Recent Near-surface Temperature Trends in the Antarctic Peninsula from Observed, Reanalysis  and Regional Climate Model Data, Adv. Atmos. Sci., 37, 477–493, https://doi.org/10.1007/s00376-020-9183-x, 2020.

Carrasco, J. F., Bozkurt, D., and Cordero, R. R.: A review of the observed air temperature in the Antarctic Peninsula. Did the warming trend come back after the early 21st hiatus?, 100653, https://doi.org/10.1016/j.polar.2021.100653, 2021.

Chyhareva, A., Krakovska, S., and Pishniak, D.: Climate projections over the Antarctic  Peninsula region to the end of the 21st century. Part 1: cold temperature indices, UAJ, 62–74, https://doi.org/10.33275/1727-7485.1(18).2019.131, 2019.

Datta, R. T., Tedesco, M., Fettweis, X., Agosta, C., Lhermitte, S., Lenaerts, J. T. M., and Wever,  N.: The Effect of Foehn-Induced Surface Melt on Firn Evolution Over the Northeast Antarctic  Peninsula, Geophysical Research Letters, 46, 3822–3831, https://doi.org/10.1029/2018GL080845, 2019.

Massom, R. A., Scambos, T. A., Bennetts, L. G., Reid, P., Squire, V. A., and Stammerjohn, S.  E.: Antarctic ice shelf disintegration triggered by sea ice loss and ocean swell, Nature, 558, 383– 389, https://doi.org/10.1038/s41586-018-0212-1, 2018.

Turner, J., Lu, H., White, I., King, J. C., Phillips, T., Hosking, J. S., Bracegirdle, T. J., Marshall,  G. J., Mulvaney, R., and Deb, P.: Absence of 21st century warming on Antarctic Peninsula  consistent with natural variability, Nature, 535, 411–415, https://doi.org/10.1038/nature18645,  2016.

Turton, J. V., Kirchgaessner, A., Ross, A. N., and King, J. C.: The spatial distribution and  temporal variability of föhn winds over the Larsen C ice shelf, Antarctica, Q.J.R. Meteorol. Soc.,  144, 1169–1178, https://doi.org/10.1002/qj.3284, 2018.

Scambos, T. A., Hulbe, C., Fahnestock, M., and Bohlander, J.: The link between climate warming and break-up of ice shelves in the Antarctic Peninsula, J. Glaciol., 46, 516–530, https://doi.org/10.3189/172756500781833043, 2000.

Siegert, M., Atkinson, A., Banwell, A., Brandon, M., Convey, P., Davies, B., Downie, R.,  Edwards, T., Hubbard, B., Marshall, G., Rogelj, J., Rumble, J., Stroeve, J., and Vaughan, D.: The  Antarctic Peninsula Under a 1.5°C Global Warming Scenario, Front. Environ. Sci., 7, 102, https://doi.org/10.3389/fenvs.2019.00102, 2019.

van den Broeke, M.: Strong surface melting preceded collapse of Antarctic Peninsula ice shelf, Geophys. Res. Lett., 32, L12815, https://doi.org/10.1029/2005GL023247, 2005.

---

## Referee Report (RR1)

I commend the authors for taking the time to improve the quality of the paper. The tone of the narrative is much more compelling and now fixes many of the previous issues of claiming novelty. The manuscript is nearly ready for publication and will surely be a great addition to the literature on ice-shelf instability, but I have one major comment and then some minor comments I would like to see addressed before publication.

Major comment

Abstract and Line 325-326: I was glad to see the authors include discussion on the Massom et al., 2018 paper which discussed how a lack of sea-ice cover allows swells to apply a strain to the ice-shelf fronts which happened during the collapses of the Larsen A and B. Although, I am not sure if the authors can claim the offshore foehn wind direction pushed sea ice away from the calving front without making a small analysis of this. While there are papers that discuss sea ice being influenced by large scale circulation patterns that create northwesterly flow across the Antarctic Peninsula (see Turner et al., 2002, Massom et al., 2006, Massom et al., 2008), to my knowledge there have been no studies that specifically connected foehn winds to sea ice conditions east of the Antarctic Peninsula. Likely the northwesterly direction of the foehn wind is the same as the northwesterly flow mentioned in previous studies, but perhaps it is only during these foehn periods when the sea ice is being blown of the ice shelf front. It could be a very interesting and insightful result if we see that the sea ice primarily responds to foehn winds and then this would really complete the argument for the importance of foehn winds on ice-shelf instability. Therefore, I recommend that if the authors want to connect the occurrence of foehn winds to sea-ice displacement, then they should analyze if there is a sea-ice response to periods of foehn winds or at least if there is a correlation between seasonal foehn wind occurrence and sea ice fraction. The sea ice fraction variable is readily available as an output in the ERA5 reanalysis.

Minor comments

Line 53: Fix parenthesis in citation.

Line 56: Figure 1b shows January 10, 1993, which was not during the collapse of the Larsen A or B. Please fix figure reference in text.

Line 105: Please elaborate more on the RACMO temperature biases? It is a little concerning to see the model has trouble simulating high temperature extremes when being used to study melting causes by high temperature extremes.

Line 118 – 121: I still don't understand how the authors verify that the FöhnDA is the most accurate detection method. Looking at Supplementary Table 1, it appears the false negative and false positive foehn detections are simply foehn detections that did not agree with the FöhnDA. These false positives and negatives should be classified as something like detected agreement or

disagreement. One way to test if a foehn event is "real" is to test with a different method such as an isentropic test (pressure levels showing a difference).

Line 151: Can the authors elaborate on how the variations in fohn jet location and wind strength explain why the SCAR inlet and Larsen C ice shelf are still intact? It seems like the difference in annual surface temperature is the main reason why these further south ice shelves are still intact as regions along the Larsen C receive frequent foehn winds. (Datta et al., 2019; Turton et al., 2018)

Line 163: The addition of figure 2b and the inclusion of the total melt hours percentage are very helpful and show that the foehn wind produces melt efficiently. It still would be good to know how frequent the foehn winds are as a percentage of all hours during the summer instead of just when melt is present.

Line 176: Replace with "large-scale"

Line 179: Previous studies have shown this to not be entirely true. Elvidge et al., 2016 explains the physical mechanisms for linear foehn winds which cause melt at the Larsen C ice shelf terminus and nonlinear foehn events which only cause melting by the base of the mountains. Still melting does occur more often at the base of the Larsen C mountains than at the ice shelf terminus. Please correct this sentence to account for the limited foehn melting that does occur at the ice shelf terminus.

Line 180: Good analysis, but the authors should cite Turton et al., 2018 which shows that foehn is detected more frequently at the base of the Larsen C mountains (see Cole Peninsula AWS vs AWS2).

Line 194: Should also cite Rott et al., 1996 here as that study first discusses the northwesterly flow that pushed the sea ice from the ice-shelf front

Line 226-228: The authors should cite Massom et al., 2018 and Turner et al., 2002 here as they discuss the lack of sea-ice cover and the atmospheric circulation patterns that caused the off-coast wind. Also going back to the major comment, this could be a good place to show if foehn winds progressively pushed the sea ice away from the ice-shelf front during the austral spring and summer.

Line 257-258: It could be helpful to reference Figure 6b at the end of the sentence regarding the LSR without foehn-related melt.

Line 275 – 277: I still do not understand why the temperature gradient alone does not explain the surface melt difference between ice shelves. The temperature gradient should also explain why foehn winds on the intact ice shelves do not produce large amounts of surface melt right? Unless the authors can explain that the foehn winds cause a different temperature change response between the collapsed and intact ice shelves. In fact from looking at Figure 5e and 5f, the

temperature difference between foehn and non-foehn is slightly larger on the Larsen C than the Larsen A and B.

Line 341-342: Again regarding the sea ice, this is something not really proven in the manuscript. An analysis of the foehn wind/sea ice extent relationship would be helpful and make the manuscript even more compelling.

Works Referenced

Datta, R. T., Tedesco, M., Fettweis, X., Agosta, C., Lhermitte, S., Lenaerts, J. T. M., and Wever, N.: The Effect of Foehn-Induced Surface Melt on Firn Evolution Over the Northeast Antarctic Peninsula, Geophysical Research Letters, 46, 3822–3831, https://doi.org/10.1029/2018GL080845, 2019.

Elvidge, A. D., Renfrew, I. A., King, J. C., Orr, A., and Lachlan-Cope, T. A.: Foehn warming distributions in nonlinear and linear flow regimes: a focus on the Antarctic Peninsula: Foehn Warming Distributions in Nonlinear and Linear Flow Regimes, Q.J.R. Meteorol. Soc., 142, 618–631, https://doi.org/10.1002/qj.2489, 2016.
Massom, R. A., Stammerjohn, S. E., Smith, R. C., Pook, M. J., Iannuzzi, R. A., Adams, N., Martinson, D. G., Vernet, M., Fraser, W. R., Quetin, L. B., Ross, R. M., Massom, Y., and Krouse, H. R.: Extreme Anomalous Atmospheric Circulation in the West Antarctic Peninsula Region in Austral Spring and Summer 2001/02, and Its Profound Impact on Sea Ice and Biota*, 19, 3544–3571, https://doi.org/10.1175/JCLI3805.1, 2006.

Massom, R. A., Stammerjohn, S. E., Lefebvre, W., Harangozo, S. A., Adams, N., Scambos, T. A., Pook, M. J., and Fowler, C.: West Antarctic Peninsula sea ice in 2005: Extreme ice compaction and ice edge retreat due to strong anomaly with respect to climate, J. Geophys. Res., 113, C02S20, https://doi.org/10.1029/2007JC004239, 2008.

Rott, H., Skvarca, P., and Nagler, T.: Rapid Collapse of Northern Larsen Ice Shelf, Antarctica, Science, 271, 788–792, https://doi.org/10.1126/science.271.5250.788, 1996.

Turner, J., Harangozo, S. A., Marshall, G. J., King, J. C., and Colwell, S. R.: Anomalous atmospheric circulation over the Weddell Sea, Antarctica during the Austral summer of 2001/02 resulting in extreme sea ice conditions: ANOMALOUS ATMOSPHERIC CIRCULATION OVER THE WEDDELL SEA, Geophys. Res. Lett., 29, 13-1-13–4, https://doi.org/10.1029/2002GL015565, 2002.

Turton, J. V., Kirchgaessner, A., Ross, A. N., and King, J. C.: The spatial distribution and temporal variability of föhn winds over the Larsen C ice shelf, Antarctica, Q.J.R. Meteorol. Soc., 144, 1169–1178, https://doi.org/10.1002/qj.3284, 2018.

---

## Referee Report (RR2)

**Review for *"The role of föhn winds in Antarctic Peninsula rapid ice shelf collapse"* by Laffin et al.**

General comments

I am pleased to see that the authors have taken on board the majority of my previous comments, which I thank them for. The authors now also have a much more complete reference list, and in general, the paper is written more clearly, though there are still some vague sentences that I comment on in my line by line comments below. Before those general comments, I have summarized some additional general comments below.

On many occasions of the paper a 'critical stability depth' of lakes is now referred too, e.g. in the Abstract (line 16) and in the introduction (line 46). The authors also often state that critical stability lake depth is 1 m (though on one occasion they state that this depth is 3.5 m). The papers by Glasser and Scambos (2008) and Banwell et al. (2013, 2014) are often referred too after these statements, but none of those papers actually talk about a critical stability depth for hydrofracture. What Banwell et al (2014) *do state* is that the average depth of lakes on Larsen B in the Landsat image prior to ice shelf break up is 1 m, however they do not suggest that is a depth threshold for break up. Glasser and Scambos (2008) do not mention any specific lake depth threshold for breakup, and nor do Banwell et al (2013; in fact, in that study, they model lakes under the assumption they are all 5 m deep). As it is the volume of water in a lake that determines the 'load' actually on the ice shelf surface, not the water depth, I do not think that the authors this paper under review should talk about critical depth threshold, especially as the 1 m 'threshold' has not been suggested to be a threshold in the literature previously. Some sort of lake volume threshold may exist, perhaps combined with a lake density threshold, and those may help to determine when an ice shelf is primed for rapid break up via chain reaction lake drainage, but I am not aware of a paper that has specifically studied this.

In general, I think it is too speculative to suggest that the "extant ice shelves are less likely to experience rapid collapse due to föhn-driven melt so long as surface temperatures and föhn occurrence remain within historical bounds" (from the final sentence of the abstract). Such as idea could be discussed as part of the Discussion, but personally. I do not think there is need to include this idea in the abstract. It seems fair that this suggestion may be true on the basis that föhn wind occurrence is less common on these ice shelves (I think Larsen C and the Scar Inlet, but please see my comment below) as the authors discuss, but on occasions the authors state that lakes do not form on these ice shelves, whereas in fact lakes often do form on Larsen C, where huge impermeable ice lenses have also been found (see Hubbard et al 2016).

Related to the above comment, the authors often mention 'extant ice shelves' (lines 19 and again on line 20 in the abstract), and 'remaining ice shelves' (e.g. line 354) on multiple occasions through the revised manuscript. Such statements are vague, and could be referring to *all* remaining Antarctic ice shelves, or just those remaining on the AP, or just those in Eastern AP. In fact, in such statements, I believe the authors are just referring to the Larsen C and Scar Inlet. So they should either state these two ice shelf names each time they are discussed, or at least, state 'East AP ice shelves' on these occasions. Related to this point, I also wonder if the paper title should include 'east' or 'eastern', given it is only the eastern AP ice shelves that are studied in this paper.

Finally, I don't know of any evidence (observational nor modelling) that suggests Larsen A experienced rapid chain reaction style break up, i.e. like Larsen B did (Banwell et al 2013). So the authors need to remove all references to this process having happened on Larsen A.

Line by line comments

8 – 11: Unlike Larsen B,s I do not know of any evidence that suggests that Larsen A also experience cascading hydrofracture events. Yes there may have been, but there is no evidence, so this sentence needs rewording.

10: I believe this should be 'long' period ocean swell, rather than 'large' period. E.g. see Massom et al (2018). Check throughout paper.

11: "During collapse, surface observations indicate föhn winds were present on both ice shelves" – this is very vague. What kind of surface observations are you referring too? Also, for Larsen B at least, observations in the form of optical satellite imagery were very sporadic (maybe just 3 Landsat images in a 2 month period?).

17 – 19: Be specific with what 'extant ice shelves' you are referring too.

19 – 21: As I state above, I think this sentence may be too speculative to include in the abstract.

23: Be more specific; which 'ice shelves' disintegrated? Also, it seems odd to start the paper with this conclusive statement (lines 23 – 24) saying that disintegrations were caused by regional warming trends, if you then go on to argue in this paper that föhn winds also played a role! Perhaps this sentence should be removed from this location and stated/discussed elsewhere?

28 – 32: Again, I do not think there was evidence that chain reaction lake drainage was observed (or modeled) prior to Larsen A's collapse, so this sentence needs rewording. I also suggest breaking up the long list of references after the word 'hydrofracture', as not all of these references are related to hydrofracture. Suggest moving some of these references to earlier in the sentence after 'melt pond flooding'. Glasser and Scambos (2008) should also be mentioned after 'melt pond flooding'. And Banwell et al (2013) should be added to the list after 'hydrofracture'.

41 – 42: 'In addition Massom et al., (2018) concluded that a lack of summer sea ice allowed large period ocean swells to reach the ice shelf calving front.' - It's unclear why this sentence is mentioned separately as surely it is part of (2) in the previous sentence. But in any case, this whole section from lines 38 – 44 seems very repetitive given basically the same detail in is included in the previous paragraph (lines 30 – 37). So I suggest delete much of this section.

46/47: Again, as mentioned above, I think the authors should reconsider their suggestion that 1 m is a critical depth threshold for rapid ice shelf break up.

48/48: This definition of hydrofracture is useful, but ideally it should come when 'hydrofracture' is first mentioned, which is currently on line 32 (I think).

51: Suggest replacing ''at critical water depths' with 'that rapidly drain by hydrofracture'.

67: I am unclear why just 'late season fohn melt reduces firn pore space'. Surely this process has the same effect on the fir at any time during the melt season?

78: Replace 'does' with 'did'.

83: Further to my general comment above, I Suggest replacing 'each ice shelf' with 'each eastern AP ice shelf'. (Assuming you are referring to all eastern AP shelves here, and not just LAIS and LBIS? Maybe state the specific shelves in brackets?)

86: Again, please clarify what 'extant ice shelves' are being referred too.

126/127: In the second part of the sentence: 'Therefore, we consider RACMO2 simulated estimates of surface melt caused by föhn winds to be conservative and likely higher in regions where föhn winds are funneled and concentrated', it sounds as though it is being suggested that modelled fohn winds will be higher than in reality. But based on the fact you also say modeled estimates are conservative, I think you may mean to say the opposite?

144: Is the ML algorithm being referred to in this sentence fohnDA? State that if so.

175 – 188: Another reason for Scar inlet's stability may be that it has had lots of sea ice buttressed up again it until very recently (when it broke up); you could also mention this.

194 - 196: I suggest putting 'e.g.' in front of Massom et al (2018) seeing as this study did not focus on the LAIS. And as I state above, I do not think any study has suggested that chain reaction lake drainage (aka hydrofracture cascades) contributed to Larsen A's collapse.

218: As I also mentioned in my last review, Banwell et al (2014) should be referenced after the following sentence: 'We find mean melt lake depth to be between 1.38-6.86 meters depending on lake location and föhn influence, which exceeds the average lake depth of the LBIS lakes prior to collapse (1 meter)'.

219: Where did a critical depth of 3.5 m come from? Earlier the authors said this was 1 m. But in any case, as I explain in my General Comments, I don't think that talking about a critical lake depth is useful anyway.

238: State what ice shelves you are referring too by 'all major ice shelves'.

252/253: Again, very vague; clarify which ice shelves are being talked about.

253 – 255: I suggest moving the description of the firn densification process to the introduction.

252 – 261: Somewhere it would be useful to mention that there is some evidence of firn densification, surface ponding, and expansive ice layers on Larsen C. See Hubbard et al (2016).

290/291: Similar to my earlier comment, I am unsure why just late season melt is being talked about here.

303: Which ice shelf is being referred to here?

301: Comment on why this additional snow is relevant.

314: I think you mean 'basal melting'

317: Again, I don't know why the authors talk about a 'critical melt lake depth of stability' (it is not in the Banwell et al 2013 reference given here).

329: Which 'extant ice shelves' are being referred too? (FYI: Some southwest AP ice shelves, specifically George VI and Wilkins have had lots of melt and surface ponding in recent years. E.g. see Banwell et al 2021).

341: Again, remove mention of critical 1 m lake depth.

358: Clarify that AP ice shelves are beginning referred to here.

360: what 'region'? And surely fohn winds should be mentioned in this sentence too? What about future fohn winds?

Figures

Figure 1: Clarify that the LAIS and LBIS in the figure are no longer present, and perhaps give their collapse dates in the first sentence of the caption.

Figure 3: In the caption, I think 'graph' would be a more appropriate word than 'curve'.

Figure 6: What time period are the data for?

Additional references

Banwell, A. F., Datta, R. T., Dell, R. L., Moussavi, M., Brucker, L., Picard, G., Shuman, C. A., and Stevens, L. A. The 32-year record-high surface melt in 2019/2020 on the northern George VI Ice Shelf, Antarctic Peninsula, The Cryosphere, 15, 909–925, https://doi.org/10.5194/tc-15-909-2021, 2021.

Hubbard, B. et al. Massive subsurface ice formed by refreezing of ice-shelf melt ponds. Nat. Commun. 7:11897 doi: 10.1038/ncomms11897
(2016).

---

## Author Response (AR2)

**List of all relevant changes**

1. Added a small correlation study between fohn occurrence and sea ice concentration.

**Point by Point Response to Reviewers**

**RC2-**
**Thank you for your comments and suggestions. We believe this manuscript has improved significantly with your suggestions and we sincerely appreciate your valuable contributions. We have addressed your comments below marked with [Author Response].**

I commend the authors for taking the time to improve the quality of the paper. The tone of the narrative is much more compelling and now fixes many of the previous issues of claiming novelty. The manuscript is nearly ready for publication and will surely be a great addition to the literature on ice-shelf instability, but I have one major comment and then some minor comments I would like to see addressed before publication.

Major comment

Abstract and Line 325-326: I was glad to see the authors include discussion on the Massom et al., 2018 paper which discussed how a lack of sea-ice cover allows swells to apply a strain to the ice-shelf fronts which happened during the collapses of the Larsen A and B. Although, I am not sure if the authors can claim the offshore foehn wind direction pushed sea ice away from the calving front without making a small analysis of this. While there are papers that discuss sea ice being influenced by large scale circulation patterns that create northwesterly flow across the Antarctic Peninsula (see Turner et al., 2002, Massom et al., 2006, Massom et al., 2008), to my knowledge there have been no studies that specifically connected foehn winds to sea ice conditions east of the Antarctic Peninsula. Likely the northwesterly direction of the foehn wind is the same as the northwesterly flow mentioned in previous studies, but perhaps it is only during these foehn periods when the sea ice is being blown off the ice shelf front. It could be a very interesting and insightful result if we see that the sea ice primarily responds to foehn winds and then this would really complete the argument for the importance of foehn winds on ice-shelf instability. Therefore, I recommend that if the authors want to connect the occurrence of foehn winds to sea-ice displacement, then they should analyze if there is a sea-ice response to periods of foehn winds or at least if there is a correlation between seasonal foehn wind occurrence and sea ice fraction. The sea ice fraction variable is readily available as an output in the ERA5 reanalysis.
**[Author Response] This is a very insightful suggestion. We have added a small fohn occurrence and sea ice concentration study summarized in a new figure (Figure 5). It is clear that there is a negative correlation between fohn occurrence and sea ice concentration, both in the long term but also on short time scales (hours to days) (Lines 211-217)**

Minor comments

Line 53: Fix parenthesis in citation.

[Author Response] We have corrected the manuscript.

Line 56: Figure 1b shows January 10, 1993, which was not during the collapse of the Larsen A or B. Please fix figure reference in text.
[Author Response] Thank you for identifying this oversight. We have altered the manuscript to exclude Figure 1b.

Line 105: Please elaborate more on the RACMO temperature biases? It is a little concerning to see the model has trouble simulating high temperature extremes when being used to study melting causes by high temperature extremes.
[Author Response] We did a little deeper research and found that RACMO has a slight warm bias likely due to model resolution, as well as under/over-estimates shortwave/longwave radiation because of how clouds and moisture are simulated. This may also be why temperature is slightly warmer. We have added these comments and references into the manuscript.

Line 118 – 121: I still don't understand how the authors verify that the FöhnDA is the most accurate detection method. Looking at Supplementary Table 1, it appears the false negative and false positive foehn detections are simply foehn detections that did not agree with the FöhnDA. These false positives and negatives should be classified as something like detected agreement or disagreement. One way to test if a foehn event is "real" is to test with a different method such as an isentropic test (pressure levels showing a difference).
[Author Response] Thank you for this comment. It was not clear that we compared two other fohn identification studies to AWS observed and our ML algorithm. I have made this more clear in the table notes.

Line 151: Can the authors elaborate on how the variations in fohn jet location and wind strength explain why the SCAR inlet and Larsen C ice shelf are still intact? It seems like the difference in annual surface temperature is the main reason why these further south ice shelves are still intact as regions along the Larsen C receive frequent foehn winds. (Datta et al., 2019; Turton et al., 2018)
[Author Response] Yes, temperature gradient does drive fohn melt strength, however the fact that there is no fohn jet on SCAR inlet may be why it is still intact. We see how the wording in the manuscript is not clear and have altered it for clarification.

Line 163: The addition of figure 2b and the inclusion of the total melt hours percentage are very helpful and show that the foehn wind produces melt efficiently. It still would be good to know how frequent the foehn winds are as a percentage of all hours during the summer instead of just when melt is present.
[Author Response] We have altered Figure 2b to include the percent of all hours that fohn winds occur. Thank you for the suggestion.

Line 176: Replace with "large-scale"
[Author Response] Thank you, the manuscript has been altered.

Line 179: Previous studies have shown this to not be entirely true. Elvidge et al., 2016

explains the physical mechanisms for linear foehn winds which cause melt at the Larsen C ice shelf terminus and nonlinear foehn events which only cause melting by the base of the mountains. Still melting does occur more often at the base of the Larsen C mountains than at the ice shelf terminus. Please correct this sentence to account for the limited foehn melting that does occur at the ice shelf terminus.

[Author Response] **This is a good point. We have altered the manuscript to say "However, the vast size of the LCIS limits the amount of föhn-induced melt at the terminus." and added reference.**

Line 180: Good analysis, but the authors should cite Turton et al., 2018 which shows that foehn is detected more frequently at the base of the Larsen C mountains (see Cole Peninsula AWS vs AWS2).

[Author Response] **Another good point. We have added the citation to the manuscript.**

Line 194: Should also cite Rott et al., 1996 here as that study first discusses the northwesterly flow that pushed the sea ice from the ice-shelf front

[Author Response] **The reference has been added to the manuscript.**

Line 226-228: The authors should cite Massom et al., 2018 and Turner et al., 2002 here as they discuss the lack of sea-ice cover and the atmospheric circulation patterns that caused the offcoast wind. Also going back to the major comment, this could be a good place to show if foehn winds progressively pushed the sea ice away from the ice-shelf front during the austral spring and summer.

[Author Response] **We have added this reference and added reference to figure 5.**

Line 257-258: It could be helpful to reference Figure 6b at the end of the sentence regarding the LSR without foehn-related melt.

[Author Response] **We have added the reference to Figure 6b.**

Line 275 – 277: I still do not understand why the temperature gradient alone does not explain the surface melt difference between ice shelves. The temperature gradient should also explain why foehn winds on the intact ice shelves do not produce large amounts of surface melt right? Unless the authors can explain that the foehn winds cause a different temperature change response between the collapsed and intact ice shelves. In fact from looking at Figure 5e and 5f, the temperature difference between foehn and non-foehn is slightly larger on the Larsen C than the Larsen A and B.

[Author Response] **Yes, this is a good point. Temperature is a driver of the strength of fohn winds and their ability to melt. Areas that are colder will not experience as much melt. We have added a sentence that discusses this idea in the discussion section ("Temperature gradient however, could explain why fohn wind events cause less melt on more southern ice shelves and may cause super melt events on collapsed ice shelves because temperature is already elevated on more northern ice shelves prior to the effect fohn has on temperature.")**

Line 341-342: Again regarding the sea ice, this is something not really proven in the manuscript. An analysis of the foehn wind/sea ice extent relationship would be helpful and make the manuscript even more compelling.

[Author Response] **We have added a sea ice concentration study as you suggested. Thank you!**

Works Referenced

Datta, R. T., Tedesco, M., Fettweis, X., Agosta, C., Lhermitte, S., Lenaerts, J. T. M., and Wever, N.: The Effect of Foehn-Induced Surface Melt on Firn Evolution Over the Northeast Antarctic Peninsula, Geophysical Research Letters, 46, 3822–3831, https://doi.org/10.1029/2018GL080845, 2019.

Elvidge, A. D., Renfrew, I. A., King, J. C., Orr, A., and Lachlan-Cope, T. A.: Foehn warming distributions in nonlinear and linear flow regimes: a focus on the Antarctic Peninsula: Foehn Warming Distributions in Nonlinear and Linear Flow Regimes, Q.J.R. Meteorol. Soc., 142, 618–631, https://doi.org/10.1002/qj.2489, 2016.

Massom, R. A., Stammerjohn, S. E., Smith, R. C., Pook, M. J., Iannuzzi, R. A., Adams, N., Martinson, D. G., Vernet, M., Fraser, W. R., Quetin, L. B., Ross, R. M., Massom, Y., and Krouse, H. R.: Extreme Anomalous Atmospheric Circulation in the West Antarctic Peninsula Region in Austral Spring and Summer 2001/02, and Its Profound Impact on Sea Ice and Biota*, 19, 3544–3571, https://doi.org/10.1175/JCLI3805.1, 2006.

Massom, R. A., Stammerjohn, S. E., Lefebvre, W., Harangozo, S. A., Adams, N., Scambos, T. A., Pook, M. J., and Fowler, C.: West Antarctic Peninsula sea ice in 2005: Extreme ice compaction and ice edge retreat due to strong anomaly with respect to climate, J. Geophys. Res., 113, C02S20, https://doi.org/10.1029/2007JC004239, 2008. Rott, H., Skvarca, P., and Nagler, T.: Rapid Collapse of Northern Larsen Ice Shelf, Antarctica, Science, 271, 788–792, https://doi.org/10.1126/science.271.5250.788, 1996.

Turner, J., Harangozo, S. A., Marshall, G. J., King, J. C., and Colwell, S. R.: Anomalous atmospheric circulation over the Weddell Sea, Antarctica during the Austral summer of 2001/02 resulting in extreme sea ice conditions: ANOMALOUS ATMOSPHERIC CIRCULATION OVER THE WEDDELL SEA, Geophys. Res. Lett., 29, 13-1-13–4, https://doi.org/10.1029/2002GL015565, 2002.

Turton, J. V., Kirchgaessner, A., Ross, A. N., and King, J. C.: The spatial distribution and temporal variability of föhn winds over the Larsen C ice shelf, Antarctica, Q.J.R. Meteorol. Soc., 144, 1169–1178, https://doi.org/10.1002/qj.3284, 2018.

**RC1-**

Thank you for your comments and suggestions. We believe this manuscript will improve significantly with your suggestions and we sincerely appreciate your valuable contributions. We have addressed your comments below marked with [Author Response].

**Review for "The role of föhn winds in Antarctic Peninsula rapid ice shelf collapse" by Laffin et al.**

General comments

I am pleased to see that the authors have taken on board the majority of my previous comments, which I thank them for. The authors now also have a much more complete reference list, and in general, the paper is written more clearly, though there are still some vague sentences that I comment on in my line by line comments below. Before those general comments, I have summarized some additional general comments below.

On many occasions of the paper a 'critical stability depth' of lakes is now referred too, e.g. in the Abstract (line 16) and in the introduction (line 46). The authors also often state that critical stability lake depth is 1 m (though on one occasion they state that this depth is 3.5 m). The papers by Glasser and Scambos (2008) and Banwell et al. (2013, 2014) are often referred too after these statements, but none of those papers actually talk about a critical stability depth for hydrofracture. What Banwell et al (2014) do state is that the average depth of lakes on Larsen B in the Landsat image prior to ice shelf break up is 1 m, however they do not suggest that is a depth threshold for break up. Glasser and Scambos (2008) do not mention any specific lake depth threshold for breakup, and nor do Banwell et al (2013; in fact, in that study, they model lakes under the assumption they are all 5 m deep). As it is the volume of water in a lake that determines the 'load' actually on the ice shelf surface, not the water depth, I do not think that the authors this paper under review should talk about critical depth threshold, especially as the 1 m 'threshold' has not been suggested to be a threshold in the literature previously. Some sort of lake volume threshold may exist, perhaps combined with a lake density threshold, and those may help to determine when an ice shelf is primed for rapid break up via chain reaction lake drainage, but I am not aware of a paper that has specifically studied this.

[Author Response] We agree and never want to miss represent other authors' findings and research. We have altered the manuscript in the abstract, introduction and conclusion to better clarify and represent previous research. We no longer state that 1m or 5m lake depths are critical lake depths, only that they are observed or modeled lake depths.

In general, I think it is too speculative to suggest that the "extant ice shelves are less likely to experience rapid collapse due to föhn-driven melt so long as surface temperatures and föhn occurrence remain within historical bounds" (from the final sentence of the abstract). Such an idea could be discussed as part of the Discussion, but personally. I do not think there is a need to include this idea in the abstract. It seems fair that this suggestion may be true on the basis that föhn wind occurrence is less common on these ice shelves (I think Larsen C and the Scar Inlet, but please see my comment below) as the authors discuss, but on occasions the authors state that lakes do not form on these ice shelves, whereas in fact lakes often do form on Larsen C, where huge impermeable ice lenses have also been found (see Hubbard et al 2016).

[Author Response] We agree that our comments may be to speculative, however we still feel that this

a reasonable result that should be stated in the abstract. We did however change the sentence to; "extant ice shelves may be less likely to experience rapid collapse due to föhn-driven melt so long as surface temperatures and föhn occurrence remain within historical bounds."

Related to the above comment, the authors often mention 'extant ice shelves' (lines 19 and again on line 20 in the abstract), and 'remaining ice shelves' (e.g. line 354) on multiple occasions through the revised manuscript. Such statements are vague, and could be referring to all remaining Antarctic ice shelves, or just those remaining on the AP, or just those in Eastern AP. In fact, in such statements, I believe the authors are just referring to the Larsen C and Scar Inlet. So they should either state these two ice shelf names each time they are discussed, or at least, state 'East AP ice shelves' on these occasions. Related to this point, I also wonder if the paper title should include 'east' or 'eastern', given it is only the eastern AP ice shelves that are studied in this paper.
[Author Response] We agree clarifying which ice shelves we mean by "extant" is important. We have clarified our meaning throughout the manuscript by using "SCAR inlet and the LCIS" or "extant ice shelf on the eastern AP."

Finally, I don't know of any evidence (observational nor modeling) that suggests Larsen A experienced rapid chain reaction style break up, i.e. like Larsen B did (Banwell et al 2013). So the authors need to remove all references to this process having happened on Larsen A.
[Author Response] We agree we do not want to speculate how Larsen A collapsed, so we have changed all language that suggests it collapsed through hydrofracture chain reaction break up.

Line by line comments

8 – 11: Unlike Larsen B,s I do not know of any evidence that suggests that Larsen A also experience cascading hydrofracture events. Yes there may have been, but there is no evidence, so this sentence needs rewording.
[Author Response] We agree we do not want to speculate how Larsen A collapsed, so we have changed all language that suggests it collapsed through hydrofracture chain reaction break up.

10: I believe this should be 'long' period ocean swell, rather than 'large' period. E.g. see Massom et al (2018). Check throughout the paper.
[Author Response] We have altered the manuscript to say long period ocean swells. Thank you for identifying this oversight.

11: "During collapse, surface observations indicate föhn winds were present on both ice shelves" – this is very vague. What kind of surface observations are you referring too? Also, for Larsen B at least, observations in the form of optical satellite imagery were very sporadic (maybe just 3 Landsat images in a 2 month period?).
[Author Response] We have clarified our meaning of observations with field observations and satellite observations.

17 – 19: Be specific with what 'extant ice shelves' you are referring too.
[Author Response] We have clarified our meaning throughout the manuscript by using "SCAR inlet and the LCIS" or "extant ice shelf on the eastern AP."

19 – 21: As I state above, I think this sentence may be too speculative to include in the abstract.
[Author Response] We agree that our comments may be to speculative, however we still feel that this a reasonable result that should be stated in the abstract. We did however change the sentence to; "extant ice shelves may be less likely to experience rapid collapse due to föhn-driven melt so long as surface temperatures and föhn occurrence remain within historical bounds."

23: Be more specific; which 'ice shelves' disintegrated? Also, it seems odd to start the paper with this conclusive statement (lines 23 – 24) saying that disintegrations were caused by regional warming trends, if you then go on to argue in this paper that föhn winds also played a role! Perhaps this sentence should be removed from this location and stated/discussed elsewhere?
[Author Response] We agree that we do not want to contradict our findings with this sentence so we have taken this out of the manuscript.

28 – 32: Again, I do not think there was evidence that chain reaction lake drainage was observed (or modeled) prior to Larsen A's collapse, so this sentence needs rewording. I also suggest breaking up the long list of references after the word 'hydrofracture', as not all of these references are related to hydrofracture. Suggest moving some of these references to earlier in the sentence after 'melt pond flooding'. Glasser and Scambos (2008) should also be mentioned after 'melt pond flooding'. And Banwell et al (2013) should be added to the list after 'hydrofracture'.
[Author Response] We altered the sentence to not suggest the LAIS was observed to collapse through chain reaction lake drainage. We also altered the location of references and added those you suggested. Thank you.

41 – 42: 'In addition Massom et al., (2018) concluded that a lack of summer sea ice allowed large period ocean swells to reach the ice shelf calving front.' - It's unclear why this sentence is mentioned separately as surely it is part of (2) in the previous sentence. But in any case, this whole section from lines 38 – 44 seems very repetitive given basically the same detail in is included in the previous paragraph (lines 30 – 37). So I suggest delete much of this section.
[Author Response] We have taken out line 41-42 as suggested, because it was repetitive. We thank you for your suggestion about deleting likes 38-44 because of their repetitive nature, but we feel that those lines, other than those already deleted, provide a good background of the "4 essential prerequisites for rapid collapse" theorized by Massom et al., that are not covered in lines 30-37.

46/47: Again, as mentioned above, I think the authors should reconsider their suggestion that 1 m is a critical depth threshold for rapid ice shelf break up. 48/48: This definition of hydrofracture is useful, but ideally it should come when 'hydrofracture' is first mentioned, which is currently on line 32 (I think).
[Author Response] Thank you, we agree, see response above.

51: Suggest replacing ''at critical water depths' with 'that rapidly drain by hydrofracture'.
[Author Response] This is an excellent and clarifying alteration. We have incorporated it into the manuscript.

67: I am unclear why just 'late season fohn melt reduces firn pore space'. Surely this process has the

same effect on the fir at any time during the melt season?

[Author Response] Our understanding is that late season fohn melt does not stay liquid for long, it will freeze overnight filling in surface pore space, whereas early and mid summer melt can stay liquid for longer, leading to firn percolation, runoff, evaporation, drainage etc.

78: Replace 'does' with 'did'.
[Author Response] We have altered the manuscript.

83: Further to my general comment above, I Suggest replacing 'each ice shelf' with 'each eastern AP ice shelf'. (Assuming you are referring to all eastern AP shelves here, and not just LAIS and LBIS? Maybe state the specific shelves in brackets?) 86: Again, please clarify what 'extant ice shelves' are being referred too.
[Author Response] We have altered the manuscript, see note above.

126/127: In the second part of the sentence: 'Therefore, we consider RACMO2 simulated estimates of surface melt caused by föhn winds to be conservative and likely higher in regions where föhn winds are funneled and concentrated', it sounds as though it is being suggested that modeled fohn winds will be higher than in reality. But based on the fact you also say modeled estimates are conservative, I think you may mean to say the opposite?
[Author Response] We mean to say that fohn winds smaller than the model simulations are likely to exist, but because model resolution is not simulated and therefore melt is not calculated but is likely greater under their influence. We have altered the manuscript to clarify this point.

144: Is the ML algorithm being referred to in this sentence fohnDA? State that if so.
[Author Response] We have added reference to the ml algorithm (FöhnDA)

175 – 188: Another reason for Scar inlet's stability may be that it has had lots of sea ice buttressed up again it until very recently (when it broke up); you could also mention this.
[Author Response] We have added reference to decreased buttress force from sea ice.

194 - 196: I suggest putting 'e.g.' in front of Massom et al (2018) seeing as this study did not focus on the LAIS. And as I state above, I do not think any study has suggested that chain reaction lake drainage (aka hydrofracture cascades) contributed to Larsen A's collapse.
[Author Response] We have altered the manuscript to better represent previous research and understanding of both ice shelf collapse events, that does not include linking LAIS collapse with chain reaction lake drainage.

218: As I also mentioned in my last review, Banwell et al (2014) should be referenced after the following sentence: 'We find mean melt lake depth to be between 1.38-6.86 meters depending on lake location and föhn influence, which exceeds the average lake depth of the LBIS lakes prior to collapse (1 meter)'.
[Author Response] We have added reference to this paper and apologize for this oversight from the previous review.

219: Where did a critical depth of 3.5 m come from? Earlier the authors said this was 1 m. But in any case, as I explain in my General Comments, I don't think that talking about a critical lake depth

is useful anyway.

[Author Response] Thank you for identifying this oversight, we have altered the sentence and reference to include a simulated 5m lake depth.

238: State what ice shelves you are referring too by 'all major ice shelves'.

[Author Response] We have included a reference to all easter AP ice shelves in parentheses.

252/253: Again, very vague; clarify which ice shelves are being talked about.

[Author Response] We agree and have added…"Our analysis of firn density or available firn pore space identifies significant differences in ice shelves that have collapsed (LAIS, LBIS) and those that remain intact (SCAR inlet, LCIS)"

253 – 255: I suggest moving the description of the firn densification process to the introduction.

[Author Response] We have altered the manuscript.

252 – 261: Somewhere it would be useful to mention that there is some evidence of firn densification, surface ponding, and expansive ice layers on Larsen C. See Hubbard et al (2016).

[Author Response] We have added a sentence to discuss this point. "It is important to note that there is evidence that the LCIS experiences regions of firn densification through melt processes, however these regions are mostly focused close to the AP mountains, likely formed from the location of fohn jets."

290/291: Similar to my earlier comment, I am unsure why just late season melt is being talked about here.

[Author Response] Our understanding is that late season fohn melt does not stay liquid for long, it will freeze overnight filling in surface pore space, whereas early and mid summer melt can stay liquid for longer, leading to firn percolation, runoff, evaporation, drainage etc.

303: Which ice shelf is being referred to here?

[Author Response] We have clarified which ice shelves we mean.

301: Comment on why this additional snow is relevant.

[Author Response] The additional snow mass leads to a thicker firn layer that can store more water before melt lakes form. Additionally this added additional stress to the ic shelf.

314: I think you mean 'basal melting'

[Author Response] We did mean basal melting and have made that change. Thank you.

317: Again, I don't know why the authors talk about a 'critical melt lake depth of stability' (it is not in the Banwell et al 2013 reference given here).

[Author Response] We have altered this language to better represent previous work and no longer use "critical melt lake depth"

329: Which 'extant ice shelves' are being referred too? (FYI: Some southwest AP ice shelves, specifically George VI and Wilkins have had lots of melt and surface ponding in recent years. E.g.

see Banwell et al 2021).
[Author Response] We have change the manuscript to say "extant eastern AP ice shelves"

341: Again, remove mention of critical 1 m lake depth. 358: Clarify that AP ice shelves are beginning referred to here.
[Author Response] We have altered the manuscript to no longer mention a critical lake depth. "long period ocean swells to trigger large-scale hydrofracture cascades on the LBIS and possibly LAIS…" We have also clarified which ice shelves we mean.

360: what 'region'? And surely fohn winds should be mentioned in this sentence too? What about future fohn winds?
[Author Response] We have altered the sentence to clarify our meaning and mention fohn winds. "Nevertheless, this research highlights a new understanding behind föhn melt mechanisms for ice shelf collapse and suggests that SCAR inlet and the LCIS may remain stable so long as surface liquid water from melt and precipitation remains within historical bounds".

Figures

Figure 1: Clarify that the LAIS and LBIS in the figure are no longer present, and perhaps give their collapse dates in the first sentence of the caption.
[Author Response] We have added clarification of which ice shelves collapsed as well as when collapse occurred.

Figure 3: In the caption, I think 'graph' would be a more appropriate word than 'curve'.
[Author Response] We have changed the manuscript.

Figure 6: What time period are the data for?
[Author Response] We have added the data time period

Additional references

Banwell, A. F., Datta, R. T., Dell, R. L., Moussavi, M., Brucker, L., Picard, G., Shuman, C. A., and Stevens, L. A. The 32-year record-high surface melt in 2019/2020 on the northern George VI Ice Shelf, Antarctic Peninsula, The Cryosphere, 15, 909–925, https://doi.org/10.5194/tc-15-909-2021, 2021.

Hubbard, B. et al. Massive subsurface ice formed by refreezing of ice-shelf melt ponds. Nat. Commun. 7:11897 doi: 10.1038/ncomms11897 (2016).

---

## Author Response (AR3)

Point by Point Response to Reviewers

Thank you for your comments and suggestions. We believe this manuscript has improved significantly with your suggestions and we sincerely appreciate your valuable contributions. We have addressed your comments below marked with [Author Response].

Dear Authors,
Thanks for this revised version with successfully deals with all the reviewers' remarks. I have only one concern about the new and relevant sea ice extent analysis.

- Section 2.4 (lines 141-149) At which time step have you performed the correlation ? Mean summer sea ice extent vs mean summer foehn occurrence ? It not clear. Moreover, the considered area should be plotted on Fig5c. What is the sea area considered here for the 3 Larsen? Finally, as the aim here is to show the correlation between foehn and sea ice, why did you limit your statistics to 1979-2002? Why do not consider here also the more recent summers?
[Author Response] Thank you for bringing this to our attention. We have clarified the manuscript to say we correlate summer fohn occurrence to summer sea ice concentration. We have also added boxes to Figure 5c to show the location of the sea ice study regions per ice shelf. As far as the correlation analysis of fohn to sea ice, we mis-typed the date range in the manuscript. We already did the correlation analysis for 1979-2018.

- line 213 I think it is rather "föhn wind occurrence THAN air temperature".
[Author Response] Thank you, we have fixed the manuscript.

- Fig 5: The period considered (1979-2002) as well as how are taken the sea ice contraction are missing in the legend. Idem on which area is taken the summer RACMO based air temperature? On the sea ice pixels only? Moreover, in this figure as well as in Figs 2 and 6, the dashed contours of lat/lon are missing while they are there in Fig 3 for example. Finally, in Fig 5a/b, I suggest to add a regression line for the 3 Larsen areas.
[Author Response] We have clarified the manuscript to identify the study region more clearly. We have added lat/lon lined to figure 6, however we felt that the lat/lon lines on Figure 2 would detract from the figure so we did not include them. We also added regression lines to Figure 5a/b to better show the correlation.